# Molecular condensation of the CO/NF-YB/NF-YC/ *FT* complex gates floral transition in *Arabidopsis*

Xiang Huang [ID] [1,2,5], Zhiming Ma [ID] [3,5], Danxia He[3], Xiao Han[3], Xu Liu [ID] [1,2], Qiong Dong[1,2], Cuirong Tan[1,2], Bin Yu[1,2], Tiedong Sun[3], Lars Nordenskiöld [ID] [3], Lanyuan Lu[3], Yansong Miao [ID] [3,4 ✉] & Xingliang Hou [ID] [1,2 ✉]

## Abstract

The plant master photoperiodic regulator CONSTANS (CO) interacts with Nuclear Factor-Y subunits B2 (NF-YB2) and C9 (NF-YC9) and transcriptionally activates the florigen gene *FLOWERING LOCUS T* (*FT*), regulating floral transition. However, the molecular mechanism of the functional four-component complex assembly in the nucleus remains elusive. We report that co-phase separation of CO with NF-YB2/NF-YC9/*FT* precisely controls heterogeneous CO assembly and *FT* transcriptional activation. In response to light signals, CO proteins form functional percolation clusters from a diffuse distribution in a B-box-motif-dependent manner. Multivalent coassembly with NF-YC9 and NF-YB2 prevents inhibitory condensate formation and is necessary to maintain proper CO assembly and material properties. The intrinsically disordered region (IDR) of NF-YC9, containing a polyglutamine motif, fine-tunes the functional properties of CO/NF-YB/NF-YC condensates. Specific *FT* promoter recognition with polyelectrolyte partitioning also enables the fluidic functional properties of CO/NF-YB/NF-YC/ *FT* condensates. Our findings offer novel insights into the tunable macromolecular condensation of the CO/NF-YB/NF-YC/*FT* complex in controlling flowering in the photoperiod control.

**Keywords** Floral Transition; CO/NF-Y; Macromolecular Condensation; Photoperiodic Regulation
**Subject Categories** Chromatin, Transcription & Genomics; Plant Biology

## Introduction

Spatiotemporally regulated macromolecular assembly and their formation of biomolecular condensates (BMCs) are fundamental mechanisms underlying diverse signal transduction pathways for development and resilience (Banani et al, 2017). Specific scaffolding and client factors determine the assembly pattern and BMC formation in the cytoplasm (Brangwynne et al, 2009; Zavaliev et al, 2020), nucleoplasm (Fang et al, 2019; Sabari et al, 2018), or a lipid membrane-associated form (Zeng et al, 2019), depending on the molecular grammar of participating constituents. The regulation of the expression of certain genes is vital for the determination of cell identity (Whyte et al, 2013) and for a proper response to environmental stimuli (Hazen et al, 2003), in which transcription factors have been suggested to play vital roles through phase separation (Banani et al, 2016; Boija et al, 2018; Wang et al, 2019).

Temperature and light are the main environmental stimuli that control plant flowering (Andrés and Coupland, 2012; Mouradov et al, 2002). Exposure to low temperatures can accelerate plant flowering in the vernalization pathway (Michaels and Amasino, 2000). Recent studies have reported that the transcriptional expression and RNA processing of *FLOWERING LOCUS C* and its antisense transcript (*COOLAIR*), the major target genes in the vernalization pathway, are regulated by phase-separated regulatory condensates (Fang et al, 2019; Zhang et al, 2022; Zhu et al, 2021). The duration of the daily light period (photoperiod) is also used as an important environmental cue by plants to control flowering time (Andrés and Coupland, 2012; Shim et al, 2017). In the photoperiod pathway, CONSTANS (CO) plays a central regulatory role that integrates various external and internal signals to activate the florigen gene *FLOWERING LOCUS T* (*FT*), which is closely associated with plant flowering (Shim et al, 2017; Tiwari et al, 2010). Studies have shed light on the general mechanism whereby CO specifically regulates the *FT* gene. The NF-YC and NF-YB proteins form a heterodimer in the cytoplasm, which are then translocated to the nucleus, where CO interacts with NF-YB/NF-YC through its C-terminal conserved CONSTANS, CONSTANS-LIKE, TOC1 (CCT) domain to form a heterotrimeric complex (Frontini et al, 2004; Gnesutta et al, 2017; Gnesutta et al, 2018; Kahle et al, 2005). The CO/NF-YB/NF-YC complex specifically recognizes multiple *cis*-elements in the *FT* promoter (Lv et al, 2021). In addition to the CCT domain, the N-terminal tandem B-box domains of CO contribute to the CO oligomeric configuration required for *FT* activation (Zeng et al, 2022). In addition, other subunits, such as NF-YA, can replace CO and physically associate with NF-YB/NF-YC to form nuclear factor Y (NF-Y) complexes, which are conserved combinatorial transcription factors in

[1]Guangdong Provincial Key Laboratory of Applied Botany & State Key Laboratory of Plant Diversity and Specialty Crops, South China Botanical Garden, Chinese Academy of Sciences, Guangzhou, China. [2]University of the Chinese Academy of Sciences, Beijing, China. [3]School of Biological Sciences, Nanyang Technological University, Singapore 637551, Singapore. [4]Institute for Digital Molecular Analytics and Science, Nanyang Technological University, Singapore 636921, Singapore. [5]These authors contributed equally: Xiang Huang, Zhiming Ma. ✉E-mail: yansongm@ntu.edu.sg; houxl@scib.ac.cn

eukaryotes (Dolfini et al, 2012; Nardini et al, 2013). In mammals, the NF-Y complex consists of the single gene-encoded subunits NF-YA, NF-YB, and NF-YC (Maity, 2017), while in plants, there are multiple members of each category of NF-Y subunits (Siefers et al, 2009). For instance, NF-YC3, NF-YC4, and NF-YC9 have overlapping functions in photoperiod-dependent *Arabidopsis* flowering (Kumimoto et al, 2010). Despite the known partnership of CO and associated factors, the molecular mechanisms underlying the floral transition mediated by their precise inter- or intramolecular interactions for the functional assembly of the CO complex, which determine CO activities, remain unknown.

Our study reveals that a multicomponent assembly consisting of CO, NF-YB, NF-YC, and *FT* can form functional CO assemblies that undergo molecular condensation, enabling the regulation of *FT* transcriptional activation. Through this process, CO proteins transition from an inactive, diffuse state, to active, nano-meter scale liquid functional condensates, facilitated by the B-box region of CO. These condensates' fluidic material properties are crucial for activating *FT*, achieved by the synergistic partition of NF-YB and NF-YC. Each uses distinct associative interactions and material properties to guide the functional assembly of the CO complex. The absence of NF-YC results in the formation of slow-diffusive CO, while the lack of NF-YB's fluidic ensemble significantly impairs CO's ability to activate *FT* and induce flowering. Furthermore, the optimal number of polyglutamine (polyQ) repeats within the intrinsically disordered region (IDR) of NF-YC9 fine tunes the functional fluidic properties of the CO/NF-YB/NF-YC/*FT* condensates. The mutation of NF-YC9 polyQ impairs the activation of the *FT* gene and normal flowering. Additionally, we found that *FT* can promote the dynamic properties of multicomponent condensates in vitro. Our study demonstrates the importance of higher-order structural organization in gene regulation and establishes a new regulatory framework for the transcriptional control of the plant floral transition process through the tunable condensation of CO/NF-YB2/NF-YC9 and *FT* to form macro-molecular complexes.

## Results

### CO condensates require NF-YCs to maintain a functional state for floral transition

CO, NF-YC, and NF-YB were previously shown to form a heterotrimeric complex that recognizes the CO response elements (COREs) of the *FT* promoter (Gnesutta et al, 2017). To elucidate the molecular mechanism by which the CO/NF-YB/NF-YC complex modulates *FT* transcription, we transformed *35S:mCherry-CO* into Col-0 and the *nf-yc3 nf-yc4 nf-yc9* triple mutant (*ycT*) and *35S:NF-YC9-GFP* into *ycT*, respectively, and crossed the resulting plants to generate *35S:mCherry-CO 35S:NF-YC9-GFP ycT* plants. Consistent with previous studies (Hou et al, 2014; Kumimoto et al, 2010), *35S:mCherry-CO* showed a strong early-flowering phenotype in Col-0, whereas *35S:mCherry-CO ycT* maintained a severe late-flowering phenotype similar to *ycT* (Fig. 1A,B). However, NF-YC9 could significantly rescue the late flowering of *35S:mCherry-CO ycT* (Fig. 1A,B), supporting the notion that the function of CO in flowering promotion requires NF-YC. As the CO protein is stabilized by light but degraded in

darkness (Suárez-López et al, 2001; Valverde et al, 2004), we observed the CO localization pattern in the nucleus of 5-day-old transgenic seedlings that were pretreated with 15 h of darkness and then transferred to light. CO fluorescent signals gradually accumulated in *35S:mCherry-CO*, *35S:mCherry-CO ycT*, and *35S:mCherry-CO 35S:NF-YC9-GFP ycT* roots once the plants were exposed to light (Fig. EV1A). Cluster index analysis (Tran et al, 2020) showed that CO proteins started to assemble from the diffuse state to form condensates after 1 h under light (Fig. EV1B). Interestingly, CO condensates were concentrated into larger foci assemblies in the nuclei of *35S:mCherry-CO ycT* than in those of *35S:mCherry-CO* and *35S:mCherry-CO 35S:NF-YC9-GFP ycT* after 2 h of exposure to light (Fig. EV1B), indicating that NF-YCs fine tunes CO assembly status. Next, we examined whether CO and NF-YC9 colocalized in the CO condensates. mCherry-CO signals formed clear condensation in the nuclei of *35S:mCherry-CO ycT* cells (Fig. 1C), while NF-YC9-GFPs were diffusely distributed in the nuclei of *35S:NF-YC9-GFP ycT* cells (Fig. EV1C). Notably, mCherry-CO and NF-YC9-GFP colocalized and assembled into smaller foci in *35S:mCherry-CO 35S:NF-YC9-GFP ycT* plants (Figs. 1D and EV1B). These results suggest that NF-YC9 may be essential for maintaining the proper size of CO assemblies.

To better understand how the CO/NF-Y ensemble, starting from a monomeric state, regulates its functionality in the transcriptional activation of *FT*, we conducted molecular simulations to assess CO's binding affinity on COREs of the *FT* promoter. In this study, structures of CO in various oligomeric states, derived from AlphaFold2, were integrated with resolved CO-CCT-NF-Y (CO-CCT domain in complex with HFD domains of NF-YB3 and NF-YC4) structures (Lv et al, 2021) to form multimeric CO-NF-Y complexes (Fig. EV1D). Our simulations indicated that the monomeric CO-NF-Y complex exhibited comparable binding abilities across four known sites (Lv et al, 2021) (Fig. EV1E). Further systematic examination of *FT* promoter association with a series of oligomeric CO-NF-YC complexes showed a clear positive correlation between the number of bound CO-NF-YC on *FT* fragments and the oligomeric state of the complexes (Fig. 1E). These results suggest that the enhanced assembly of CO-NF-YC complexes increases binding affinity, and consequently higher activities, to the *FT* promoter compared to monomeric CO, likely due to the reduced diffusion entropy of the oligomeric CO.

To further assess whether NF-YC9 mediates the transition property of CO condensates in vivo, we employed fluorescence recovery after photobleaching (FRAP) to determine the dynamics of CO condensates in plants. The fluorescence intensity of CO condensates was rapidly redistributed from the unbleached area to the bleached area after photobleaching in either *35S:mCherry-CO* or *35S:mCherry-CO 35S:NF-YC9-GFP ycT*, while the mCherry-CO redistribution was remarkably retarded in the absence of NF-YC9-GFP (Fig. 1F,G). These findings indicate that NF-YC9 plays an important role in maintaining the proper molecular dynamics of CO assemblies and prevents undesired slow-diffusive CO condensates. Here, the slow-diffusive condensates exhibit reversible properties characterized by slow diffusion and kinetic trapping at non-equilibrium states, in contrast to the dynamic small clusters that form in the presence of NF-YC. Next, we investigated whether NF-YC9-fluidized CO condensates offer a better *FT* transcription in plants. Quantitative RT-PCR (qRT-PCR) analyses showed that overexpression of *CO* failed to promote *FT* transcription in the *ycT*

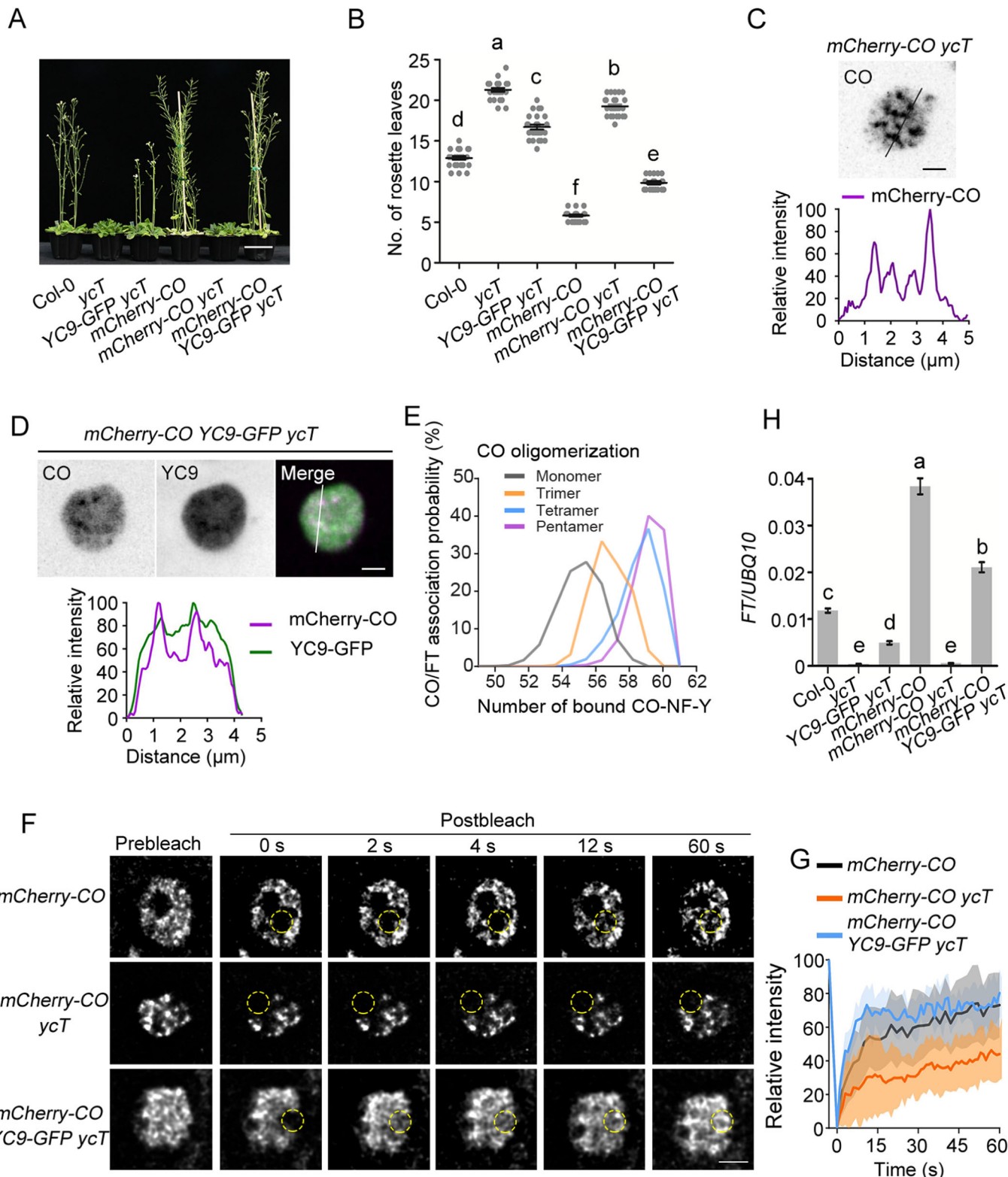

◄ **Figure 1.  The functional fluidic state of CO condensates depends on NF-YCs in plants.**

(A) Flowering phenotype of representative 28-day-old Col-0, *ycT*, *35S:NF-YC9-GFP ycT*, *35S:mCherry-CO*, *35S:mCherry-CO ycT*, and *35S:mCherry-CO 35S:NF-YC9-GFP ycT*. Scale bar, 5 cm. (B) Comparison of rosette leaf number for Col-0, *ycT*, *35S:NF-YC9-GFP ycT*, *35S:mCherry-CO*, *35S:mCherry-CO ycT*, and *35S:mCherry-CO 35S:NF-YC9-GFP ycT*. Error bars, means ± SD; $n = 25$ seedlings. Different lowercase letters above the columns indicate the significant difference among different groups (one-way ANOVA, $P < 0.0001$). (C) mCherry-CO formed obvious condensates in 5-day-old *35S:mCherry-CO ycT* seedlings, pretreated with 15 h darkness and then transferred to light for 2 h. Scale bars, 2 µm. (D) Colocalization of mCherry-CO and NF-YC9-GFP in smaller condensates in 5-day-old *35S:mCherry-CO 35S:NF-YC9-GFP ycT* seedlings, pretreated with 15 h darkness and then transferred to light for 2 h. Scale bars, 2 µm. (E) The distribution of the number of bound CO-NF-Y on *FT* promotor fragments in each indicated oligomerized CO-NF-Y complexes. The method for doing molecular simulations of oligomerized CO-NF-Y complex binding on *FT* promotor fragments containing the four binding motifs of CO was shown in Fig. EV1D. (F) FRAP assay of CO condensates in *35S:mCherry-CO*, *35S:mCherry-CO ycT* and *35S:mCherry-CO 35S:NF-YC9-GFP ycT*. Deconvolution was applied to denoise the image due to weaker signal was obtained in time-series imaging as lower exposure was set for long-term image. Time indicates the duration after the photobleaching pulse. Dash circles indicate the bleached areas. Scale bar, 2 µm. (G) FRAP recovery plot of CO nuclear condensates in *35S:mCherry-CO*, *35S:mCherry-CO ycT* and *35S:mCherry-CO 35S:NF-YC9-GFP ycT*. Solid lines and shaded area represent means ± SD; $n = 10$. (H) qRT-PCR analysis of the *FT* expression in 5-day-old transgenic seedlings of Col-0, *ycT*, *35S:NF-YC9-GFP ycT*, *35S:mCherry-CO*, *35S:mCherry-CO ycT*, and *35S:mCherry-CO 35S:NF-YC9-GFP ycT*. Gene expression levels are normalized to *UBQ10*, acting as an internal control. Error bars, means ± SD, $n = 3$. Different lowercase letters above the columns indicate the significant difference among different groups (one-way ANOVA, $P < 0.0001$). Source data are available online for this figure.

background. However, CO dramatically increased *FT* transcription in the presence of NF-YC9 (Fig. 1H), which was in line with the observations of the flowering phenotype (Fig. 1A,B). These findings suggest that NF-YC-dependent CO fluidic condensates are strongly linked to *FT* gene expression, crucial for floral transition.

## NF-YB/NF-YC facilitate the functional state of CO condensates for *FT* activation

It is known that CO is a transcriptional factor that regulates multiple gene expression (Samach et al, 2000), such as *FT* and *SOC1* (Hou et al, 2014), while *FT* is an essential downstream regulator for floral transition. To better understand how the interaction dynamics between components of the CO/NF-YB/NF-YC complex influence *FT* transcription, we analyzed a wide range of expressions by combining CO, NF-YC9, and NF-YB2 in transient expression experiments using *Arabidopsis* mesophyll protoplasts. All three components exhibited nuclear localization, either diffusing in the nucleoplasm or condensing into subnuclear foci. To quantitatively assess the assembly status of CO, enhancing our grasp of its material properties-dependent biochemical activities, we analyzed the overall CO signal distribution across different co-expressions of CO, NF-YC9, and NF-YB2. We then measured the variation in CO signal within each individual nucleus (Figs. 2A and EV2A). Here, higher variance in CO signal suggests greater local accumulation of CO. We identified that the CO signal peaks primarily in three forms: diffuse (16%), spherical condensation (36%), and amorphous aggregate (48%) (Figs. 2B and EV2B), indicating stages of increasing CO accumulation. In FRAP assay, the amorphous aggregates showed no fluorescence recovery from surrounding areas (Fig. EV2C), leading us to describe these as irreversible aggregates, which better differentiate their physical properties. Observations indicated that CO predominantly formed irreversible aggregates in the nucleus (Fig. 2A). In contrast, NF-YC9 or NF-YB2 alone displayed a uniform and diffuse distribution across the cell in both the cytoplasm and nucleoplasm (Fig. EV2D). We then explored whether NF-YC9 and/or NF-YB2 affect the assembly status of nuclear CO proteins. Strikingly, when co-expressed with both NF-YC9 and NF-YB2, CO assembly shifted from irreversible aggregates to predominantly spherical condensates (60%) (Fig. 2A). On the other hand, we examined whether the CO proteins affect the subnuclear localization of NF-YC9 and NF-

YB2. Neither the expression of NF-YC9 or NF-YB2 alone nor their co-expression resulted in the formation of nuclear condensates in *Arabidopsis* protoplasts (Fig. EV2D,E). The addition of only NF-YC did not sufficiently fluidize CO assembly, and the addition of only NF-YB2 failed to disrupt tight self-aggregating assemblies. This indicates that NF-YC9 and NF-YB2 work synergistically to promote the proper functional and fluid assembly of the CO complex (Figs. 2A and EV2F).

The two B-box motifs of CO mediate CO oligomerization and the subsequent activation of *FT* transcription in plants (Zeng et al, 2022). Without the B-box motif, CO is non-functional regarding *FT* activation and promoting flowering (Zeng et al, 2022). To explore the CO assembly under such non-functional states, we constructed a truncated version of CO for further analysis. CO-ΔB-box, B-boxes domain-deleted variant of CO, displayed diffuse signal that was unable to assemble into nuclear condensates regardless of whether the NF-YC and NF-YB components were present (Fig. EV2G,H), suggesting a B-box motif-dependent assembly of CO condensates, which is indispensable for its functionality.

We thus next focused on studying those spherical condensates of CO and were motivated to ask whether and how the biophysical properties of CO/NF-YB/NF-YC complex condensation regulate proper CO function in the transcriptional activation of *FT*. We examined the population distribution of CO assemblies (Fig. 2A), the fluidity of different CO-condensates, and the corresponding activation of *FT*. In the absence of NF-YC9 and NF-YB2, the majority (76%) of the spherical condensates of the CO complex exhibited a slow-diffusive state and showed minimal molecular exchange with the surrounding diluted phase in the FRAP assay (Fig. 2C–E; Appendix Fig. S1). This observation indicates a non-equilibrium state with restricted dynamics and exchange rates of CO molecules between dense and dilute phases. However, with additional NF-YB2/NF-YC9 components, except a small fraction (5%) of CO exhibit slow diffusion, the majority (95%) of spherical condensates showed remarkable recovery after photobleaching, displaying fluidic properties (Fig. EV2C; Appendix Fig. S1). Thus, these results further confirm that the coassembly of NF-YB2/NF-YC9 into CO assemblies maintains CO condensates in a highly dynamic state, whereas they would otherwise transition into slow-diffusive condensates. Next, we examined how the assembly and condensation status of the

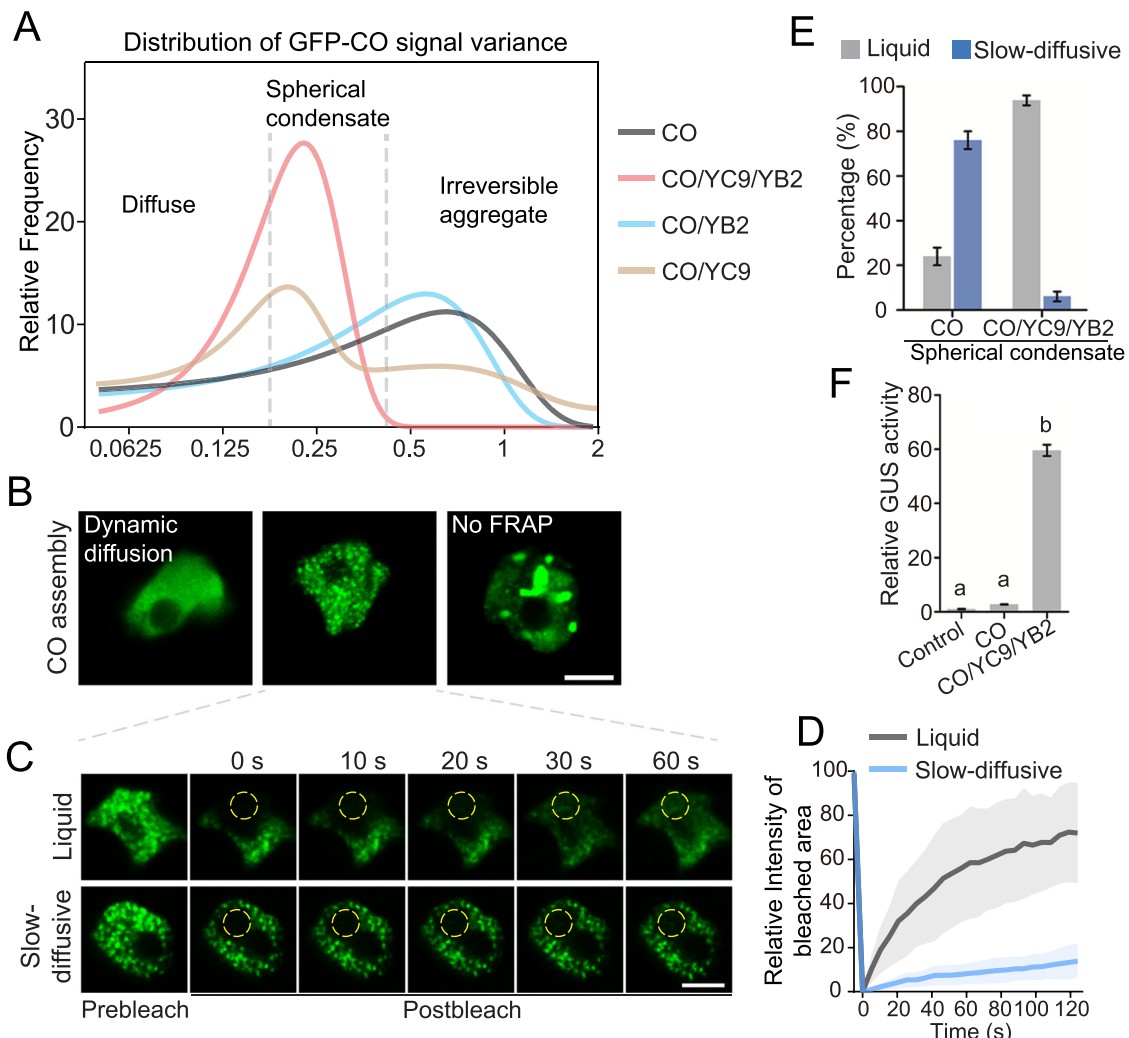

**Figure 2.  NF-YC9 and NF-YB2 preserve liquid CO condensate properties for *FT* expression.**

(A) The distribution of GFP-CO signal variance under indicated combinations of transient expression, subjecting to multipeak fitting. The distribution peaks highlight the major population of CO assembly at either 'Diffuse' (signal variance <0.21), 'Spherical condensate' (signal variance 0.21–0.45), or 'Irreversible aggregate' (signal variance >0.45), respectively, as classified in Fig. EV2A,B. $n = 75$. (B) The three representative CO assembly status as in A, which were obtained from the transient expression of GFP-CO alone in *Arabidopsis* protoplasts. Scale bars, 5 μm. (C) FRAP assay of spherical condensates of CO as shown in (B), indicating either liquid or slow-diffusive material properties of those CO condensation. Time indicates the duration after the photobleaching pulse. Dash circles indicate the bleached areas. Scale bar, 5 μm. (D) FRAP recovery curves from the intensity quantification of the bleached area in (C). Five independent observations were plotted for each category. Solid lines and shaded areas represent means ± SD. (E) Quantification of the distribution ratios of CO spherical condensates showing either liquid or slow-diffusive material property in transient expression of CO alone or CO/YC9/YB2 co-expression in *Arabidopsis* protoplasts. Error bars, means ± SD; $n = 50$ nuclei. (F) Transient expression assay indicating that the expression of *FT* was synergistically promoted by CO, NF-YC9, and NF-YB2 in *Arabidopsis* protoplasts. Error bars, means ± SD, $n = 3$. Different lowercase letters above the columns indicate the significant difference among different groups (one-way ANOVA, $P < 0.0001$). Source data are available online for this figure.

CO/NF-YB2/NF-YC9 complex affects its activities in the transcriptional activation of *FT*. In the presence of both NF-YC9 and NF-YB2, the CO complex dramatically induced *FT* promoter-driven GUS activity in *Arabidopsis* protoplasts (Fig. 2F). These findings suggest a likely mechanistic relationship between the assembly status and material property-dependent function of CO/NF-YB2/NF-YC9 condensates in *FT*-expression (Fig. 1). The fluidic assembly of CO/NF-YB2/NF-YC9 provides an optimized biophysical environment for CO function for *FT* activation, whereas slow-diffusive condensates restrict CO activities.

## CO, NF-YC9, and NF-YB2 undergo macromolecular condensation in vitro

Next, we were motivated to understand how multicomponent partitions control CO complex assembly and function. We reconstituted the molecular condensates of CO, NF-YC9, and NF-YB2 in vitro via bacterial produced recombinant mCherry-CO, NF-YC9-GFP, and NF-YB2 proteins (Appendix Fig. S2A). Both mCherry-CO and NF-YC9-GFP eluted early in gel filtration chromatography, indicating that they form high oligomeric states. However, they remained soluble, as evidenced by their presence in

the supernatant after high-speed ultracentrifugation at the tested concentrations (Appendix Fig. S2B), demonstrating their solubility in high-oligomeric states. We found that the homotypic assembly of mCherry-CO or NF-YC9-GFP resulted in amorphous condensates, showing limited molecular dynamics based on a lack of recovery in the FRAP assay (Fig. EV3A,B). In contrast, highly intrinsically disordered NF-YB2 exhibited typical liquid-liquid phase separation by forming spherical droplets in a protein concentration- and ionic strength-dependent manner. In line with this, the FRAP assay also revealed recovery of the NF-YB2 signal over time, indicating either molecular exchange between the dense and dilute phases or fluidic diffusion within the droplet (Fig. EV3A–D). The results indicate that homotypic CO or NF-YC9 assemblies exhibit limited conformational flexibility, while NF-YB2 shows a highly flexible conformation and fluidic properties.

Next, to better understand the robust formation of CO/NF-YC9/NF-YB2 assemblies, we performed systematic characterization by surface plasmon resonance (SPR) to map their inter- and intramolecular interactions. The SPR sensorgrams showed robust inter- and intramolecular interactions for all examined interactions of the CO, NF-YC9, and NF-YB2 proteins (Fig. 3A). The SPR results showed that CO presented a high first-order affinity with a Kd of approximately 50 nM and slow disassociation, suggesting high valency within homotypic assemblies, similar to the CO/NF-YC9 complex and NF-YC9 homotypic assembly. Interestingly, NF-YB2 also exhibited a high first-order interaction, with a Kd of ~20 nM, but presented a faster dissociation rate than CO, NF-YC9, or CO/NF-YC9 (Fig. 3A). When NF-YB2 was flowed on CO, NF-YC9, or CO/NF-YC9, relatively faster dissociation could also be observed at high concentrations of NF-YB. Such physicochemical properties of NF-YB2 are critical for modulating the robust assembly and material properties of the CO/NF-YB2/NF-YC9 complex through their tri-component intermolecular interactions, with additional flexibility during macromolecular condensation.

Informed by the tri-component interactions observed in bulk solutions, we were motivated to reconstruct these complex assemblies in vitro at the nanoscale, starting from nanomolar concentrations. This approach is crucial as transcription factors typically exist in low quantities within cells, and such reconstruction at the single particle level allows us to explore the lowest possible physiological concentration range to elucidate their nanometer-scale interactions. We began by examining bi- or tri-component co-assembly of CO, NF-YC9, and NF-YB2 using TIRF microscopic observation (Figs. 3B and EV3E), which highlighted the intermolecular interactions among these components at nanomolar scales although low proportion of co-assembly was observed, suggesting a kinetic unstable interaction at the nucleation stage of multicomponent assembly. We quantitatively analyzed the CO assembly status through single-particle intensity quantification. NF-YC9 alone or NF-YB2 alone did not significantly affect CO assembly across serially titrated concentrations (Fig. EV3F,G). However, introducing additional serial concentrations of NF-YB2, specifically in the presence of NF-YC9, markedly reduced CO foci intensity, suggesting a fluidization of CO assemblies (Fig. 3C,D). These findings not only corroborate in vivo observations of CO condensation assembly (Figs. 2A,B and EV3D–F) and in vitro SPR sensorgrams (Fig. 3A), but also provide direct evidence of their interaction at extremely low concentrations on a single-particle

level. This supports the effective synergy of NF-YB2 and NF-YC9 in modulating the material properties of CO complexes under physiologically relevant conditions.

We then examined their fluidic material properties of their condensates at micro-molar scale. The in vitro reconstitution of bi- or tri-component condensates showed the colocalization of different combinations of mCherry-CO, NF-YC9-GFP, and NF-YB2 in all cases. However, neither amorphous CO/NF-YC9 condensates nor spherical and heterogeneous CO/NF-YB2, NF-YC9/NF-YB2, or CO/NF-YB2/NF-YC9 assemblies exhibited obvious diffusivity (Fig. EV3H–J), which is critical for *FT* activation as show in Fig. 2. We next considered to reconstitute and examine more physiologically relevant assemblies in the presence of the *FT* promoter. We additionally introduced plasmid DNA containing the 5.7-kb *FT* promoter sequence to mimic triple component assembly and phase separation of the CO/NF-YB2/NF-YC9/*FT* complex. Although the *FT* promoter sequence did not enhance molecular dynamics within single-component homotypic assemblies (Fig. EV3K,L), it significantly promoted the FRAP recovery of CO in the triple-component complex without changing the CO condensation size (Figs. 3E–G and EV3M), implying fluidity enhancement via the formation of DNA polyelectrolyte condensates. In addition, such DNA-enhanced CO fluidity was *FT* promoter specific, as control DNA without the *FT* promoter sequence did not change the complex material properties (Fig. 3E–G). Taken together, our triple component reconstitution recalculates the essential *FT* promoter recognition-dependent molecular dynamics and functions of CO/NF-YB2/NF-YC9/*FT* condensates.

## IDR domains of NF-YC9 facilitate the functional condensation of CO

IDRs contribute to dynamic and cooperative multivalent interactions during biomolecular phase separation (Alberti et al, 2019; Banani et al, 2017). We sought to decipher how the essential bridging factor NF-YC9 regulates multicomponent condensation formation and CO activities. We analyzed the protein sequences of *Arabidopsis* NF-YC family members and found that all NF-YCs contained one or two IDRs (Appendix Fig. S3A). IDRs also existed in CO and NF-YB2 (Appendix Fig. S3B). We then constructed several truncated NF-YC9 variants with the deletion of IDR1 (YC9-ΔIDR1), IDR2 (YC9-ΔIDR2), or both (YC9-ΔIDR1&2) (Fig. EV4A) for yeast two-hybrid assays. Similar to full-length NF-YC9, the NF-YC9-ΔIDR variants maintained their interaction with CO (Fig. EV4B). In addition, cell imaging showed that, similar to CO/NF-YB2/NF-YC9 (Figs. 2B and EV2F), the CO, NF-YB2, and NF-YC9-ΔIDR variants colocalized in the same condensate structures, which exhibited mostly as 40% of diffuse signal and 60% of spherical condensation (Fig. EV4C,D). These results indicate that the folded middle region of NF-YC9 (55–166 aa) is able to interact with other components for proper complex assembly.

We further examined the role of NF-YC9 IDRs in assembling functional CO complexes by characterizing in vivo CO dynamics. FRAP analysis showed that NF-YC9-ΔIDR variants were less efficient than full-length NF-YC9 in facilitating the transition of CO spherical condensates into the fluidic state. NF-YC9-ΔIDR variants all resulted in an increasing population of slow-diffusive

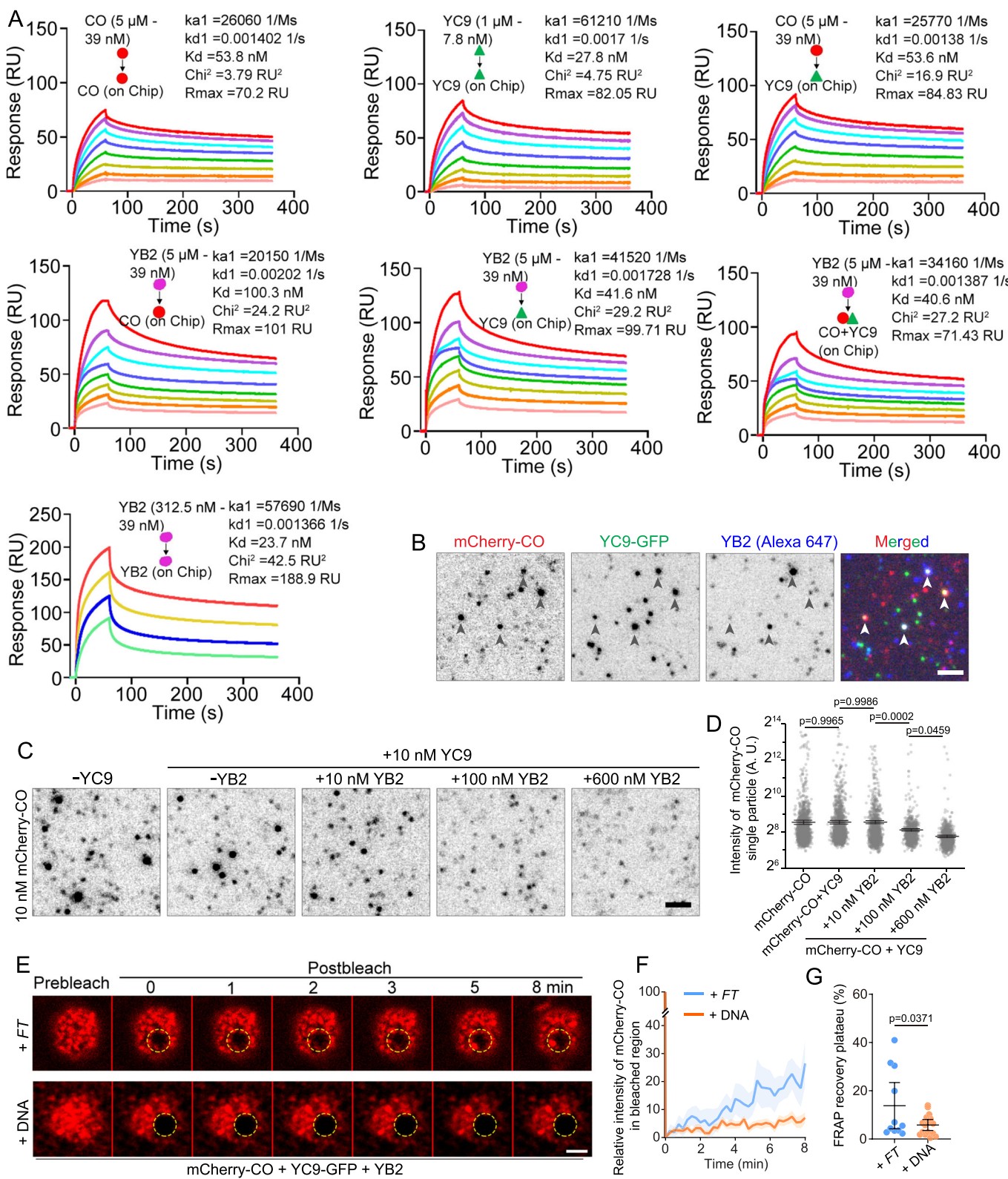

**Figure 3. Inter- and intramolecular multivalent interactions underlying CO/NF-YB2/NF-YC9/FT phase separation.**

(A) Representative SPR kenetics of the inter/intra molecular interaction of mCherry-CO, NF-YC9-GFP, and NF-YB2. Magentas cloud, green triangle, and red circle indicated NF-YB2, NF-YC9, and mCherry-CO proteins, respectively. Series two-times diluted proteins, with indicated highest and lowest concentrations, were used as analytes that flowed through the chip immobilized with indicated proteins. Binding parameters were generated by fitting the sensorgrams using a bivalent model. (B) Coassembly of CO, NF-YC9, and NF-YB2 as nano molar scale. 10 nM of recombinant mCherry-CO, NF-YC9-GFP and NF-YB2 (labeled Alexa fluor 647) was mixed in buffer of 20 mM HEPES, 150 mM NaCl, pH 7.4 for 5 min before being added on cover glass and then checked under TIRFM. Merged image shows the colocalization of those three components. Arrow heads indicate the colocalized foci and the related protein single particles in each channel. Scale bar, 2 μm. (C) Single particle images of 10 nM recombinant mCherry-CO mixed with 10 nM NF-YC9-GFP and further supplied with or without a series concentration of NF-YB2 as indicated. The proteins were mixed in the buffer of 20 mM HEPES, 150 mM NaCl, pH 7.4 for 5 min before being added on cover glass and then checked under TIRFM. Scale bar, 2 μm. (D) Quantification of the mCherry-CO single particle total intensity in (C). $n > 1000$ single particles in each chart. The middle lines indicate the mean values. Error bars represent 95% confidence intervals. Significant differences were determined by one-way ANOVA, $P < 0.0001$. (E) FRAP of mCherry-CO in the mixed droplets of 0.25 μM mCherry-CO/0.25 μM NF-YC9-GFP/15 μM NF-YB2/5 μg/mL FT and 0.25 μM mCherry-CO/0.25 μM NF-YC9-GFP/15 μM NF-YB2/ 5 μg/mL DNA, respectively. All the proteins were diluted in 20 mM HEPES, 150 mM NaCl, pH 7.4 buffer. Time indicates the duration after the photobleaching pulse. Dash circles indicate the bleached areas. Scale bar, 2 μm. (F) FRAP recovery plot of mCherry-CO in the mixed droplets of 0.25 μM mCherry-CO/0.25 μM NF-YC9-GFP/15 μM NF-YB2/5 μg/mL FT, $n = 14$, and 0.25 μM mCherry-CO/0.25 μM NF-YC9-GFP/15 μM NF-YB2/5 μg/mL DNA, $n = 19$, respectively. Solid lines and shaded area represent means ± SD. (G) Quantification of the mCherry-CO single recovery plateau in the bleached areas in (E). FRAP recovery plots in (F) were fitted with one one-phase decay model. The middle lines indicate the mean values. Error bars represent 95% confidence intervals. Significant difference was determined by Student's t test, $P = 0.0371$. Source data are available online for this figure.

CO (Fig. EV4E–G), revealing the important role of NF-YC9 IDRs in modulating the biophysical properties of CO/NF-YB2/NF-YC9 condensates. Accordingly, transgenic *Arabidopsis* plants expressing NF-YC9-ΔIDR variants were unable to fully compensate for the loss of full-length NF-YC9 function in *ycT*. All NF-YC9-ΔIDR variant-expressing plants failed to activate *FT* transcription and rescue the late flowering of *ycT* (Fig. EV4H–K). We attempted to use all IDR-truncated versions of NF-YC9, as shown in Fig. EV4A, to reconstitute the CO/NF-YB2/NF-YC9 co-assembly but were unable to produce soluble recombinant proteins for these variants. However, in vivo results indicate that the IDRs of NF-YC9 are crucial in optimizing the material properties of the CO/NF-YB2/ NF-YC9/*FT* complex, essential for proper floral transition.

## Length of polyQ repeats in NF-YC9 correlates with CO complex condensation

PolyQ repeats play a role in the assembly status and protein functionality of diverse macromolecular condensates, depending on the length of glutamine residues, the polyQ repeat pattern and the physicochemical environments surrounding polyQ (Press and Queitsch, 2017). We found that *Arabidopsis* flowering-related NF-YC1/4/3/9/2 (Petroni et al, 2012) shared a conserved polyQ repeat with similar lengths (6–7 tandem glutamines) within the N-terminal IDR (Fig. EV5A), implying an evolutionarily conserved function of polyQ repeats in the plant NF-YC family. AlphaFold3 prediction suggested that NF-YC9 polyQ is embedded in one of the two α-helices ligated with a flexible sequence (Fig. EV4B) (Abramson et al, 2024). To test whether the NF-YC9 polyQ repeat influences the functional condensation of the CO/NF-YB2/NF-YC9 complex, we constructed NF-YC9 variants with deletion of the polyQ repeat (YC9-0Q) or another longer polyQ repeat (YC9-37Q) (Fig. 4A). From the AlphaFold3 prediction, deleting or adding Q residue just shortened or extended the same α-helices in NF-YC9 (0Q) or NF-YC9 (37Q), respectively, but did not change the overall conformation of CO/ NF-Y/*FT* complex (Fig. EV5B). Similar to the NF-YC9-ΔIDR variants, both YC9-0Q and YC9-37Q still physically interacted with CO (Fig. EV5C) and colocalized with CO and NF-YB2 in the same condensates (Fig. 4B). In addition, although neither YC9-0Q nor YC9-37Q changed the proportions of different CO assemblies, which both showed diffuse or spherical condensates

with equal proportion (Fig. 4C), the material properties of spherical multicomponent condensates were impaired by the increase in slow-diffusive condensates (Figs. 4D and EV5D, E). These results suggest that NF-YC9 polyQ optimizes multivalent network formation within the spherical CO condensates, which maintains the functional fluidic state of CO.

We next tested whether CO/NF-YB2/NF-YC9-0Q or CO/NF-YB2/NF-YC9-37Q condensates with altered material properties influence CO function in plants. A transient analysis of *pFT:GUS* activity showed that the activation of *FT* was compromised in the multicomponent assemblies of CO, NF-YB2, and YC9-0Q (or YC9-37Q) compared with the assemblies of CO/NF-YB2/NF-YC9 (Fig. 4E). In addition, we transformed *35S:NF-YC9-0Q-GFP* and *35S:NF-YC9-37Q-GFP* into the *ycT* background (Fig. EV5F). Both of the transgenic *Arabidopsis* plants showed impaired *FT* transcription (Fig. 4F), and the constructs failed to rescue the late flowering of the *ycT* mutant (Figs. 4G and EV5G). These data indicate that the optimal NF-YC9 polyQ repeat plays an essential role in supporting CO biological function. To assess whether the polyQ repeat is a widespread characteristic of NF-YC proteins in different organisms, we aligned the amino acid sequences of NF-YC9 from *Arabidopsis* and other species. A similar polyQ length was observed in the IDRs of NF-YC9 homologs in other plant species, but not in the homologs of yeast, animals, and humans (Appendix Fig. S4). Therefore, we speculate that a certain polyQ length might be critical for NF-YC9 and its homologs to regulate phase separation in plants, but this remains to be further verified.

## Discussion

As one of the most important developmental events in plants, the floral transition proceeds under the control of multiple internal and external environmental signals (Parcy, 2005). Several main pathways exist to integrate signals for flowering control, including the photoperiod, vernalization, autonomous, ageing, gibberellin (GA), and thermosensory pathways (Banani et al, 2017; Capovilla et al, 2015). Emerging studies are showing that the phase separation of regulatory proteins contributes to flowering in the autonomous pathway (Fang et al, 2019) and the vernalization pathway (Zhu et al, 2021). Whether phase-separation-based regulation plays a role

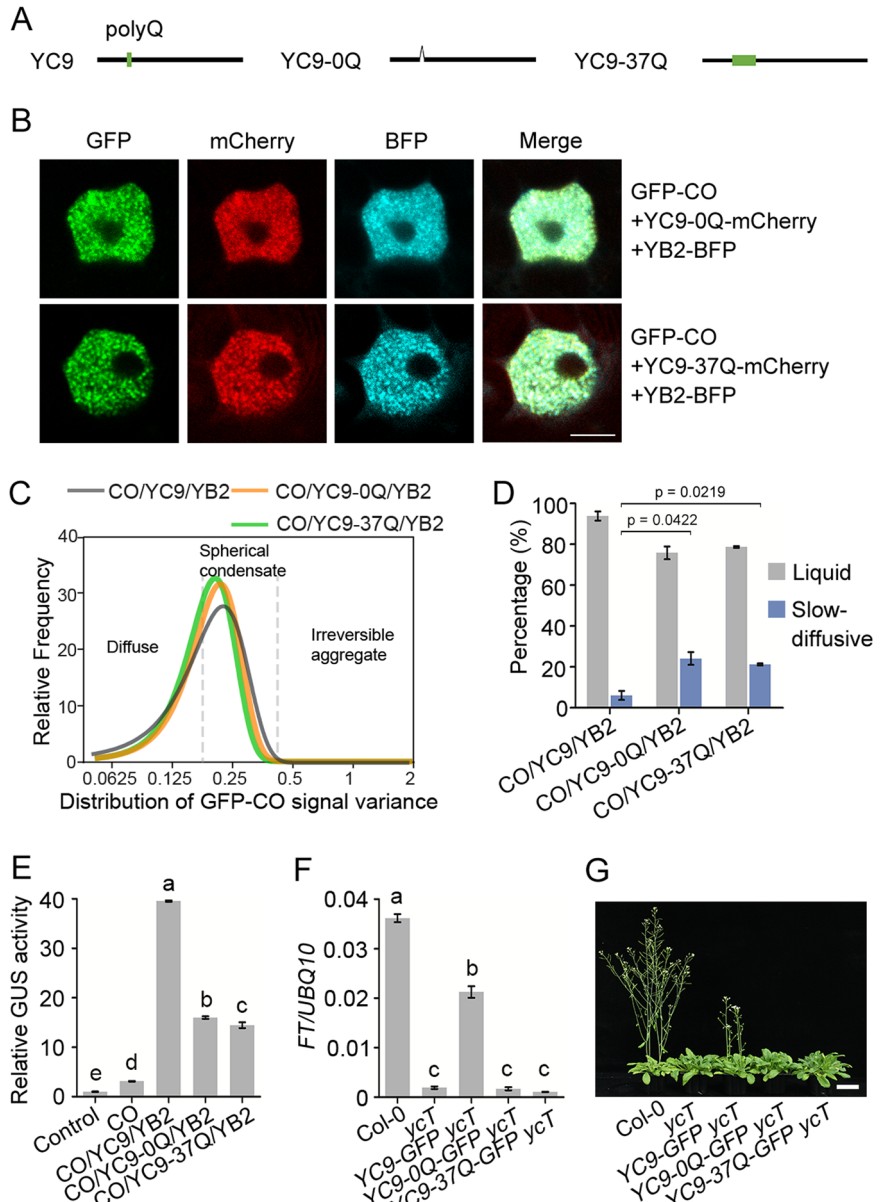

**Figure 4. The polyQ repeat in NF-YC9 IDR is essential for NF-YC9 mediating CO phase separation and CO function to control flowering time.**

(A) Schematic of *NF-YC9* and *NF-YC9-polyQ* mutant constructs, which are deleted polyQ repeat (*NF-YC9-0Q*) or with a longer polyQ repeat (*NF-YC9-37Q*). (B) Subcellular localization of GFP-CO, NF-YB2-BFP, and NF-YC9-0Q-mCherry (or NF-YC9-37Q-mCherry) in *Arabidopsis* protoplasts. Scale bars, 5 μm. (C) The distribution of GFP-CO signal variance under indicated combinations of transient expression, subjecting to Gaussian fitting. Dashes lines indicate the cut off on signal variance values to classify the CO assembly to diffuse, spherical condensation and irreversible aggregate. *n* = 75. (D) Quantification of the distribution ratios of CO spherical condensates showing either liquid of slow-diffusive material property in transient expression under combinations of GFP-CO, NF-YB2-BFP, and NF-YC9-mCherry (or NF-YC9-0Q/37Q-mCherry) in *Arabidopsis* protoplasts. Error bars, means ± SD; *n* = 50 nuclei. Significant differences were determined by Student's t test. (E) Transient expression assay of the *FT* promoter activity modulated with combinations of GFP-CO, NF-YB2-BFP, and NF-YC9-0Q-mCherry (or NF-YC9-37Q-mCherry) in *Arabidopsis* protoplasts. Error bars, means ± SD, *n* = 3. Different lowercase letters above the columns indicate the significant difference among different groups (one-way ANOVA, *P* < 0.0001). (F) qRT-PCR analysis of *FT* expression in 5-day-old seedlings of Col-0, *ycT*, *35S:NF-YC9-GFP ycT*, *35S:NF-YC9-0Q-GFP ycT*, and *35S:NF-YC9-37Q-GFP ycT*. Gene expression levels are normalized to *UBQ10*, acting as an internal control. Error bars, means ± SD, *n* = 3. Different lowercase letters above the columns indicate the significant difference among different groups (one-way ANOVA, *P* < 0.0001). (G) Flowering phenotype of Col-0, *ycT*, *35S:NF-YC9-GFP ycT*, *35S:NF-YC9-0Q-GFP ycT*, and *35S:NF-YC9-37Q-GFP ycT*. Scale bar, 5 cm. Source data are available online for this figure.

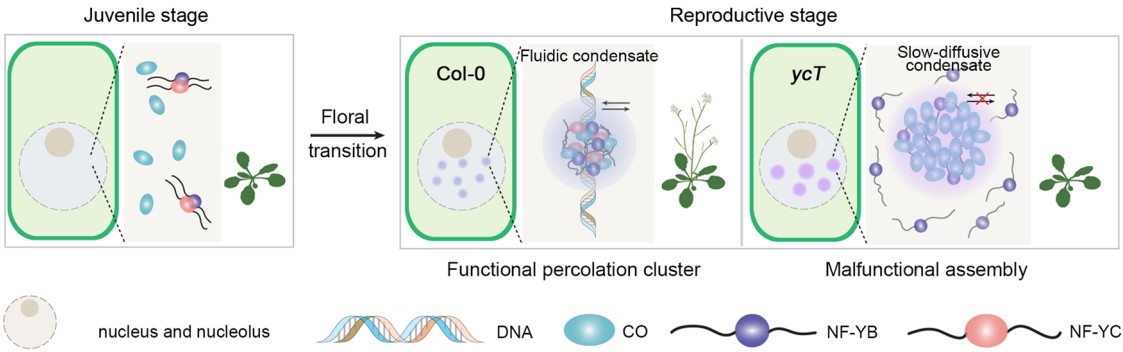

**Figure 5. A model of fine-tuned CO condensation in regulating *FT* expression and flowering.**

The condensation and function of CO are driven by multivalent interactions. Coassembly with NF-YC9, NF-YB2, and *FT* is essential for the formation of percolation clusters of CO condensates, which maintain functional fluidic properties required for activating *FT* transcription and triggering flowering transition. However, in the absence of NF-YC, CO proteins lead to over-assembly via strong self-interaction into slow-fluidic condensates, resulting in the improper partition of flexible NF-YB proteins and leading to the formation of malfunctional gelated condensates.

in the photoperiod pathway remains elusive. In this study, we reveal that the macromolecular condensation of CO, a central regulator in the photoperiod pathway, controls plant flowering in a condensate biophysical property-dependent manner. CO proteins assemble into multicomponent functional percolation clusters in the nucleus via B-box motifs when these proteins accumulate in response to long-day conditions. Coassembly with NF-YC9, NF-YB2, and *FT* is essential for condensing CO but maintains fluidic assemblies that are able to activate *FT* transcription. Disorganized component partitioning of the CO complex impairs the maintenance of the balance of inter- and intramolecular interactions, and the functional percolation clusters thus transition into malfunctional phase separation assemblies (Mittag and Pappu, 2022; Pappu et al, 2023). Here, we propose a working model in which the precisely regulated assembly of CO/NF-YB/NF-YC and *FT* into CO condensates with proper dynamic properties is critical to control the floral transition in the photoperiod pathway (Fig. 5). To be noted, we applied 35S-promotor driven CO expression for a direct visualization of CO in vivo given the fact that CO is specifically expressed in phloem companion cells (An et al, 2004) and its low abundance which presents challenging to be visualized at native expression. However, 35S-promotor driven CO is functional enough to rescue the flowering phenotype in *co* and we have shown that a robust regulation of CO assembly by NF-YB and NF-YC at different CO abundance level spanning form nanomolar- to micromolar-scale. At a low concentration, the CO percolation cluster in vivo might not be well resolved due to the optical diffraction limitation at endogenous expression. A precise characterization of the stoichiometry in endogenous CO percolation clusters, though more advanced image technology such as recently developed SlimVar single molecule tracking microscope in plants to investigate the functional condensates of other flowering controlling proteins (Jang et al, 2024; Payne-Dwyer et al, 2024), would help to better understand at which stoichiometry of CO multi-component condensation it will be the best CO assembly to activate *FT* transcription, which could be our future direction following this study. Nevertheless, it does not change our major conclusion that CO by itself tends to be over-assembled to be non-functional. It needs the NF-Y components that synergistically

optimize the CO assembly to a proper state and provide the conformation flexibility to activate the *FT* transcription.

Increasing evidence is elucidating the regulation of biochemical activities via the formation of membraneless biomolecular condensates (BMCs). The optimized fluidity of regulator-containing BMCs is an important biophysical feature for maintaining proper activities during signal transduction via phase separation (Banani et al, 2017; Sun et al, 2021). For instance, *Arabidopsis* SEUSS (SEU) can rapidly concentrate into liquid nuclear condensates in response to the hyperosmotic stress response (Wang et al, 2022). Similar to SEU, WNK1 kinase acts as a molecular crowding sensor and responds to hyperosmotic stress by forming functional condensates in animals (Boyd-Shiwarski et al, 2022). Here, our findings reveal a light-dependent local condensation of CO proteins. Through molecular simulation, we demonstrated that CO complexes (i.e., trimers, tetramers, and pentamers) with NF-Y exhibit progressively higher binding affinity for the *FT* promoter compared to the monomeric CO-NF-Y complex. This aligns with previous findings that the tetrameric CO-CCT-NY-F complex has stronger binding to COREs than its monomeric counterpart and can partially mitigate the late flowering phenotype caused by diminished CO oligomerization (Lv et al, 2021; Milo and Phillips, 2015; Zeng et al, 2022). These results underscore that locally condensed CO and NF-Y complexes more effectively associate with the *FT* promoter, enhancing *FT* transcription activation. However, the detailed mechanism of how CO condensation increases its binding affinity with *FT* promoter and enhance *FT* transcription would need more detailed biochemical and structural studies, but not only from the simulation, to be answered in the future. One may speculate the multiple CO binding sites on *FT* promotor would enable a cooperative binding to promote CO condensation. Such speculation can not be excluded but may barely happened in physiological condition due to the low copy number of *FT* in genome.

When expressing in protoplast, CO proteins exist in three states in vivo: a non-functional diffuse population, functional spherical condensation depending on the material properties, and malfunctional irreversible aggregate. The assembly of CO condensates involves NF-YB/NF-YC, which fine tune CO activities by mediating the transition of the diffuse CO into functional CO condensates and

preventing CO overassembly into a slow-diffusive malfunctional state (Fig. 5). Fluidic functional CO/NF-YB2/NF-YC9/*FT* condensates are modulated by their inter- and intramolecular interactions. Therefore, the network of interaction patterns of each component would need detailed characterization of functional or non-functional domain-based interaction and regulation, and the conformational changes within CO condensates require future biophysical and structural characterization.

Extensive studies have revealed that the NF-YC protein, also known as histone-associated protein 5 (HAP5), interacts with NF-YB (HAP3) to form a heterodimer analogous to the histone 2A/2B dimer with a highly conserved histone fold domain (Huber et al, 2012; Nardini et al, 2013), implying their universal functions in organisms. The NF-YB/NF-YC dimer forms a complex with NF-YA (HAP2) to generate the NF-Y heterotrimer complex, which is a canonical transcription factor that binds to the *CCAAT* box. This sequence element is among the most widely distributed in eukaryotic promoters, and the NF-Y heterotrimer is responsible for regulating the expression of a large number of genes in eukaryotic cells (Dolfini et al, 2009; Laloum et al, 2013; Mantovani, 1999). Despite the essential role of the NF-Y complex in numerous cellular processes, the mechanisms by which its subunits modulate transcriptional regulation are not fully understood. A recent study demonstrated that H2B.8, a histone variant, can compact unexpressed chromatin globally via phase separation in *Arabidopsis* sperm nuclei (Buttress et al, 2022), shedding some light on the mechanism whereby the NF-Y complex may regulate transcriptional processes. While previous studies have mainly focused on the phase separation of a single component, our study shows that three transcription factors (CO, NF-YB, and NF-YC) together with *FT* modulate the macromolecular assembly status to fine tune CO function. Particularly, our SPR and in vitro reconstitution assays at nanomolar scale indicate NF-YC and NF-YB cooperatively regulate a proper assembly of the CO condensate to maintain its functionality in a balanced manner, supporting the involvement of NF-Y in the phase separation-mediated transcriptional regulation of floral transition.

Interactive motif-containing IDRs contribute to phase separation through weak multivalent interactions (Huang et al, 2021; Lin et al, 2015; Zhang et al, 2022). Glutamine, aromatic residues, and arginine are present in many phase-separating proteins at a high frequency (Banani et al, 2017). The N-terminal IDR of NF-YC9 also contains a polyQ repeat, which is embedded in one of two hypothetical α-helices (Fig. EV5B), which may suggest dynamic conformational changes between folding and unfolding depending on the surrounding biochemical environment (Boersma et al, 2015; Liu et al, 2017). The deletion of IDRs or mutation of polyQ sequences in NF-YC9 hinders the proper transition of multi-component CO condensates into the functional fluidic assembly, thereby controlling CO conformation and motility dynamics. PolyQ sequences with comparable repeat lengths is conserved in NF-YC9 homologues across plant species, indicating the potential importance of these sequences in evolution (Appendix Fig. S4). PolyQ repeat-dependent processes have implications for physiology and disease. For example, the length of the polyQ repeat in ELF3 is correlated with thermal sensing in *Arabidopsis* via phase separation (Jung et al, 2020), and the abnormally long polyQ tract in the exon1 protein sequence facilitates pathological huntingtin aggregation in humans (Peskett et al, 2018). Surprisingly, in current case, both

longer and shorter polyQ in NF-YC9 led to similar CO condensation properties. Thus, evaluating how different length of polyQ in NF-YC9 affect the synergistic effect of NF-Y on CO condensation and whether other NF-Y subunits with IDRs or a certain length of polyQ repeats in plants modulate phase separation-dependent gene expression and developmental processes are of great interest. Additionally, it is worth investigating whether NF-YCs in other species, such as mammals, which lack the polyQ repeat feature, play roles in transcriptional regulation involving phase separation.

# Methods

**Reagents and tools table**

| Reagent/Resource | Reference or Source | Identifier or Catalog Number |
|---|---|---|
| **Experimental Models** | | |
| Arabidopsis: *nf-yc3 nf-yc4 nf-yc9* (*ycT*) | Tang et al, 2017 | |
| Arabidopsis: *35S:mCherry-CO* | This study | |
| Arabidopsis: *35S:mCherry-CO ycT* | This study | |
| Arabidopsis: *35S:NF-YC9-GFP ycT* | This study | |
| Arabidopsis: *35S:mCherry-CO 35S:NF-YC9-GFP ycT* | This study | |
| Arabidopsis: *35S:NF-YC9-ΔIDR1-GFP ycT* | This study | |
| Arabidopsis: *35S:NF-YC9-ΔIDR2-GFP ycT* | This study | |
| Arabidopsis: *35S:NF-YC9-ΔIDR1&2-GFP ycT* | This study | |
| Arabidopsis: *35S:NF-YC9-0Q-GFP ycT* | This study | |
| Arabidopsis: *35S:NF-YC9-37Q-GFP ycT* | This study | |
| Escherichia coli: Rosetta pLysS | AlpalifeBio | Cat#KTSM112L |
| Agrobacterium: GV3101 | Weidi | Cat#AC1001 |
| **Recombinant DNA** | | |
| pGreen-35S-GFP-CO | This study | *GFP-CO* was inserted into the pGreenII 0000 vector (Addgene Cat#44465) |
| pGreen-35S-GFP-CO-ΔB-box | This study | *GFP-CO-ΔB-box* was inserted into the pGreenII 0000 vector (Addgene Cat#44465) |
| pGreen-35S-NF-YC9-mCherry | This study | *35S-NF-YC9-mCherry* was inserted into the pGreenII 0000 vector (Addgene Cat#44465) |
| pGreen-35S-NF-YC9-ΔIDR1-mCherry | This study | *35S-NF-YC9-ΔIDR1-mCherry* was inserted into the pGreenII 0000 vector (Addgene Cat#44465) |
| pGreen-35S-NF-YC9-ΔIDR2-mCherry | This study | *35S-NF-YC9-ΔIDR2-mCherry* was inserted into the pGreenII 0000 vector (Addgene Cat#44465) |

| Reagent/Resource | Reference or Source | Identifier or Catalog Number |
|---|---|---|
| pGreen-35S-NF-YC9-ΔIDR1&2-mCherry | This study | 35S-NF-YC9-ΔIDR1&2-mCherry was inserted into the pGreenII 0000 vector (Addgene Cat#44465) |
| pGreen-35S-NF-YC9-0Q-mCherry | This study | 35S-NF-YC9-0Q-mCherry was inserted into the pGreenII 0000 vector (Addgene Cat#44465) |
| pGreen-35S-NF-YC9-37Q-mCherry | This study | 35S-NF-YC9-37Q-mCherry was inserted into the pGreenII 0000 vector (Addgene Cat#44465) |
| pGreen-35S-NF-YB2-BFP | This study | 35S-NF-YB2-BFP was inserted into the pGreenII 0000 vector (Addgene Cat#44465) |
| SUMO-mCherry-CO | This study | mCherry-CO was inserted into the pSUMO-LIC vector (Lab stock) |
| MBP-NF-YC9-GFP | This study | NF-YC9-GFP was inserted into the MBP containing purification construct (Lab stock) |
| His-NF-YB2 | This study | NF-YB2 was inserted into the pQE-30 vector (Qiagen) |
| **Antibodies** | | |
| Mouse anti-GFP | TRANS | Cat#HT801-02 |
| **Oligonucleotides and other sequence-based reagents** | | |
| PCR primers | This study | Appendix Table S2 |
| **Chemicals, Enzymes and other reagents** | | |
| BamHI | New England Biolabs | Cat#R3136 |
| Bcul (SpeI) | New England Biolabs | Cat#R3133 |
| XbaI | New England Biolabs | Cat#R0145 |
| HindIII | New England Biolabs | Cat#R3104 |
| EcoRI | New England Biolabs | Cat#R3101 |
| Cellulose R10 | Yakult | Cat#LOT240221-02 |
| Macerozyme | Yakult | Cat#LOT220510-02 |
| Total RNA Extraction Kit | Promega | Cat#LS1040 |
| ChamQ Universal SYBR qPCR Master Mix | Vazyme | Cat#Q711-03 |
| Rifampicin | Coolaber | Cat#CR9551 |
| Gentamicin | YUANYE | Cat#S17024 |
| Tetracycline | YUANYE | Cat#S17051 |
| Ampicillin | Solarbio | Cat#A7490 |
| Kanamycin | Coolaber | Cat#CK6731 |
| Chloramphenicol | Coolaber | Cat#CC3451 |
| Murashige and Skoog Medium | Solarbio | Cat#M8521 |

| Reagent/Resource | Reference or Source | Identifier or Catalog Number |
|---|---|---|
| IPTG | Sigma-Aldrich | Cat#I6758-5G |
| Pierce™ Protease Inhibitor Tablets, EDTA-free | Thermo Fisher | Cat#A32965 |
| Imidazole | MP Biomedicals | Cat#SKU 0210203301 |
| HEPES | Sigma-Aldrich | Cat#H3375 |
| Sucrose | Bio Basic | Cat#SB0498 |
| Yeast extract | Bio Basic | Cat#G0961 |
| Tryptone | Bio Basic | Cat#TG217 |
| Tris Base | Sigma-Aldrich | Cat#252859 |
| HisTrap™ HP column | Cytiva | Cat#17524802 |
| HiLoad 16/600 Superdex 200 pg column | Cytiva | Cat#28989335 |
| Sodium chloride | Merck | Cat#S9888 |
| Magnesium chloride hexahydrate | Bio Basic | Cat#M0250 |
| **Software** | | |
| Fiji | NIH | https://imagej.net/ |
| MetaMorph | Molecular Devices | https://www.moleculardevices.com/products/cellular-imaging-systems/high-content-analysis/metamorph-microscopy |
| Prism 6.0 | GraphPad | https://www.graphpad.com/scientific-software/prism/ |
| Origin 2022 | OriginLab | https://www.originlab.com/ |
| Biacore T200 | GE Healthcare | |
| Alphafold2 | DeepMind | https://colab.research.google.com/github/sokrypton/ColabFold/blob/main/AlphaFold2.ipynb |
| **Other** | | |
| LEICA SP8 STED 3X | Leica | |
| LightCycler 480 | Roche | |
| Nikon ECLIPSE Ti2-E inverted microscope | Nikon Instruments | |
| CM5 chip | Cytiva | |

## Plant materials and growth conditions

*Arabidopsis* plants used in this study were in the *A. thaliana* Col-0 background. The *nf-yc3 nf-yc4 nf-yc9* triple mutant (*ycT*) used in this study were obtained in our laboratory (Tang et al, 2017). All plant materials used in this study are listed in Appendix Table S1. The *35S:mCherry-CO 35S:NF-YC9-GFP ycT* was generated by crossing *35S:mCherry-CO ycT* with *35S:NF-YC9-GFP ycT*. The seeds were incubated at 4 °C for 3 d in the dark and then grown in

soil at 22 °C under long photoperiod conditions (16 h of light at 100 µmol m$^{-2}$ s$^{-1}$ and 8 h of dark). For microscopy imaging and gene expression analysis, the seeds were surface sterilized and sown on standard half-strength Murashige and Skoog agar (MS-agar) and kept at 4 °C in the dark for 3 d before being transferred to long photoperiod conditions.

## Plasmid construction and plant transformation

To generate the constructs for *35S:mCherry-CO*, *35S:NF-YC9-GFP*, and *35S:NF-YC9-variants-GFP* overexpression transgenic lines, the coding regions of CO, NF-YC9, and NF-YC9-variants were cloned into the *pGreen-35S-mCherry* and *pGreen-35S-GFP* vectors, respectively. By using *Agrobacterium*-mediated transformation, the *35S:mCherry-CO* construct was introduced into *Arabidopsis* Col-0 background and the *ycT* background, respectively. The *35S:NF-YC9-GFP* and *35S:NF-YC9-variants-GFP* constructs were inserted into the *ycT* background, respectively. These transgenic plants were selected with Basta.

To generate the constructs for in vivo protein distribution, the coding regions of CO, NF-YB2, NF-YC9, CO-ΔB-box, and NF-YC9-variants were cloned into the *pGreen-35S-GFP*, *pGreen-35S-BFP*, and *pGreen-35S-mCherry* vectors, respectively. The primers used for plasmid construction are listed in Appendix Table S2.

## Fluorescence microscopy

To image the mCherry-CO and NF-YC9-GFP clustering in nuclei of root cells, roots of 5-day-old transgenic seedlings with the indicated treatments were observed under Nikon ECLIPSE Ti2-E inverted microscope (Nikon Instruments) equipped with a CSU-W1 confocal spinning unit (Yokogawa) and a 100× 1.49NA Plan-Apo objective lens (Nikon Instruments). The mCherry and GFP signals were illuminated at 561 nm (Coherent) and 488 nm (Vortran) excitation, and the emissions were collected at 589–625 nm and 505–545 nm, respectively, using an ORCA-Fusion sCMOS camera (Hamamatsu). Z-stack images were carried out to record the signals in nuclei of root cells with a step size as 0.4 µm. Image acquisition was controlled by MetaMorph (Molecular Devices) software. For FRAP assay, photobleaching of the mCherry-CO signal was carried out using the same excitation laser and the signal recovery was recorded by time-series image with a time interval as 1 s till the fluorescence emission reached a plateau. The fluorescence intensity of the bleached region in each frame of the time-series images was determined in Image J. To subtract the overall signal bleaching during time-series image acquisition, the intensity of an ROI with the same size in unbleached region was measured to normalize the signal bleaching. All the image parameters for the same transgenic lines were kept the same in this study. Single particle image of mcherry-CO, NF-YC9-GFP, and NF-YB2 was performed in the same microscope mentioned above. 10 nM of recombinant mCherry-CO and the indicated concentrations of NF-YC9-GFP and NF-YB2 (labeled Alexa fluo 647) was mixed in buffer of 20 mM HEPES, 150 mM NaCl, pH 7.4 for 5 min before being added on cover glass and then checked under TIRFM, for which the illumination was controlled by iLas2 optical system. Single particle intensity quantification of mCherry-CO was conducted in Image J, the single particle identification and selection was accomplished through the TrakeMate plugin.

To image subcellular distribution of CO, NF-YB2, NF-YC9, CO-ΔB-box, and NF-YC9-variants in protoplasts, fluorescence images were generated using a confocal laserscanning microscope (Leica TCS SP5). 380 nm laser, 488 nm laser, and 587 nm laser were used to detect BFP, GFP, and mCherry excitation, respectively. For FRAP assay, photobleaching of the GFP-CO signal was carried out using the same excitation laser and the signal recovery was recorded by time-series image with a time interval as 5 s till the fluorescence emission reached a plateau. The fluorescence intensity of the bleached region in each frame of the time-series images was determined in Image J.

## Fluorescent-tagged protein clustering analysis

Spatial clustering index (SCI) of mCherry-CO was quantified to analyze the assembly of mCherry-CO condensation in nuclei of root cells. Z-stack images were first projected by collecting maximum signal intensity and then subjected to region of interest (ROI) selection by drawing a line width of 3 µm in each nucleus. The SCI was then analyzed by measuring the ratio of the average intensity of the 5% highest intensity to the 5% lowest intensity in each ROI (Ma et al, 2022). At least 35 ROIs from more than 10 images were quantified for each condition.

GFP-CO signal variances were analyzed to classify the assembly of GFP-CO condensates in nuclei of protoplasts. The GFP signal in nucleus was selected in Image J through the particle selection function after applying threshold and making binary image. GFP-CO signal variances were then calculated as (standard deviation of the intensity/average intensity)$^2$. About 60 nuclei of protoplasts from three repeats were analyzed for each condition. The distribution of GFP-CO signal variances was further plotted and subjected to multiple peak fitting in Origin (OriginLab) to classify the assembly of GFP-CO condensates into three populations. The distribution of each GFP-CO populations in different genetic backgrounds were analyzed according to the above-mentioned GFP-CO signal variance classification.

## Plasmid construction, protein expression, and purification in vitro

To generate the constructs for in vitro protein expression, the *mCherry-CO* was amplified by PCR and inserted into the pSUMO-LIC vector with 6x histidine plus SUMO tag at N-terminal. The *NF-YC9-GFP* was amplified by PCR and inserted into the MBP vector with 6xhistindine-MBP at N-terminal. The *NF-YB2* was amplified by PCR and inserted into the pQE-30 backbone as a N-terminal 6x histindine tagged protein. All constructs were generated through homologous recombination using clonExpress One-Step Cloning kit (vazyme). The primers used for plasmid construction are listed in Appendix Table S2.

To obtain prokaryotic expressed proteins, plasmids including *SUMO-mCherry-CO*, *MBP-NF-YC9-GFP*, and *His-NF-YB2* were transformed into *Escherichia coli* Rosetta pLysS competent cells, respectively. A single colony was picked from transformation plate and grown overnight in 10 mL LB at 37 °C with shaking (220 r.p.m.). The overnight culture was scaled to 1 L TB culture medium (2.4% yeast extract, 2% tryptone, 0.4% glycerol, 0.017 M KH$_2$PO$_4$ and 0.072 M K$_2$HPO$_4$) with kanamycin (50 µg/ml) and chloramphenicol (35 µg/ml) and grown at 37 °C with shaking until OD$_{600}$ reached 1.2, after which the temperature was lowered to 18 °C and protein expression was induced by adding isopropyl-β-D-

thiogalactoside (IPTG) to a final concentration of 0.5 mM. After overnight induction with shaking, cells were harvested by centrifugation for 15 min at $5000 \times g$ (rotor JA10, Beckman Coulter) at 4 °C and pellet were stored at $-80$ °C.

Cells were resuspended in binding buffer (50 mM Tris, 500 mM NaCl, 20 mM imidazole, pH 8.0) with 1 mM PMSF and protease inhibitor tablets (Thermo Fisher) and lysis was performed using an LM20 microfluidizer (20,000 psi), after which the lysate was centrifuged at $20,000 \times g$ and 4 °C for 1 h (JA 25.5 rotor, Beckman Coulter). Supernatants were clarified by a 0.45 μm syringe filter (Pall Corporation) and loaded on a Histrap column (Cytiva). Column was washed with binding buffer followed by gradient elution from 20 mM to 500 mM under same buffer condition. Peak fractions were checked by SDS–PAGE, after which good fractions were combined and dialyzed in protein buffer (50 mM Tris, 500 mM NaCl, pH 8.0) overnight at 4 °C. When necessary, the dialyzed protein solution was further separated by size exclusion chromatography using a HiLoad 16/600 Superdex 200 pg column (Cytiva) pre-equilibrated with protein buffer. The final protein solution was concentrated using centrifugal concentrators (Merck Millipore) with a 10 kDa or 30 kDa cutoff. Protein concentration was determined by running SDS-PAGE gels with a series amount of BSA standards.

## Fluorescence recovery after photobleaching for proteins in vitro

Single- or multicomponent of assemblies were incubated in FRAP buffer (20 mM HEPES, 150 mM NaCl, pH 7.4) for 5 min before image acquisition. Bleach laser power is 100% with $15 \times 15$ bit bleach region. Time-lapse images of fluorescence images were recorded at 30 s interval within 10 min after bleach.

## Surface plasmon resonance

SPR was performed based on our previous work (Sun et al, 2021). Briefly, assays were carried out at 25 °C on a Biacore T200 (GE Healthcare) using HBST running buffer (10 mM HEPES, 150 mM NaCl, 3 mM EDTA, 0.05% Tween 20, pH 7.4). CM5 chip (Cytiva) was activated with 0.2 M 1-ethyl-3-(3-dimethylpropyl)-carbodiimide and 0.1 M N-hydroxysuccinimide. Ligand proteins of CO, NF-YC9, and mixture of CO and NF-YC9 diluted in 10 mM sodium acetate pH 4.0 were then immobilized by amine coupling to 1000–2000 RU on flow cells 2, 3, and 4, respectively. The CO/NF-YC9 mixture consisted of 1 part NF-YC9 to 4 parts CO as NF-YC9 preconcentrated on sensor surface more efficiently. Analyte proteins (NF-YB2, CO, and NF-YC9) were then injected for 60 s before a dissociation phase of 300 s, at a flow rate of 30 μL/min. For regeneration, two 30 s pulses of 3 M MgCl₂ injected at 30 μL/min were carried out at the end of every cycle. All runs were double-referenced. Biacore T200 Evaluation software 3.0 was used for binding kinetics analysis with the bivalent analyte model.

## Molecular simulation

The molecular model was built with the SMOG2 protein and 3SPN2 DNA models (Hinckley et al, 2013; Noel et al, 2016), as implemented in the OpenABC package (Liu et al, 2023). Prediction of the structure of full length of CO was conducted on Google Colab Alphafold2 server thought the function of multimer

prediction (Evans et al, 2022; Jumper et al, 2021). CO-NF-Y complex was constructed by connecting AlphaFold2-predicted CO and experimentally resolved CO-CCT-NF-Y structure (PDB: 7CVO). All protein parameters were following the default values from the SMOG2 model, except that the native pair interaction strength was set to 6.0 kJ/mol to keep the structured domains stable. Double helical DNA containing COREs (sequence 5′-TATTTCCAGTGTATTAGTGTGGTGGGTTTGGAATACCAC AAACAGAAATAAAAAGAAAGAAAAATATGAAATAAGACG ACAATGTGTGATGTACGTAGAATCAGTTTTAGATTCTAGT ACATCAATAGACAAGAAAAAGATTGTGGTTATGATTTCAC CGACCCGAGTT-3′) was modeled with 3SPN2 model with default parameters.

All simulations were conducted within cubic boxes with 120 nm on all three dimensions. Fifteen DNA double helices and sixty CO-NF-Y molecules were used in each simulation. The oligomer form of CO was defined and maintained by the native interaction in the SMOG2 model based on AlphaFold2-predicted structures. In all oligomer CO-DNA simulations, we maintained sixty CO-NF-Y molecules (corresponding to monomeric CO-NF-Y complex), though the number of oligomers was not constant. All simulations were started from randomly placed molecules, and performed under the NVT ensemble with constant temperature at 300 K. The timestep was set to 10 fs. Time integration was evolved according to Langevin dynamics with the friction coefficient being 0.01 ps$^{-1}$. All simulations were performed for 1 microsecond, with the first halves treated as equilibration. The binding statistics were performed by counting the number of CCT-NF-Y complex (corresponding to monomeric CO-NF-Y complex) contacting the DNA/COREs. A molecular contact was counted whenever the particles of the two groups under examination were within 1.2 nm.

## RNA extraction and qRT-PCR analysis

For the *FT* gene expression analysis in plants, 5-day-old seedlings with 14 h of light were collected and immediately placed in liquid nitrogen, then stored at $-80$ °C. Total RNA was extracted using Eastep Super Total RNA Extraction Kit (Promega, Madison, WI, USA) according to the manufacturer's instruction. qRT-PCR was performed on LightCycler 480 thermal cycler system (Roche, Basel, Switzerland) with KAPA SYBR Fast qPCR Kit Master Mix (KAPA BIO, Wilmington, MA, USA). The relative gene expression was quantified using the comparative Ct method ($2^{-\Delta\Delta Ct}$), and *UBQ10* was used as an internal control. The primers used for gene expression analysis are listed in Appendix Table S2.

## Transient expression assay

To generate the *pFT:GUS* reporter construct, ~5.7 kb *FT* promoter was cloned into the *HY107-GUS* plasmid. The *35S:GFP-CO*, *35S:NF-YB2-BFP*, *35S:NF-YC9-mCherry*, and *35S:NF-YC9-variants-mCherry* constructs were used as effectors. The *Arabidopsis* mesophyll protoplast system was used to detect GUS expression. The data represent the averages of three biological replicates. The primers used for the transient expression assay are listed in Appendix Table S2.

## Yeast two-hybrid assay

The coding regions of *CO*, *NF-YC9*, and *NF-YC9-variants* were amplified and cloned into the pGBKT7 and pGADT7 (Clontech)

vectors, respectively. The primers used are listed in Appendix Table S2. Yeast two-hybrid assays were performed using the Yeastmaker Yeast Transformation System 2 (Clontech). Yeast AH109 cells were co-transformed with the specific bait and prey constructs. All yeast transformants were grown on SD/-Trp/-Leu or SD/-Trp/-Leu/-His/10 mM 3-AT medium for protein interaction test. The primers used for the yeast two-hybrid assay are listed in Appendix Table S2.

## Statistical analysis

All statistical analyses were performed in GraphPad Prism 6. *p* values were determined by two-tailed Student's t-test assuming equal variances and the one-way analysis of variance (*p* values were indicated in the figures or figure legends, ns = not significant). Error bars indicate the standard deviation (S.D.). Graphic illustrations were custom-drawn or adapted from icons on BioRender.com.

# Data availability

All data are available within the article, appendix data, and source data files. All materials used in this study are available by request from the corresponding author. Source data are provided with this paper.

The source data of this paper are collected in the following database record: biostudies:S-SCDT-10_1038-S44318-024-00293-0.

# Peer review information

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

## Acknowledgements

We thank the Institutional Center for Shared Technologies and Facilities of South China Botanical Garden, CAS for the microscopy imaging system and the

NTU Protein Production Platform (www.proteins.sg) Singapore for several initial protein expression and purification trials. This study was supported by grants from the Science and Technology Projects in Guangzhou (E3330900-01 to XH), the Key-Area Research and Development Program of Guangdong Province (2022B1111230001 to XH), and Ministry of Education (MOE2019-T3-1-012, MOE-T2EP30121-0015, MOE-T2EP30122-0021), IDMxS (EDUN C-33-18-279-V12), and National Research Foundation (MOH-000955, NRF-NRFI08-2022-0012) to YM in Singapore.

## Author contributions

**Xiang Huang**: Conceptualization; Data curation; Formal analysis; Validation; Methodology; Writing—original draft; Writing—review and editing. **Zhiming Ma**: Conceptualization; Resources; Data curation; Software; Formal analysis; Validation; Investigation; Visualization; Methodology; Writing—original draft; Writing—review and editing. **Danxia He**: Data curation; Formal analysis. **Xiao Han**: Data curation; Formal analysis; Investigation; Methodology. **Xu Liu**: Data curation; Formal analysis; Investigation; Methodology. **Qiong Dong**: Data curation; Formal analysis; Investigation; Methodology. **Cuirong Tan**: Data curation; Formal analysis; Investigation; Methodology. **Bin Yu**: Data curation; Formal analysis; Investigation; Methodology. **Tiedong Sun**: Data curation; Formal analysis; Investigation. **Lars Nordenskiöld**: Conceptualization; Supervision; Funding acquisition; Methodology. **Lanyuan Lu**: Formal analysis; Supervision; Funding acquisition; Investigation; Methodology. **Yansong Miao**: Conceptualization; Resources; Data curation; Formal analysis; Supervision; Funding acquisition; Validation; Investigation; Visualization; Methodology; Writing—original draft; Writing—review and editing. **Xingliang Hou**: Conceptualization; Data curation; Formal analysis; Supervision; Funding acquisition; Validation; Investigation; Visualization; Methodology; Writing—original draft.

Source data underlying figure panels in this paper may have individual authorship assigned. Where available, figure panel/source data authorship is listed in the following database record: biostudies:S-SCDT-10_1038-S44318-024-00293-0.

## Disclosure and competing interests statement

The authors declare no competing interests.

# Expanded View Figures

**Figure EV1.  CO proteins accumulate and assemble to condensates in response to light.**

(A) Time course of light-dependent CO condensate formation in 5-day-old transgenic *Arabidopsis* seedlings (*35S:mCherry-CO*, *35S:mCherry-CO ycT*, and *35S:mCherry-CO 35S:NF-YC9-GFP ycT*) with 15 h darkness treatment. Scale bars, 5 µm. (B) Quantification of mCherry-CO cluster index in (A). $n = 0, 0, 0, 42, 37, 43, 59, 48, 45$ ROIs (1 ROI in each nucleus) from left to right chart. Error bars, means ± SD. Significant differences among different groups were determined by one-way ANOVA test. (C) NF-YC9-GFP displayed a diffuse fluorescence signal pattern in 5-day-old *35S:NF-YC9-GFP ycT* root epidermal cell. Noted there is no CO expression here as endogenous CO is not expressed in root epidermal cells. Scale bars, 2 µm. (D) A framework showing the procedures of computer simulation to understand oligomerized CO-NF-Y complex binding on *FT* promotor fragments. Here, the structures of monomeric, trimeric, tetrameric and pentameric CO were first predicted by Alpha-Fold2. Subsequently, the resolved structure of CO-CCT-NF-Y complex (PDB: 7CVO) was applied to replace the CCT domains in the multimeric CO structure. The resulting multimeric CO-NF-Y complexes were then modeled together with the *FT* promoter DNA fragments containing the four binding motif: P1, P2, CORE1, CORE2, colored for highlight, in the simulation boxes with same size, in which the proteins and DNA were shown after the simulation reaching equilibrium. The number of CO-NF-Y molecule and DNA fragment were fixed at 60 and 15, respectively, though the number of CO-NF-Y oligomers were not constant. (E) The distribution of the number of bound CO-NF-Y on each CO binding motif, as shown in d, in the simulation box modeling monomeric CO-NF-Y complex and DNA. Source data are available online for this figure.

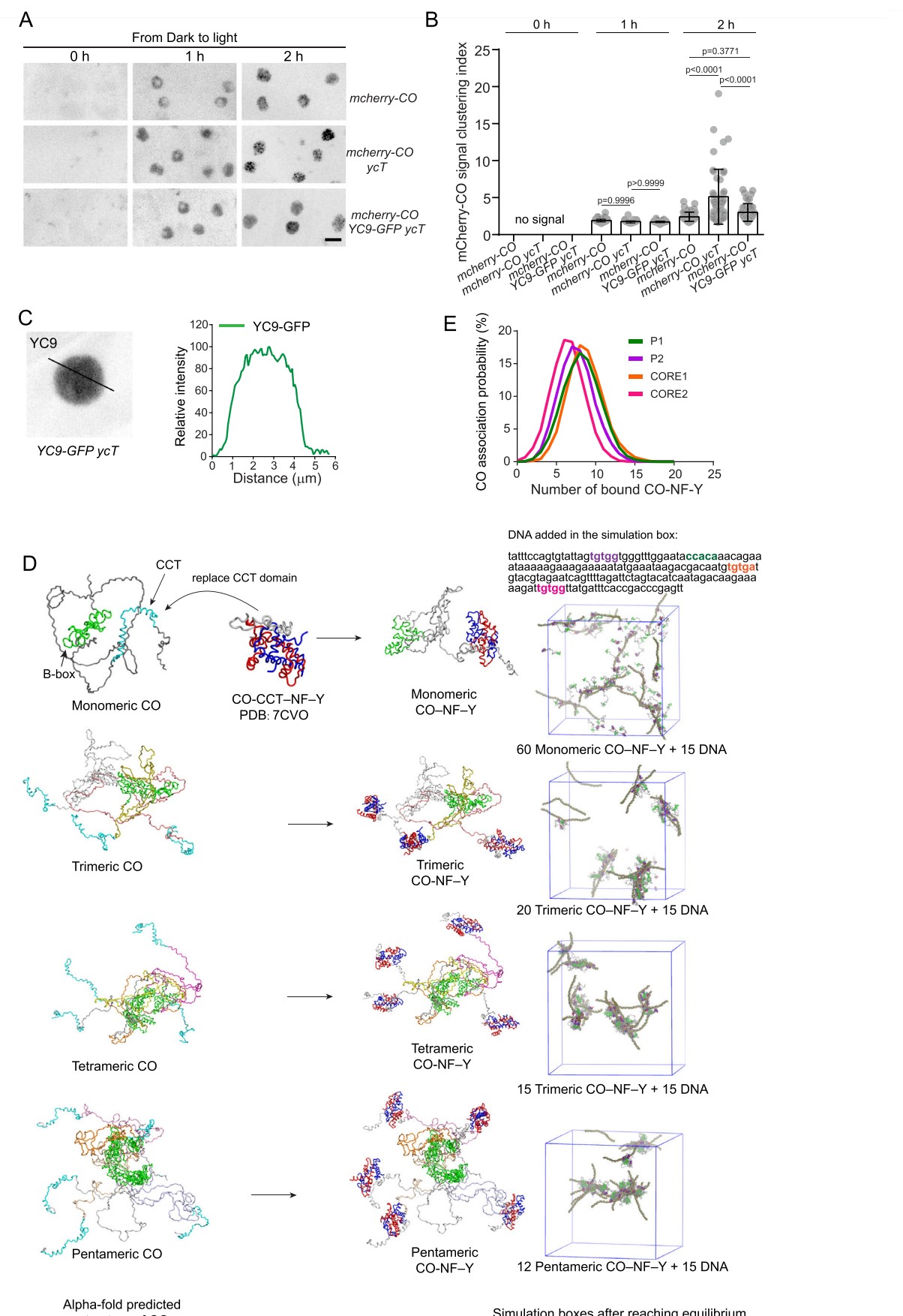

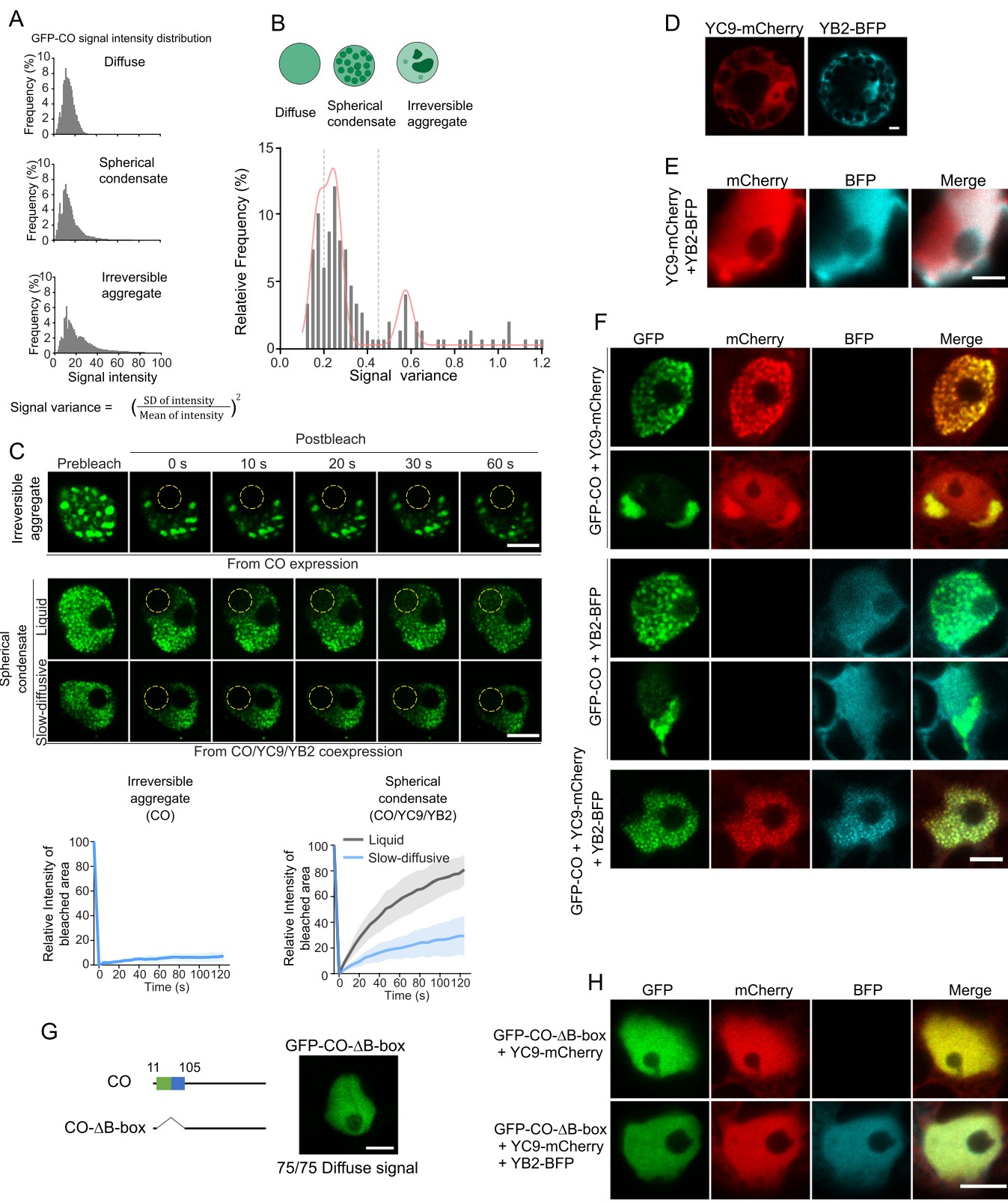

Signal variance = $\left(\dfrac{\text{SD of intensity}}{\text{Mean of intensity}}\right)^2$

◄  **Figure EV2.  Characterization of CO assembly.**

(A) The character of fluorescence signal intensity distribution of diffuse, spherical condensation, and irreversible aggregate of CO assemblies. The equation below shows the calculation of the signal variance in each individual nucleus. (B) Plotting of CO signal variance distribution after collecting the overall CO signal variance data from different combination of CO, NF-YC9, NF-YB2 co-expression. The variances of fluorescence signal intensities were peaked and thus were classified to three populations: diffuse CO (signal variance < 0.21), spherical CO condensation (signal variance 0.21–0.45), and irreversible CO aggregate signals (signal variance > 0.45). $n = 150$. (C) FRAP assay showing unrecoverable CO signal in the irreversible aggregate of CO assembly as in (B) and the liquid/slow-diffusive spherical condensates of CO on CO/YC9/YB2 coexpression, respectively. Time indicates the duration after the photobleaching pulse. Dash circles indicate the bleached areas. Recovery curves from the intensity quantification of the bleached area was plotted. $n = 5$, 11 of independent observations from left to right graph. Solid lines and shaded areas represent means ± SD. Scale bars, 5 μm. (D) Subcellular localization of NF-YC9-mCherry and NF-YB2-BFP in *Arabidopsis* protoplasts, respectively. Scale bars, 5 μm. (E) Co-expression of NF-YC9-mCherry and NF-YB2-BFP in nuclei of *Arabidopsis* protoplasts. Scale bars, 5 μm. (F) Co-expression of GFP-CO and NF-YC9-mCherry, GFP-CO and NF-YB2-BFP, and GFP-CO, NF-YC9-mCherry and NF-YB2-BFP in nuclei of *Arabidopsis* protoplasts. Scale bars, 5 μm. (G) Schematic of *CO* and *CO-ΔB-boxes* (deletion of two B-boxes, which are indicated by green and blue boxes) constructs. Subcellular localization of GFP-CO-ΔB-boxes in *Arabidopsis* protoplasts. Scale bars, 5 μm. (H) Co-expression of GFP-CO-ΔB-boxes and NF-YC9-mCherry, and GFP-CO-ΔB-boxes, NF-YC9-mCherry and NF-YB2-BFP in nuclei of *Arabidopsis* protoplasts. Scale bars, 5 μm. Source data are available online for this figure.

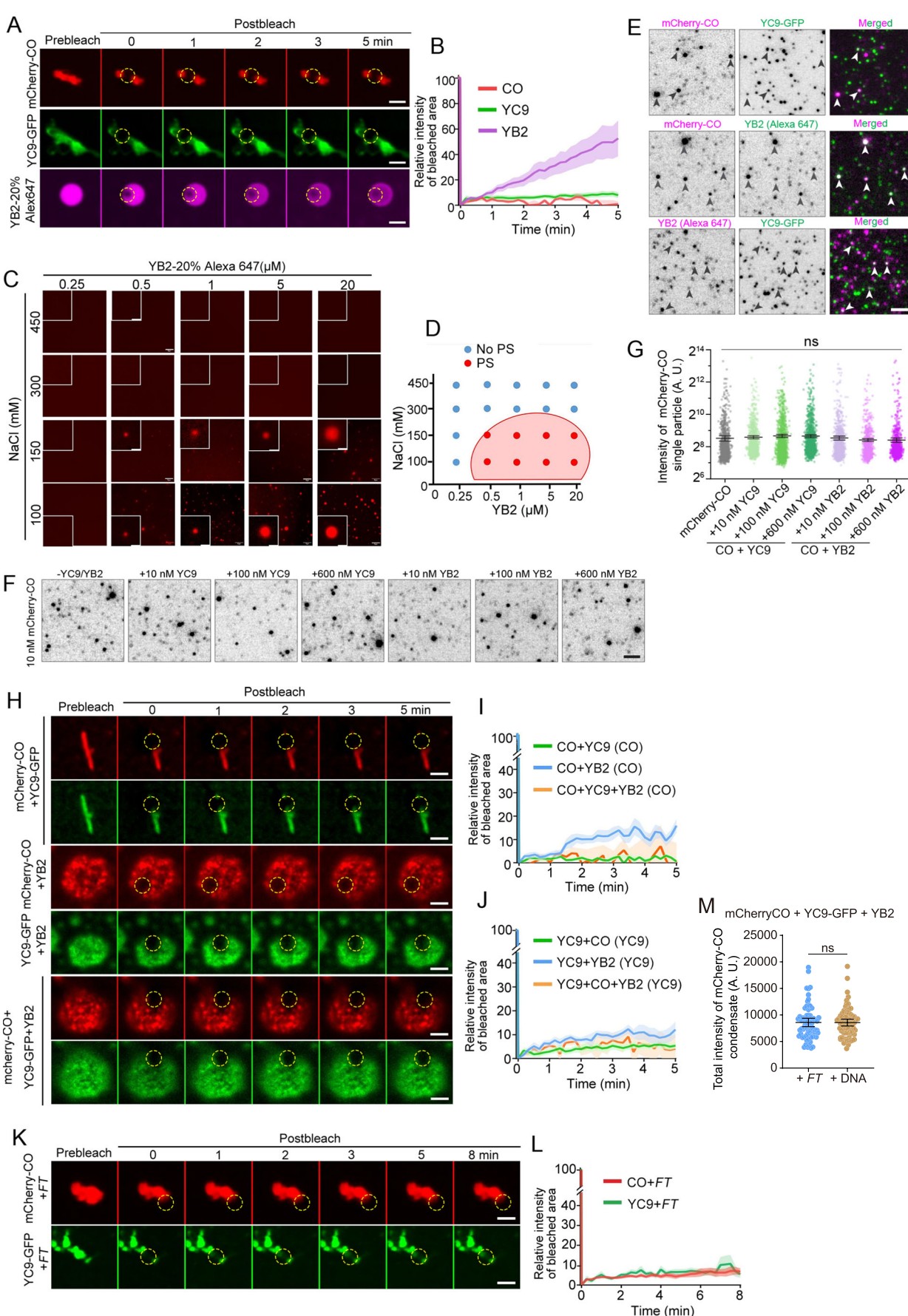

◀ **Figure EV3. The condensation properties of mCherry-CO, NF-YC9-GFP, and NF-YB2 assemblies in vitro at nano- or micro-molar scale.**

(A) FRAP of 0.25 μM mCherry-CO, 0.25 μM NF-YC9-GFP, and 15 μM NF-YB2 with Alex647. All the proteins, including the following experiments, were diluted in 20 mM HEPES, 150 mM NaCl, pH 7.4 buffer. Scale bar, 2 μm. (B) FRAP recovery plot of 0.25 μM of mCherry-CO ($n = 5$), 0.25 μM of NF-YC9-GFP ($n = 6$), and 15 μM of NF-YB2 with Alex647 ($n = 14$), respectively. (C, D) Phase diagram of NF-YB2. The size of NF-YB2 droplets is dependent on the concentration of protein (0.25 μM to 20 μM) and salt (100 mM to 450 mM). Scale bar, 5 μm. (E) Single particle image of recombinant mCherry-CO, NF-YC9-GFP, and NF-YB2 (labeled by Alexa fluor 647), indicating the intermolecular interaction of any two of those three components. 10 nM indicated two proteins were mixed in the buffer of 20 mM HEPES, 150 mM NaCl, pH 7.4 for 5 min before being added on cover glass and then checked under TIRFM. Merged image shows the colocalization of indicated two components. Arrow heads indicate the colocalized foci and the related protein single particles in each channel. Scale bar, 2 μm. (F) Single particle images of 10 nM recombinant mCherry-CO mixed with a series of concentrations of YC9-GFP or NF-YB2. The proteins were mixed in the buffer of 20 mM HEPES, 150 mM NaCl, pH 7.4 for 5 min before being added on cover glass and then checked under TIRFM. Scale bar, 2 μm. (G) Quantification of the mCherry-CO single particle total intensity in (E). $n > 700$ single particles in each chart. The middle lines indicate the mean values. Error bars represent 95% confidence intervals. No signal difference was detected over each group by one-way ANOVA test. (H) FRAP of mixed droplets of 0.25 μM of mCherry-CO, 0.25 μM of NF-YC9-GFP, and 15 μM of NF-YB2 in pairs or together. Scale bar, 2 μm. (I) FRAP recovery plot of mCherry-CO in the mixed droplets of 0.25 μM mCherry-CO/0.25 μM NF-YC9-GFP ($n = 6$), 0.25 μM mCherry-CO/15 μM NF-YB2 ($n = 11$), and 0.25 μM mCherry-CO/0.25 μM NF-YC9-GFP/15 μM NF-YB2 ($n = 14$), respectively. (J) FRAP recovery plot of NF-YC9-GFP in the mixed droplets of 0.25 μM mCherry-CO/0.25 μM NF-YC9-GFP ($n = 6$), 0.25 μM NF-YC9-GFP/15 μM NF-YB2 ($n = 10$), and 0.25 μM mCherry-CO/0.25 μM NF-YC9-GFP/15 μM NF-YB2 ($n = 14$), respectively. (K) FRAP of mixed droplets of 0.25 μM mCherry-CO/5 μg/mL *FT* and 0.25 μM NF-YC9-GFP/5 μg/mL *FT*, respectively. Scale bar, 2 μm. (L) FRAP recovery plot of mCherry-CO in the mixed droplets of 0.25 μM mCherry-CO/5 μg/mL *FT* ($n = 19$), and NF-YC9-GFP in the mixed droplets of 0.25 μM NF-YC9-GFP/5 μg/mL *FT* ($n = 16$), respectively. (M) Quantification of total intensity of mCherry-CO condensates in Fig. 3E. $n = 67, 73$ from left to right. The middle lines indicate the mean values. Error bars represent 95% confidence intervals. ns indicate no significant differences (Student's t test). Noted for all the FRAP assays in (A), (H), and (K), the time indicates the duration after the photobleaching pulse and the dash circles indicate the bleached areas. The solid lines and shaded areas in (B), (I), (J), and (L) represent means ± SD. Source data are available online for this figure.

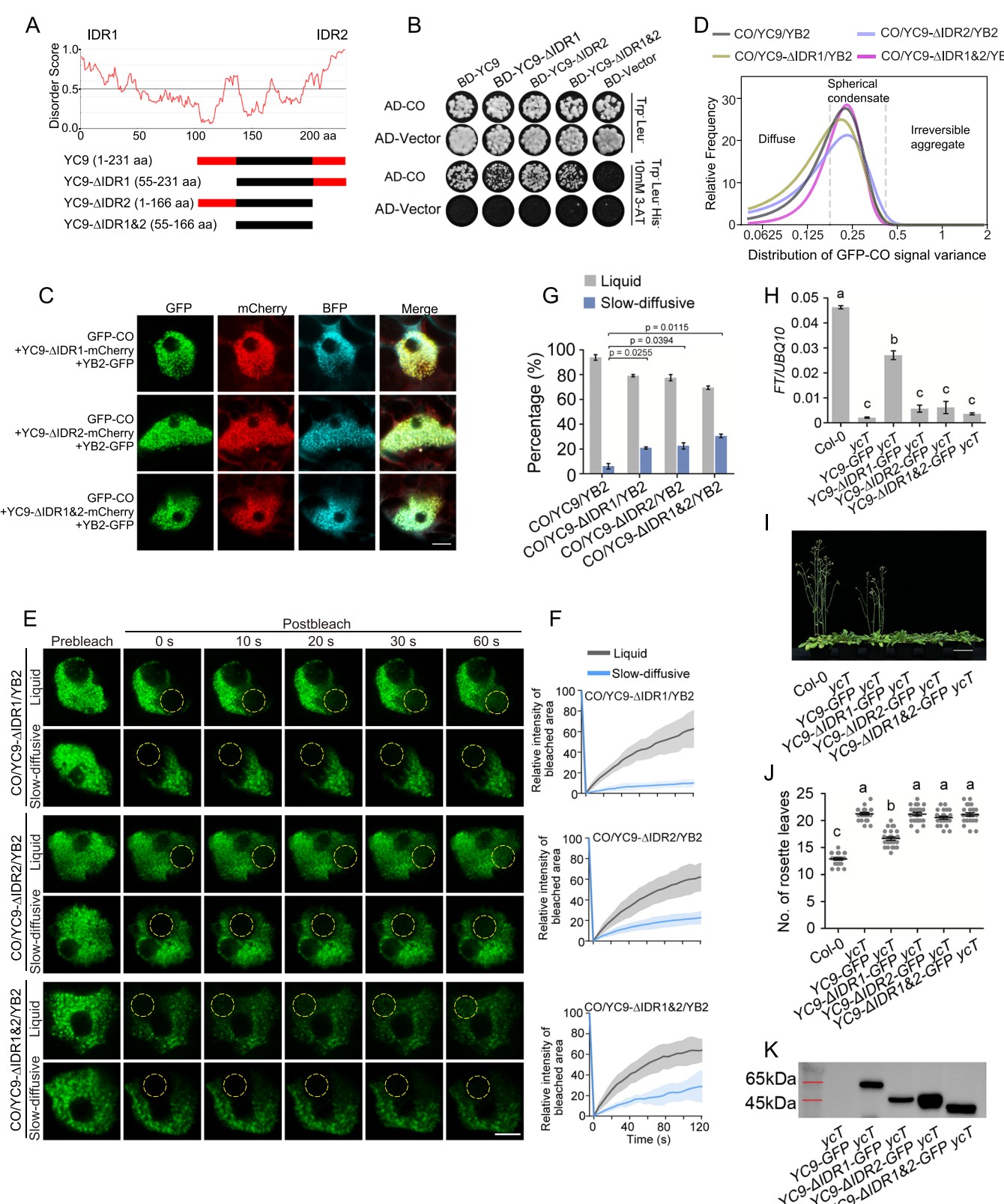

◀

**Figure EV4. IDRs within NF-YC9 are required for CO phase separation and its function in flowering control.**

(A) Sequence analysis of NF-YC9, containing two IDR domains. Schematic of *NF-YC9* and *NF-YC9-ΔIDR mutants* (deletion of IDR1, IDR2, or both) constructs. (B) Yeast two-hybrid assays show the interactions between CO and NF-YC9, and CO and NF-YC9-ΔIDR mutants. Transformed yeast cells were grown on SD/Trp⁻/Leu⁻ and SD/Trp⁻/Leu⁻/His⁻ (containing 10 mM 3-AT) medium. AD/BD-vector, vector-only controls; AD, activation domain; BD, DNA-binding domain. (C) Subcellular localization of GFP-CO, NF-YB2-BFP, and NF-YC9-ΔIDRs-mCherry in *Arabidopsis* protoplasts. Scale bars, 5 μm. (D) The distribution of GFP-CO signal variance under indicated combinations of transient expression, subjecting to Gaussian fitting. Dashes lines indicate the cut off on signal variance values to classify the CO assembly to diffuse, spherical condensation and irreversible aggregate. $n = 75$. (E) FRAP assays for the spherical GFP-CO condensates in the nucleus with CO/NF-YC9-ΔIDRs/NF-YB2 co-expression. Time indicates the duration after the photobleaching pulse. Dash circles indicate the bleached areas. Scale bar, 5 μm. (F) FRAP recovery curves from the intensity quantification of the bleached area in (E). Data from ≥7 and 4 of independent observations were plotted for liquid and slow-diffusive condensates, respectively. Solid lines and shaded areas represent means ± SD. (G) Quantification of the distribution ratios of CO spherical condensates showing either liquid or slow-diffusive material property in transient expression under combinations of GFP-CO, NF-YB2-BFP, and NF-YC9 (NF-YC9-ΔIDRs-mCherry) in *Arabidopsis* protoplasts. Error bars, means ± SD, $n = 50$ nuclei. Asterisks indicate significant differences (Student's t test, *$P < 0.05$). (H) qRT-PCR analysis of *FT* expression in 5-day-old seedlings of Col-0, *ycT*, *35S:NF-YC9-GFP ycT*, *35S:NF-YC9-ΔIDR1-GFP ycT*, *35S:NF-YC9-ΔIDR2-GFP ycT*, and *35S:NF-YC9-ΔIDR1&2-GFP ycT*. Gene expression levels were normalized to *UBQ10*, acting as an internal control. Error bars, means ± SD, $n = 3$. Different lowercase letters above the columns indicate the significant difference among different groups (one-way ANOVA, $P < 0.0001$). (I) Flowering phenotype of Col-0, *ycT*, *35S:NF-YC9-GFP ycT*, *35S:NF-YC9-ΔIDR1-GFP ycT*, *35S:NF-YC9-ΔIDR2-GFP ycT*, and *35S:NF-YC9-ΔIDR1&2-GFP ycT*. Scale bar, 5 cm. (J) Comparison of rosette leaf number for the representative transgenic plants in (I). $n = 25$ seedlings. Different lowercase letters above the columns indicate the significant difference among different groups (one-way ANOVA, $P < 0.0001$). (K) Western blot analysis of NF-YC9-GFP, NF-YC9-ΔIDR1-GFP, NF-YC9-ΔIDR2-GFP, and NF-YC9-ΔIDR1&2-GFP expression in the representative transgenic plants. Source data are available online for this figure.

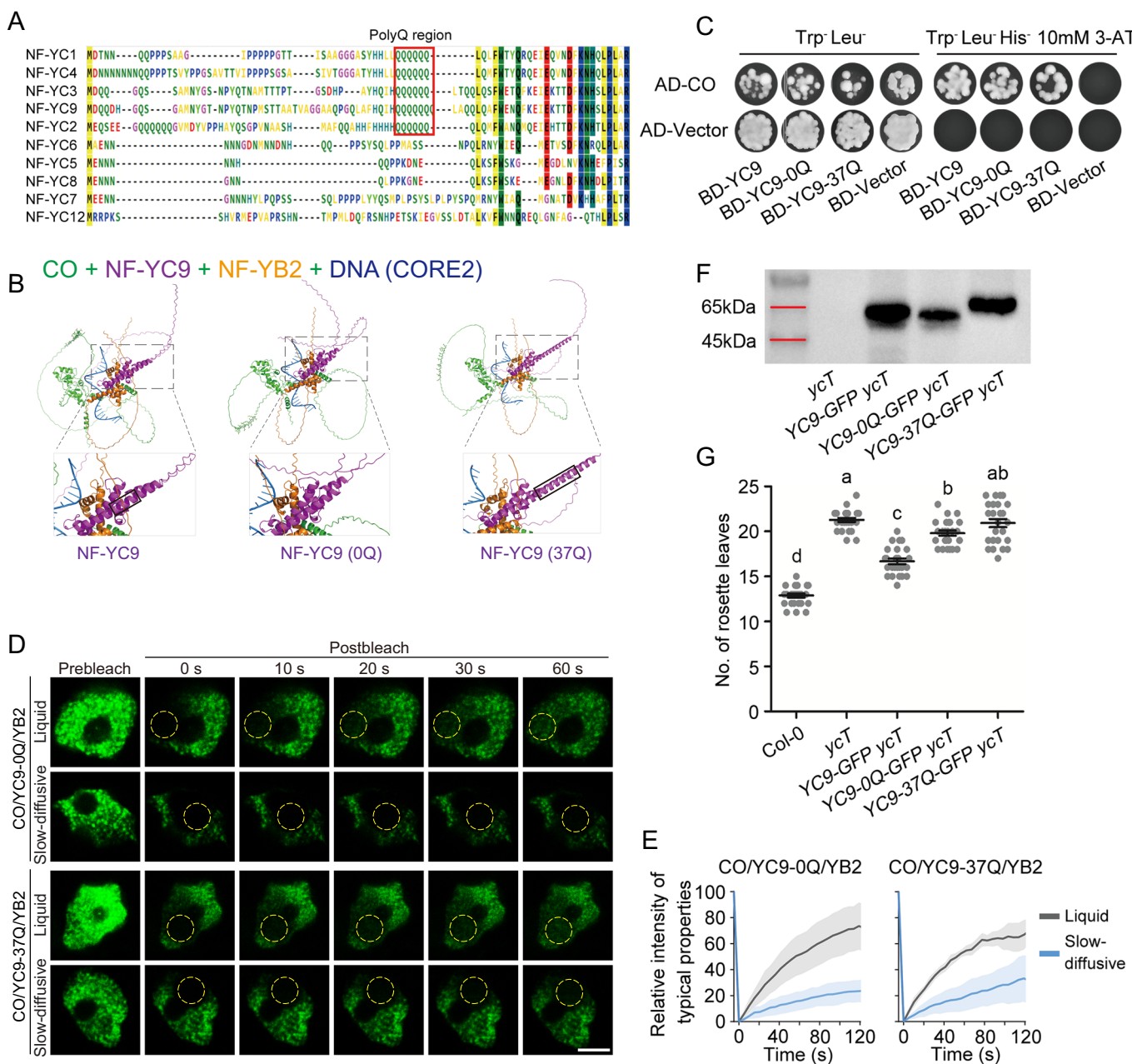

**Figure EV5.** The polyQ repeat embedded within NF-YC9 IDR is required for mediating CO function in controlling flowering.

(A) Sequence analysis of NF-YC family members in *Arabidopsis thaliana* revealed a conserved polyQ repeat among NF-YC1/4/3/9/2, which are related to flowering. (B) Structures of CO/NF-YC9/NF-YB2/DNA complex with original NF-YC9, NF-YC9 (0Q) or NF-YC9 (37Q) as predicted by AlphaFold3. The DNA fragment input contains the CORE2 site for CO/NF-Y binding. Zoomed regions show the polyQ region embedded α-helices, the polyQ repeats were highlighted with the black box. (C) Yeast two-hybrid assays show the interactions between CO and NF-YC9, and CO and NF-YC9-polyQ mutants. Transformed yeast cells were grown on SD/Trp-/Leu- and SD/Trp-/Leu-/His- (containing 10 mM 3-AT) medium. AD/BD-vector, vector-only controls; AD, activation domain; BD, DNA-binding domain. (D) FRAP assays for the spherical GFP-CO condensates in the nucleus with CO/YC9-0Q/YB2 or CO/YC9-37Q/YB2 co-expression. Time indicates the duration after the photobleaching pulse. Dash circles indicate the bleached areas. Scale bar, 5 μm. (E) FRAP recovery curves from the intensity quantification of the bleached area in (D). Data from ≥7 and 4 of independent observations were plotted for liquid and slow-diffusive condensates, respectively. Solid lines and shaded areas represent means ± SD. (F) Western blot analysis of NF-YC9-GFP, NF-YC9-0Q-GFP, and NF-YC9-37Q-GFP expression in the representative transgenic plants. (G) Comparison of rosette leaf number of Col-0, *ycT*, *35S:NF-YC9-GFP ycT*, *35S:NF-YC9-0Q-GFP ycT*, and *35S:NF-YC9-37Q-GFP ycT*. Error bars, means ± SD. $n = 25$ seedlings. Different lowercase letters above the columns indicate the significant difference among different groups (one-way ANOVA, $P < 0.0001$). Source data are available online for this figure.

