## [Peer Review File · The EMBO Journal]

Molecular condensation of the CO/NF-YB/NF-YC/FT complex gates floral transition in Arabidopsis

Xiang Huang, Zhiming Ma, Danxia He, Xiao Han, Xu Liu, Qiong Dong, Cuirong Tan, Bin Yu, Tiedong Sun, Lars Nordenskiöld, Lanyuan Lu, Yansong Miao, and Xingliang Hou

Corresponding author(s): Yansong Miao (yansongm@ntu.edu.sg) , Xingliang Hou (houxl@scib.ac.cn)

Review Timeline:

Submission Date:	26th Apr 23
Editorial Decision:	24th May 23
Appeal Received:	27th May 24
Editorial Decision:	17th Sep 24
Revision Received:	10th Oct 24
Accepted:	16th Oct 24

Editor: Cornelius Schneider

Transaction Report:

Dear Yansong,

Thank you for submitting your manuscript entitled 'Molecular condensation of the CO/NF-YB/NF-YC/FT complex gates plant floral transition' (EMBOJ-2023-114369) for consideration by the EMBO Journal. It has now been seen by three referees whose comments are shown below. I have now read and discussed your manuscript and these reports with my editorial colleagues; however, I am sorry to say that we have concluded that we cannot offer publication at the EMBO Journal.

While the referees agree that the manuscript explores important aspects of CONSTANS-mediated regulation of flowering and agree that your proposed hypothesis of regulation by phase separation is interesting, they are concerned that many of the key findings in this study need further investigation and validation. In particular, all referees state that the *in vivo* experiments are performed with highly overexpressed proteins and point to a lack of genetic evidence that supports the mechanisms of co-regulation which you propose. In addition, the referees agree that the performed *in vitro* experiments are insufficient to deduce physicochemical properties of the condensates.

Therefore, while we find the topic and scope of this manuscript very interesting, the requested list of further experiments is too substantial to be carried out within a timeframe which we can reasonably assign to a manuscript revision.

If you are able to add further *in vitro* experiments which clarify the biophysical properties of the condensates, and can substantiate your findings *in vivo* with experiments under conditions which are closer to physiological protein concentrations, I will enthusiastically consider the manuscript again. For such resubmissions, we take novelty over the original manuscript into consideration but given their constructive feedback will involve the same referee(s). If you are considering a resubmission of the paper once you have gained further mechanistic insight, please contact me in advance.

Thank you in any case for the opportunity to consider this manuscript. I am sorry we cannot be more positive on this occasion, but we hope nevertheless that you will find our referees' comments helpful.

Yours sincerely,

Cornelius Schneider, PhD
Editor
The EMBO Journal
c.schneider@embojournal.org

Referee #1:

This manuscript examines the molecular condensation of the flowering regulator CONSTANS (CO) and its regulation by two CO interactors, NF-YB and NF-YC. The authors observed the formation of two types of CO condensates: high-order, gel-like and low-order, fluidic. They discovered that the former type is predominantly mediated by CO homotypic interaction, while the latter type requires the participation of NF-YC and NF-YB. Using FT expression as a readout, they concluded that the fluidic CO-NF-YB/YC condensates are functional in promoting flowering transition, whereas the gel-like CO condensates are not. While I find the study intriguing in terms of CO-related cell biology and biophysical dynamics, the weak genetic evidence to biologically connect different CO condensate types with their function in flowering transition diminishes my enthusiasm. Moreover, I am concerned about whether the observed types of CO condensates and the proposed tunable system exist *in vivo*, given that the majority of experiments were performed using overexpression and transient expression systems. For the above reasons, I believe the story needs significant improvement before it can be published.

Major:

1. The manuscript presents intriguing data showing potential correlations between changes in CO condensates and flowering phenotype/FT expression. However, they do not fully support the proposed tunable CO condensates model. Instead, the data essentially support the well-established conclusion that CO function requires functional NF-YB/YC.

It is important to note that all of the experiments were conducted using the 35S promoter-driven constructs, which raises concerns about the relevance of the observed gel-like condensates to native CO expression levels. The reviewer doubts that native CO could ever reach an expression level that promotes gel-like condensates, which could well be an artifact of the overexpression system. Therefore, the proposed condensation optimization system may not be necessary in the context of endogenous CO expression.

In addition, what is the evidence supporting CO/NF condensates functioning at FT locus for transcription? Any FISH data?

2. The claim of "low-order" and "high-order" assemblies may require further clarification as the data provided in the manuscript may not be sufficient to conclude the physical or molecular properties of these two types of condensates. It is important to ensure that the terminology used is precise and accurate, and it may be necessary to suggest alternative terms until there is more experimental evidence.

Moreover, the use of "gel-like" to describe one of the proposed condensates is concerning because it implies a solid phase separation. Without *in vitro* evidence to support this claim, it is important to be cautious about using such terminology.

3. Overinterpretation and misinterpretation of data is another major concern about the manuscript, and I highlighted a few but all in below.

a) The conclusion drawn from Extended Data 3 appears to be inconsistent with the results, as the data only show that NF-YC, but not NF-YB, can switch CO condensates.

b) While Fig. 2c supports the idea that the B-box domain is necessary for CO phase separation, it is not sufficient to conclude that it acts as the scaffolding motif.

c) In line 208-211, the authors make a claim based on Extended Data Fig.6 that may be overly broad or not entirely supported by the data.

d) The authors claim that deletion of NF-YC IDR regions results in increased gel-like CO condensates, but the evidence supporting this conclusion in Extended Data Fig. 9 is not entirely clear. Even if the claim is true, the data only suggest that the IDR of NF-YC is essential for their function, but not sufficient to conclude that they optimize CO condensation. The same limitation applies to the subsequent polyQ mutant study.

4. Some potential data conflicts within the manuscript that needs further clarification.

a) The conclusion drawn from the SPR data presented in Fig.3a is inconsistent with other genetic and cell biology data that have shown NF-YC to be the primary regulator of CO condensation.

b) The extremely low FRAP recovery rate of CO trimetric condensates appears to conflict with other FRAP data presented in the manuscript (e.g., Fig. S4). It is worth noting that even small condensates or soluble fluorescence failed to recover after photobleaching in cells with high-order CO condensates (Fig. 2d). It is also unclear why low-order condensates did not recover after photobleaching (Fig. 2d).

c) Figure 3b shows gel-like condensates in the CO+FT sample, which is confusing given that small condensates are expected.

d) How do the authors interpret that both reducing and increasing polyQ in NF-YC results in similar CO regulation pattern, and how that supports the optimization model?

Referee #2:

The work by Huang et al describes experiments to characterize the biochemical properties of the CO-NFYC9/NFYB2 trimer assembly, aimed at proving coassociation conditions. Notably, they find that the NF-YC9/NF-YB2 dimer maintains the proper CO assembly status on the FT promoter. Furthermore, the Bbox domain of CO and a short poly-Q tract in NF-YC9 are required. The work represents a considerable amount of confocal microscopy experiments with FRAP, coupled to some *in vitro* experiments. The *in vivo* part mostly relies on vast 35S-driven overexpressions, the biochemical conclusion relies on analysis of individual parts that do not represent the physiologic assembly of the components. Although appealing, the conclusions are not supported by the current data.

Major points.

The Authors describe nuclear condensates as being polymerized (l.119), gel-like (l.132), with high viscoelasticity (l.167), gel-like state (l.180), just to mention some. These conclusions indicate very precise physico-chemical properties of the CO-NFYB/C system. Yet, they are drawn from the analysis of confocal images of nuclei, either with or without FRAP observations. I doubt that such precise definition of a physical state can be inferred only by microscopy analyses without time-resolved analysis of particles.

The Authors do not provide any definition of gel-like/liquid properties of the analyzed assemblies defining the functional state of the complexes. E.g., while the Authors modify the proteins of interest by engineering motifs (polyQ) or domains (Bboxes, IDRs), they never modify the environment in which the proteins are dissolved. Feng et al., 2019, showed phase separation of FCA, a regulator of flowering in the autonomous pathway, by fusing it to a solubility agent and by performing tests where the crowding agent PEG8000 was added to the reactions. Such approaches require setting up a robust *in vitro* system, but it would provide compelling evidence of phase separation.

In extended Data 5, the single proteins (purified as soluble? Inclusion bodies?) clearly elute with the dead volume of the chromatography, indicating high MW complexes. A biochemist passionate of protein aggregates would likely find in this experiment a clear sign of irreversible aggregates. This is a major pitfall in the biochemical analysis of the fluidic properties of the complexes, as all other biochemical experiments might be affected by these findings.

Related to the above, to the best of my knowledge, the NFYB2/YC9 form soluble dimers, independently from CO: why weren't these complexes analyzed? Are these soluble? Is purified CO soluble?

FRAP experiments indicate that *nfy9* is necessary for diffusing CO condensates which otherwise aggregate in larger foci (fig. 1e). Based on these results, it might be inferred that 35S:mCherry-CO should behave like 35S:mCherry-CO 35S:NFYC9-GFP *ycT*. I believe that this control should be added.

Some links between the physical states of CO complexes and their physiological consequences are over-interpreted. For example, 'Taken together, these data indicate that NF-YC-dependent CO fluidic condensates are indispensable for the expression of the FT gene and floral transition.' and 'In the presence of both NF-YC9 and NF-YB2, the CO complex dramatically induced FT promoter-driven GUS activity in Arabidopsis protoplasts (Fig. 2f). These findings suggest the assembly status and material property-dependent function of CO/NF-YB2/NF-YC9 condensates in Arabidopsis seedlings (Fig. 1).'

While the different levels of FT transcription and the flowering time of transgenic plants depend on the genotype, whether they are caused by the physical properties of the aggregates is not demonstrated. There is a correlation that doesn't necessarily imply a causal relationship.

Additionally, I suppose that for the clusters to be active on FT transcription, they should sit on the FT promoter. In Arabidopsis there are two copies of such promoter, but many clusters are visible. Do they represent additional loci regulated by CO-NFYB2/YC9? If so, it might be more convincing to show expression of other targets, in addition to FT.

Another problematic issue relates to the use of the 35S promoter in all experiments. I am concerned that the use of such strong promoter might give artifacts. For example, in the experiments shown in fig. 2a-b, CO alone forms high order condensates in cells which contain physiological amounts of NFYB/C. When YC9 and YB2 are over expressed, they distribute CO to lower order complexes, but couldn't this reflect aspecific binding to several more loci, the effect being brought about by excess NFYB2/C9 caused by the 35S promoter? This experiment would be more convincing using NFYB2/C9s expressed under their own promoter. Furthermore, what is CO concentration at which the low/high order assembly are formed as compared to physiological level of the protein in inducing conditions?

Referee #3:

Huang and coworkers investigated CO with NF-YB2/NF-YC9/FT precisely regulates FT transcriptional activation via co-phase separation to control floral transition in Arabidopsis. The authors focus on the functional state of CO condensates in a B-box-motif-dependent manner for FT activation and floral transition, which NF-YC9 and NF-YB2 interact with CO to prevent the overassembly of CO into high-order condensates.

While this study is interesting, and the authors provided genetic and biochemical evidences to interrogate the material properties of intracellular condensates, I have several major comments.

The drawback of this manuscript is that only overexpression of CO was used to investigate its condensates. As we know, concentration could be a key factor in protein condensation. Also, CO protein concentration is in dynamic control by light. Under the physiological concentration, does CO functions via condensation? The authors should observe CO condensation in the native promoter-driven transgenic lines.

The main findings were overlaid on the basis that CO condensates are functional, but the evidence is lacking. The sole evidence might be deletion of two B-box motifs is required for the assembly of CO condensates. I suggest that mutation(s) that block condensation or replacement of oligomerization domain should be tested.

In the transient expression assay, the transfection efficiency varies a lot, resulting in varied concentrations of CO protein, therefore the appearance of three populations could be resulted from concentration variation. This data does not prove that under physiological condition, CO exists in these states.

The authors found that the dynamicity of CO condensates differed with and without NF-YC9. The data in Figure 1e, to me, is comparing condensed phase to diffused phase, of course the diffused phase has higher recovery rate. Unless the authors could specifically bleach the small condensates. Since that YC9 tunes the material property is the major point of this manuscript, the authors must improve the resolution of imaging to more carefully compare the dynamicity of condensed phases in big and small condensates. Also in Figures 2d, 3b, S10d, immediately after bleaching, the overall signal is reduced, which does suggest that the CO molecules are quite mobile. The inability of the bleached area to recover could be because the bleaching area is too large such that a large population of GFP-CO molecules are switched off and the amount available for exchange is too small. The authors should specifically bleach each condensate.

To test whether altered material properties influence CO function, the authors used the GUS reporter, which is helpful, but I think the binding of target DNA, either in vitro or in vivo using ChIP, is very important.

Minor points

1. In lines 748, 836, and legend of Figures 3, S6, "20HEPES" should be "20mM HEPES"?
2. FRAP buffer described in the method is inconsistent with that described in the legend.
3. Line 509: "MgCl" should be "MgCl₂"

** As a service to authors, EMBO Press provides authors with the possibility to transfer a manuscript that one journal cannot offer to publish to another EMBO publication or the open access journal Life Science Alliance launched in partnership between EMBO Press, Rockefeller University Press and Cold Spring Harbor Laboratory Press. The full manuscript and if applicable, reviewers' reports, are automatically sent to the receiving journal to allow for fast handling and a prompt decision on your manuscript. For more details of this service, and to transfer your manuscript please click on Link Not Available. **

Response Letter to Reviewer's comments

Referee #1:

This manuscript examines the molecular condensation of the flowering regulator CONSTANS (CO) and its regulation by two CO interactors, NF-YB and NF-YC. The authors observed the formation of two types of CO condensates: high-order, gel-like and low-order, fluidic. They discovered that the former type is predominantly mediated by CO homotypic interaction, while the latter type requires the participation of NF-YC and NF-YB. Using FT expression as a readout, they concluded that the fluidic CO-NF-YB/YC condensates are functional in promoting flowering transition, whereas the gel-like CO condensates are not. While I find the study intriguing in terms of CO-related cell biology and biophysical dynamics, the weak genetic evidence to biologically connect different CO condensate types with their function in flowering transition diminishes my enthusiasm. Moreover, I am concerned about whether the observed types of CO condensates and the proposed tunable system exist *in vivo*, given that the majority of experiments were performed using overexpression and transient expression systems. For the above reasons, I believe the story needs significant improvement before it can be published.

We acknowledge reviewer's concerns regarding the adequacy of our characterization of the physiochemical properties. CO-based photoperiod-dependent flowering presents significant challenges for direct observation of CO under most physiological conditions due to limitations in current imaging technologies. CO acts specifically in the companion cells of the phloem to promote flowering (An et al., 2004; Valverde et al., 2004), which poses difficulties for cell biology imaging, even with our advanced super-resolution spinning-disk confocal system featuring highly sensitive sCOMS with a quantum efficiency of up to 95%.

Therefore, we are compelled to adopt specific approaches to study functional phase separation and its resulting consequences for biomolecular interactions at low concentrations. These approaches are applicable to all nanomolar and nanometre-scale phase separation studies, aiming to transcend existing technical barriers to gain new insights.

Our carefully designed methods involve two key strategies. Firstly, we enhanced imaging sensitivity by elevating expression levels, employing the 35S promoter-driven expression, and conducting functional assays with FT-activation in corresponding plant cells (Fig. 1h, Fig. 2f, Fig. 4f). By achieving adequate molecular expression levels for imaging, our utilization of the 35S promoter-driven expression in *Arabidopsis* transgenic plants (Fig. 1) and the protoplast expression system (Fig. 2, Fig. 4) enables us to observe a diverse range of assembly states of the CO complex. These assembly states span from initial percolation clustering to the formation of large, irreversible aggregates under energy-trapping conditions. In parallel, from both systems using *Arabidopsis* genetics, flowering regulation, and transcriptional activation of FT-promoter, our observations encompass percolation clustering-based activation to large aggregation-based inhibition, allowing a full spectrum understanding of how CO-phase separation would affect its function. Secondly, by using the extremely low concentration of CO at the nanomolar level and utilizing quantitative titration, we have now elucidated the biophysical and biochemical mechanisms governing the assembly and interplay between NF-YB/YC and CO (Fig. 3b-d, Extended data Fig. 6d-f, new results). In addition, we have

supplemented our findings with new evidence from mathematical modelling, which supports the notion that percolation clustering of the CO complex enhances the association between CO and the binding region of the FT promoter (Fig. 1e, Extended data Fig. 1 d, e, new results).

Through these approaches, we reconstruct the multicomponent regulation of CO assembly, providing support for the primary conclusion of percolation-based CO activation. We contend that our carefully designed strategy and techniques represent one of the most sensitive and practical approaches for studying functional condensation, particularly during percolation clustering at nano- or sub-nanomolar concentrations.

Regarding to the genetic point raised by the reviewer, we agree the genetic evidence for characterizing condensation function is important. We have actually provided data of several transgenic Arabidopsis mutants carrying different NF-YC9 polyQ or IDR truncates that tune the material properties of NF-YC9 in condensation to support the importance of assembly-status-dependent function for complex condensation NF-YC9/CO/NF-YB.

We are thankful to the reviewer for highlighting key areas of concern. These points have offered us the opportunity to clarify and correct any misinterpretations from our original presentation. We should have provided a clearer correlation between the sizes and functions of protein condensates in both *in vitro* and *in vivo* studies, and better justified our use of overexpression in investigating the inherent biochemical properties of condensation and propagation. These investigations are meant to understand intrinsic protein physicochemical properties and the biophysical constraints on nucleation and phase growth, rather than directly implying the functional states of the observed condensation. To address concerns about the physiological concentration's influence on flowering, we've added a nano-scale condensation assay that displays phase transition during multi-component condensate formation at nanomolar concentrations, validating and highlighting the physiological relevance of our multicomponent condensation.

Major:

1. The manuscript presents intriguing data showing potential correlations between changes in CO condensates and flowering phenotype/FT expression. However, they do not fully support the proposed tunable CO condensates model. Instead, the data essentially support the well-established conclusion that CO function requires functional NF-YB/YC.

We appreciate this opportunity here for us to clarify the differentiate between complex formation and molecular condensation. The functional role of CO necessitates the presence of NF-YB/YC, indicating the involvement of components with physical interactions. However, this interaction alone for a complex formation does not elucidate the assembly states and material properties that govern functional regulation, which lies at the heart of the field of phase separation. The focus is not merely on identifying partners but rather on deciphering how their partnership and assembly strategies influence biological activities. There are multiple research articles and review papers explicitly outline the fundamentals of functional phase separation (Banani et al., 2017; Miao et al., 2023; Pappu et al., 2023).

The process of molecular condensation commences from direct binding and is then amplified via inter- and intra-multivalent interactions. Whether this process starts from strong "stickers" that bind with high affinity and form oligomer or complex, or weaker "stickers" that engage multiple weak interactions, it does not alter the inherent nature of phase separation. However, it can impact the progression of condensation and, notably, the nucleation threshold in terms of time and critical transition concentration. In this instance, multi-component CO condensation indeed starts with an initial interaction with NF-YC, as previously reported, as well as our newly discovered direct interactions between CO and NF-YB, and utilizes NF-YB's weak multivalent interaction abilities to maintain flexible assembly of multi-component condensation, and therefore reach tuneable CO assembly status (Fig. 3a and Extended Data Fig. 6). The existing evidence of complex formation between CO and the NF-YB/YC complex doesn't contradict any new conclusions regarding molecular condensation-regulated CO-activation. Instead, it forms part of it and engages at the early moment of clustering and nucleation during phase separation. We hope this explanation helps clarify the differences between identifying complex constituents and regulating component assembly through multivalent molecular condensation to fine-tune biochemical activity.

It is important to note that all of the experiments were conducted using the 35S promoter-driven constructs, which raises concerns about the relevance of the observed gel-like condensates to native CO expression levels. The reviewer doubts that native CO could ever reach an expression level that promotes gel-like condensates, which could well be an artifact of the overexpression system. Therefore, the proposed condensation optimization system may not be necessary in the context of endogenous CO expression.

Please refer to the above response to the first paragraph's overall evaluation.

We thank the reviewer for raising up this concern to help up improve the quality of this study. Below, we provide our response on how to address this issue within the current stage of technology development, enabling a conceptual contribution to the field of nano-scale condensation studies.

The overarching process of associative molecular condensation starts with percolation clustering, progresses through phase transition at a crucial percolation concentration, and culminates in further associative interactions. In numerous instances, including the multi-component CO condensates in our study, this process can develop beyond the critical saturation concentration of phase separation, exhibiting droplet-like behavior (Miao et al., 2023; Pappu et al., 2023; Shen et al., 2023).

Firstly, we would like to address that 35S promoter-driven expression of CO was applied not to prove that a similar level of condensation occurs under physiological conditions. Instead, it was designed to understand the underlying mechanism of tuneable molecular assembly of CO/NF-YB/YC complexes for the following reasons, which was also supported with additional functional validation.

- a. The 35S promoter-CO is functional and regulated by known photoperiod switches (Valverde et al., 2004). When 35S promoter-driven CO transgenic plants were exposed

to light, we observed that the initially non-existent CO fluorescent signals gradually accumulated (as seen in Extended Data Fig. 1a, b). During this process, the protein level ranged from negligible to high, encompassing all possible physiologically relevant concentrations of CO protein, which supports the functionality of the 35S-driven expression line.

- b. Due to the fluorescent image sensitivity, our imaging used the possible sensitive equipment to follow the initial nucleation/clustering of CO condensates with a spinning disk confocal (Yokogawa CSU-W1) and a super high quantum-yield camera (up to 95% QE, ORCA-Fusion BT Digital CMOS camera). The native promoter condition does not allow us to see both diluted and condensed phases, due to the limited photons for capturing.
- c. To the best of our understanding, no other studies have managed to achieve the level of detection sensitivity in plants that we have accomplished in this research. Notably, the ability to detect the initial nucleation of CO-condensates is paramount, as it reflects the functional states. By using the 35S promoter, we were able to extend the range of assembly states of a functional CO-complex. This enables us to embark on fundamental studies of CO condensation and function, starting from nucleation and clustering, which is the key functional assembly status found from this study.

Secondly, the native CO is specifically expressed in phloem companion cells to promote flowering with a daily rhythm in CO abundance (An et al., 2004; Valverde et al., 2004), which can be challenging to be observed due to the short laser penetration depth at high magnification, especially for those transcription factors with low abundance.

Thirdly, to mimic the low abundance of transcription factor *in vivo*, we have provided further biochemical assays by single particle imaging to analyze the multi-component assembly of CO/NF-YC9/NF-YB2 complex at the nanoscale. The single particle image analysis clearly demonstrates the intermolecular interactions of CO, NF-YB2, and NY-YC9, either bi- or tri-component assembly, start from a nanomolar concentration (Fig. 3b; Extended Data Fig. 6d). Most importantly, it shows the finetuning of CO accumulation by NF-YB and NY-YC, cooperatively, to prevent CO nanocluster over-assembly (Fig. 3c, d; Extended Data Fig. 6e, f). Those data, together with the bulky characterisation assays on CO condensation (Extended Data Fig. 6 g-i), unveil the regulatory role of NF-YB2 and NF-YC9 on CO assembly across a spectrum of protein abundances spanning from nanoscale to macroscale (see pages 10-11), which also supports the robustness of employing the 35S promoter to elucidate the mechanism of tuneable molecular assembly of CO/NF-YB/YC complexes *in vivo*. It is important to note that phase separation in thermodynamics does not inherently incorporate timing, which is a kinetic concept. Therefore, through concentration-dependent assays, we were able to describe the progressive assembly of biomolecules, thereby addressing kinetics as intended.

Lastly, determining the physiological concentration of the CO proteins in inducing conditions presents a significant challenge. To our knowledge, no such study has been done on transcription factor in plants. Again, the low/high order assembly of CO in transient expression system do not necessarily capture the physiological level of the CO proteins. It was applied to

enable robust imaging and quantitative analysis to understand the tuneable molecular assembly of CO/NF-YB/YC complexes.

In summary, we conducted substantial additional single-molecule image experiments at nanomolar protein concentrations, and provided further explanation on how the phase transition stages we presented allow us to track the progression of nanomolar condensation to understand their assembly strategies and the tuneable material properties. This gives us insights into how the various forms of condensation - from nano- to micro- or meso-scale - align with their functionality and physiological relevance.

In addition, what is the evidence supporting CO/NF condensates functioning at FT locus for transcription? Any FISH data?

There is a strong foundation in the study of CO/NF-Y's role in activating *FT* transcription (Frontini et al., 2004; Gnesutta et al., 2017; Kahle et al., 2005; Lv et al., 2021). We thank the reviewer's suggestion on providing directing evidence showing CO/NF-Y condensates on *FT* locus. Although our laboratory doesn't specialize in FISH experiments, we have tried our best to establish the new system and perform additional experiments upon Reviewer's request. We designed FISH probes recognizing the CO binding motifs on *FT* promoter in mCherry-CO transgenic lines. We eventually could detect FISH signal in the root cells, although the hybridization efficiency is low (given the nature that there is maximum only two copies of *FT* promoter in one nucleus, that's also why most people are doing RNA-FISH but not DNA-FISH to detect nucleotide on demand). However, the mCherry-CO signal was not visible after multiple steps of handling, including chemical treatments and high or low temperature treatments during FISH experiment. Alternatively, we also have tried to image the mCherry-CO signal first right after fixation and then perform FISH to detect *FT* promoter. But we eventually failed to localize back the original cells that has been recorded the mCherry-CO signal due to most of the original cells were gone after multistep of washing. Additionally, we also tried to establish a dCas9 recognition assay by creating a fusion of dCas9-GFP which contains the gRNA that targets the *FT* locus and co-expressing this system with mCherry-CO. But the expression of such dCas9 recognition system was too low to allow us get enough data for quantitative analysis for a conclusive claim. Hence, despite investing over half a year in extensive efforts and numerous optimization attempts, we regret to report that we were unable to surmount the technical challenges posed by the assays requested by the reviewer.

Here, as an alternative approach to support CO-percolation-mediated activation of *FT*-transcription, we have conducted mathematical modelling based molecular simulation, which showed locally accumulated CO/NF complex would enhance its binding affinity on *FT* promoter. The new data was now showing in Fig. 1e and Extended Data Fig. 6d, e (please also see pages 6). Additionally, we indeed have shown adding *FT* promoter, but not non-CO target DNA fragment would alter the fluidity of CO/NF condensates (now in Fig. 3e, f and Extended Data Fig. 6j, k), suggesting those CO/NF condensates recognise the *FT* promoter. We hope that the reviewer finds our explanation of the technical challenges with substantial trials and our newly implemented mathematical simulation-based approach acceptable.

2. The claim of "low-order" and "high-order" assemblies may require further clarification as the data provided in the manuscript may not be sufficient to conclude the physical or

molecular properties of these two types of condensates. It is important to ensure that the terminology used is precise and accurate, and it may be necessary to suggest alternative terms until there is more experimental evidence.

We deeply appreciate the reviewer for bringing this to our attention. We totally agree with the reviewer that the terms "low-order" and "high-order" are a bit confusing and may not sufficiently clarify the functional states of different CO condensates. To address this, we have changed the terminology of different CO condensation statuses. We have introduced a new analytical approach that quantitatively classifies the relative CO condensation statuses by analyzing the CO signal variance within individual nuclei (Extended Data Fig. 2a and Methods section, page 20). Higher CO signal variance indicates greater local CO accumulation, indicative of uniform percolation clustering with spherical condensates under microscopy, in contrast to a diffusive pattern with low signal variance and FRAP irreversible larger aggregates, as demonstrated in the FRAP assay-based fluidity test presented in Extended Data Fig. 2c. Upon compiling the overall CO signal variance data from different combinations of CO, NF-YC9, and NF-YB2 co-expression, we observed three distinct populations, namely diffuse, spherical condensate and irreversible aggregate (Extended Data Fig. 2b). This classification scheme allows us to better correlate CO functionality with the physical properties of CO condensation.

Our subsequent experiments primarily focused on spherical condensation, as CO/NF-YB/NF-YC co-expression predominantly displayed this population (Fig. 2a, Fig. 4c, and Extended Data Fig. 8d). We conducted extensive FRAP experiments to comprehend the physical properties of CO condensation. This classification approach provides insights into the material properties of the assemblies, which are critical determinants for CO functionality.

For clarification, we have added more detailed information on how we quantitatively classified different populations of CO condensation (refer to pages 7-8).

Moreover, the use of "gel-like" to describe one of the proposed condensates is concerning because it implies a solid phase separation. Without *in vitro* evidence to support this claim, it is important to be cautious about using such terminology.

Gel-like condensates fundamentally differ from solid phase separation; they exhibit reversible properties within non-equilibrium states, unlike non-reversible amorphous assemblies or fibrils. We agree with the reviewer that we should exercise caution when describing CO/NF-YC9/NF-YB2/FT macromolecular condensates *in vivo* as exhibiting gel-like properties. This term is intrinsically linked to the concept of molecular diffusion within specific conditions and timescales and should be reserved primarily for *in vitro* descriptions aiming at understanding how constituents affect the molecular dynamics of assemblies. Following the reviewer's advice, we have replaced "gel-like" to "slow-diffusive", which is directly indicated by FRAP assay, to prevent potential misleading in the manuscript and all figures (Fig. 2c-e; Fig. 4d; Extended Data Fig. 4; Extended Data Fig. 9a-c; Extended Data Fig. 10d, e).

3. Overinterpretation and misinterpretation of data is another major concern about the manuscript, and I highlighted a few but all in below.

We appreciate Reviewer for these points that allow us to clarify our statements and avoid confusion. Please see response below.

a) The conclusion drawn from Extended Data 3 appears to be inconsistent with the results, as the data only show that NF-YC, but not NF-YB, can switch CO condensates.

We appreciate the chance to clarify the synergistic roles of NF-YC and NF-YB in modulating CO condensation cooperatively. Firstly, as illustrated in Extended Data Figure 3c, NF-YC9 became incorporated into CO condensates when co-expressed with CO, whereas NF-YB2 alone could not integrate into CO condensates. However, NF-YC9 alone only partially transformed irreversible CO assemblies into a spherical condensation status (please refer to Fig. 2a for the population distribution of different types of CO condensation corresponding to Extended Data Figure 3c).

Secondly, in the presence of NF-YC9, additional NF-YB2 was able to integrate into CO condensates. Consequently, the tri-component co-expression of NF-YC9, NF-YB2, and CO led to a significant shift of CO condensates towards mostly fluidic spherical condensates, corresponding to the highest biochemical activity states (refer to Fig. 2a-e, Fig. 1f, and Fig. 1h). Therefore, our results suggest that while NF-YB2 alone did not induce the switch in CO condensation, it operates in an NF-YC dependent manner to achieve this effect.

To prevent any potential confusion, we have carefully reviewed our manuscript and provided a clear interpretation of Extended Data Fig. 3 in the results section (page 8). We trust that this additional information enhances the clarity of our findings.

b) While Fig. 2c supports the idea that the B-box domain is necessary for CO phase separation, it is not sufficient to conclude that it acts as the scaffolding motif.

We have now rephrased this point to avoid overclaim in the revised manuscript in page 8 for a more accurate interpretation.

c) In line 208-211, the authors make a claim based on Extended Data Fig.6 that may be overly broad or not entirely supported by the data.

We agree with the reviewer such claim was not well-supported at here from only Extended Data Fig. 6a-c, which only indicate the biophysical properties of CO, NF-YC9 and NF-YB2 homotypic condensation. We have now revised this sentence to avoid over-claiming (page 10). Indeed, in the following SPR assay (Fig. 3a) and *in vitro* reconstitution assay of bi- or tri-component complex assembly either in bulky solution (Fig. 3e, f; Extended Data Fig. 6g-k) or newly added results at nano molar scale (Fig. 3b-d; Extended Data Fig. 6d-f), we clearly demonstrated CO/NF-YC9 co-assembly still displayed relatively less flexibility by showing amorphous condensation and slow decay in SPR sensorgrams. Only with the addition of NF-YB2, the tri-component condensation showed spherical shapes, indicating a reduction of the surface tension, which is in a NF-YC9 dependent manner. We have now re-emphasized the function of NF-YB2 in the complex assembly in page 11.

d) The authors claim that deletion of NF-YC IDR regions results in increased gel-like CO

condensates, but the evidence supporting this conclusion in Extended Data Fig. 9 is not entirely clear. Even if the claim is true, the data only suggest that the IDR of NF-YC is essential for their function, but not sufficient to conclude that they optimize CO condensation. The same limitation applies to the subsequent polyQ mutant study.

We apologize for the unclear presentation of data in this section. Here, we aim to clarify our experimental procedures regarding how truncated or mutated versions of NF-YC influence CO condensation. Initially, we quantified the populations exhibiting different CO assembly patterns, as standardized in Extended Data Fig. 2. We observed that IDR truncations or polyQ mutations in NF-YC did not alter the overall CO assembly pattern compared to full-length CO when co-expressed with both NF-YC9 and NF-YB2 (refer to Extended Data Fig. 8c, d and Fig. 4b, c). All variants predominantly exhibited spherical condensation. To better illustrate this status, we have replaced the term "low-order CO assembly" with "spherical condensation" (please also see page 7).

Subsequently, we assessed whether these truncations or mutations affected the material properties of the major population of CO condensates using FRAP assay. We observed an increase in the proportion of slow-diffusive (originally termed as Gel-like) CO condensate (Extended Data Fig. 9a-c, Extended Data Fig. 10d, e, and Fig. 4d).

We acknowledge that we initially presented only partial representative images of the FRAP assay on spherical CO condensates. To rectify this, we have now provided a comprehensive overview of the characterization to prevent any potential misinterpretation (Extended Data Fig. 9a and Extended Data Fig. 10d). Additionally, we have revised the phrase "optimize CO condensation" to "optimizing the material properties of CO condensation" to align with the FRAP results (page 13).

Furthermore, we made efforts to purify all the IDR-truncated versions of NF-YC9, as depicted in Extended Data Fig. 8a. Unfortunately, we encountered challenges in obtaining soluble recombinant proteins. This clarification and discussion have been incorporated into our revised manuscript (pages 12-13).

4. Some potential data conflicts within the manuscript that needs further clarification.

a) The conclusion drawn from the SPR data presented in Fig.3a is inconsistent with other genetic and cell biology data that have shown NF-YC to be the primary regulator of CO condensation.

We would like to clarify that we did not claim NF-YC plays more important roles in regulating CO condensation. Although our genetic experiments and cell biology images were more focused on NF-YC9 and its truncation versions, earlier studies have strong genetic evidence showing NF-YB is also indispensable for CO functionality (Kumimoto et al., 2008; Kumimoto et al., 2010). Consistently, our cell biology image and *in vitro* biochemical assays all demonstrate NF-YC9 and NF-YB2 cooperatively maintain the fluid property of CO condensation. Particularly, NF-YB2 provide more conformational flexibility to prevent CO over-assemble in the presence but not absence of NF-YC9 (Fig. 2a, e; Fig.3; Extended Data Fig. 6). We have now revised the

statements and discussion in page 4 and page 16 to give a better interpretation for those data. We hope such clarification have addressed the reviewer's concerns on this point.

b) The extremely low FRAP recovery rate of CO trimetric condensates appears to conflict with other FRAP data presented in the manuscript (e.g., Fig. S4). It is worth noting that even small condensates or soluble fluorescence failed to recover after photobleaching in cells with high-order CO condensates (Fig. 2d). It is also unclear why low-order condensates did not recover after photobleaching (Fig. 2d).

We apologize for the unclear data presentation. The FRAP recovery curves shown in original Extended Data Fig. 4 are just indicative of different CO condensation material properties in the indicated CO assembly status. It shows both liquid and slow-diffusive CO condensation are excited in the population spherical CO condensation. While quantification of the proportions of CO condensation with either liquid or slow-diffusive material properties was show now in Fig.2e, which clear demonstrates most (95%) of the CO/YB/YC trimetric condensates display liquid like material properties. We have now parallely shown the representative FRAP images, curves, and the percentage quantification of each CO populations with either liquid or slow-diffusive material properties in Fig. 2c-e and Extended Data Fig. 4a for a clear data presentation.

It should be noted a small proportion of fluorescence recovery could still be observed in slow-diffusive CO condensations as shown in the FRAP curves (Fig. 2d, Extended Data Fig. 4a), those actually correspond to the dynamic diffusion of the soluble fluorescent proteins in the nucleus, which we did not present clearly in initial figures due to the high intensity of the CO condensation. We have now adjusted the contrast of the images to clearly display the soluble fluorescent signal in the nucleus. Those images are now showing in Extended Data Fig. 4b, which shows the recovery of the diffused fluorescent signal in bleached regions postbleach and also demonstrates the slow-diffusive CO condensates do not have significant molecular exchanged with uncondensed surrounding (emphasized in page 9). Please be noted that the size is not necessary to be aligned with the material properties of the CO condensation which were classified and quantified by different methods in this study.

c) Figure 3b shows gel-like condensates in the CO+FT sample, which is confusing given that small condensates are expected.

We apologize for the misunderstanding derived from low image quality after compression in our initial submission, as high contrast and image compression made individual assemblies appear blurred, resembling large condensates. We've corrected this in our revised manuscript by improving the image quality for presentation with reduced compression. Please refer to the enhanced images Fig. 3e for clarity.

d) How do the authors interpret that both reducing and increasing polyQ in NF-YC results in similar CO regulation pattern, and how that supports the optimization model?

Thanks for pointing out this interesting question, allow us to elaborate this further. PolyQ motif is well-known to cause protein oligomerization or aggregation in several diseases

(Peskett et al., 2018). We found increasing the number of Q residue in NF-YC-9 would result in a more rigid CO/NF-Y complex condensation, this is consistent with the common sense that polyQ promotes protein interaction. Surprisingly, similar results were also observed with no polyQ in NF-YC9, suggesting a potential effect of polyQ on NF-YB and NF-YC interaction or engagement in CO/NF-Y complex condensation. We have discussed this in the previously submitted manuscript (now in page 17) and elaborated further now to interpret our results.

Referee #2:

The work by Huang et al describes experiments to characterize the biochemical properties of the CO-NFYC9/NFYB2 trimer assembly, aimed at proving coassociation conditions. Notably, they find that the NF-YC9/NF-YB2 dimer maintains the proper CO assembly status on the FT promoter. Furthermore, the B-box domain of CO and a short poly-Q tract in NF-YC9 are required. The work represents a considerable amount of confocal microscopy experiments with FRAP, coupled to some in vitro experiments. The in vivo part mostly relies on vast 35S-driven overexpressions, the biochemical conclusion relies on analysis of individual parts that do not represent the physiologic assembly of the components. Although appealing, the conclusions are not supported by the current data.

We appreciate the reviewer's inquiries regarding the 35S promoter driven overexpression. In the following section, we elaborate on the current technological barriers to studying nano-scale condensation and function. Additionally, we outline our interdisciplinary approach aimed at providing practical and feasible solutions to provide the best possible quantitative analysis to advance our understanding in this challenging yet critically important research domain. Specifically, we utilized nano-scale reconstitution to address questions related to physiologic assembly and kinetics, along with mathematical modeling to investigate how nano-scale condensation enhances transcriptional activation. Please see the following detailed point to point response.

Major points.

The Authors describe nuclear condensates as being polymerized (l.119), gel-like (l.132), with high viscoelasticity (l.167), gel-like state (l.180), just to mention some. These conclusions indicate very precise physico-chemical properties of the CO-NFYB/C system. Yet, they are drawn from the analysis of confocal images of nuclei, either with or without FRAP observations. I doubt that such precise definition of a physical state can be inferred only by microscopy analyses without time-resolved analysis of particles.

Gel-like condensates describe reversible properties with slow diffusion but at non-equilibrium states. This term is intrinsically linked to the concept of molecular diffusion within specific conditions and timescales and should be reserved primarily for in vitro descriptions aimed at understanding how constituents affect the molecular dynamics of assemblies (Pappu et al., 2023; Miao et al., 2023). Here, such classification was introduced to compare with dynamic small clusters, instead of providing a precise physico-chemical parameters. We agree with the reviewer that we should exercise caution when describing CO/NF-YC9/NF-YB2/FT macromolecular condensates *in vivo* as exhibiting gel-like properties. Following the reviewer's advice, we have replaced "gel-like" to "slow-diffusive", which is directly indicated by FRAP

assay, to prevent potential misleading in the manuscripts and all related figures. We also have avoided to use those terminologies without any experiment evidence.

The Authors do not provide any definition of gel-like/liquid properties of the analyzed assemblies defining the functional state of the complexes. E.g., while the Authors modify the proteins of interest by engineering motifs (polyQ) or domains (Bboxes, IDRs), they never modify the environment in which the proteins are dissolved.

In the previous section, we have addressed this issue through a revised analysis and definition to clarify the functional correlation with material properties in living plant system. We have now replaced the term with "slow-diffusive" and provided its definition upon first use for living plant imaging. Additionally, we have revised the statement in the results section to clarify that slow-diffusive condensates describe reversible properties with slow diffusion but at non-equilibrium states. This term was introduced to facilitate comparison with dynamic small clusters rather than providing precise physico-chemical parameters (refer to page 7).

Regarding the query about the "environment" in the second half of the question, the transgenic plants were indeed examined and compared under physiological environmental conditions to investigate their photoperiodic response *in vivo*. This analysis is detailed in Extended Figures 9e, f and 11. *In vitro*, due to the high aggregation prone properties of polyQ mutant recombinant protein variants, we could not obtain soluble proteins for biochemical assays for detailed *in vitro* characterization as we performed for wild type full-length NF-YC9. It is actually consistent with *in vivo* slower recovery of CO condensates in the presence of these polyQ mutants (Fig. 4).

Feng et al., 2019, showed phase separation of FCA, a regulator of flowering in the autonomous pathway, by fusing it to a solubility agent and by performing tests where the crowding agent PEG8000 was added to the reactions. Such approaches require setting up a robust *in vitro* system, but it would provide compelling evidence of phase separation.

We appreciate the reviewer for raising this point and giving us an opportunity to elaborate on the use of crowding reagents for phase separation studies. Initially, crowding agents were frequently used in phase separation research to drive proteins to form droplets with high surface tension. This strategy is reminiscent of the chemical screening methods employed by structural biologists during early crystallography studies. Under such conditions, uniform folding led to crystal packing, while disordered regions formed liquid droplets.

Crowding agents, whether chemical-based or not, operate differently depending on size, chemical effects, and molecular weight. Most of these agents induce liquid droplet formation in a manner specific to the chemical bonds introduced by the molecules. Several crowding agents are typically used in biological systems. These include protein-based agents (Delarue et al., 2018) and small molecule-based agents like methylcellulose, ficoll, dextran, and polyethylene glycol (PEG) (Annunziata et al., 2002; González et al., 2003; Lodge et al., 2018).

PEG, particularly, acts swiftly by displacing water molecules and introducing hydrophobic interactions, resulting in a mixture of potential physiological and non-physiological valencies. This property has garnered significant interest in the field of phase separation, where PEG is

utilized for its crowding effect. However, the field of phase separation is gradually moving away from this method as a definitive means to replicate phase separation, instead favoring buffer conditions that are more physiologically relevant. Additionally, while a cytoplasm-based system has been suggested as a better alternative method, the isolation of pure cytosols from plant cells presents a considerable challenge to avoid contamination from vacuole components.

PEG is commonly used to augment the multivalent interaction of weakly interacting proteins, which are diffusive in solution via homotypic interactions. These interactions necessitate extensive characterization under various biochemical conditions. However, if a protein can drive homotypic interaction without the addition of a crowding agent, this is considered the better standard in the field for biochemical assays.

Moreover, if a protein contains numerous self-associative motifs and begins to form amorphous oligomers *in vitro*, as in the case of NF-YC9 and CO in this study, the addition of PEG through concentration titration and molecular-weight titration may simply cause protein aggregation. In our system, NF-YB reflects the condensation modulator to balance the YC and CO's intrinsic biochemical ability in forming higher oligomerization. In such complex protein system, adding PEG would introduce unexpected artifacts.

We hope that our detailed explanation of phase separation and the effects of PEG have clarified our thoughtful consideration and rational design process.

In extended Data 5, the single proteins (purified as soluble? Inclusion bodies?) clearly elute with the dead volume of the chromatography, indicating high MW complexes. A biochemist passionate of protein aggregates would likely find in this experiment a clear sign of irreversible aggregates. This is a major pitfall in the biochemical analysis of the fluidic properties of the complexes, as all other biochemical experiments might be affected by these findings.

Thank you for bringing up this topic, it provides us with an opportunity to elucidate our evaluation criteria to analyse the elution profile from size-exclusion columns for multivalent intrinsically disordered proteins.

The assembly and retention of proteins on size exclusion columns are influenced by factors such as protein molecular weight, conformation, and hydrodynamic radius. In this instance, we utilized Superdex 200 HiLoad 16/600 column, which separates proteins with molecular weights ranging from 10K to 600K. Our protein peaks are close to, but not within, the void volume. Both CO and NF-YC proteins form high oligomeric structures yet remain soluble under our optimized elution condition with 500 mM NaCl, as shown in Extended Data Figure 5. While doing *in vitro* biochemical assays we reduced the ionic strength to more physiological relevant concentration at 150 mM NaCl. We have now added another set of sedimentation assay for CO and YC9 proteins in the buffer containing 150 mM NaCl. Both these two proteins were remained in the supernatant after ultracentrifugation (100,000 x g for 30 min), indicating they are soluble in the buffer condition applied in this study (Extended Data Fig. 5b)

In general, while Superose 6 may be capable of separating slightly larger oligomeric species, it provides considerably lower resolution than Superdex 200. Thus, it isn't particularly beneficial

for studying flexible IDR-containing proteins with oligomeric motifs, such as CO and NF-YC here, due to their high self-association and high hydrodynamic radius. With different optimizations, we have managed to obtain soluble high oligomeric and reversible proteins, although peak is close to void peak on SEC. Another similar example of optimization and obtaining protein soluble oligomeric assemblies is a phase separation protein yeast Aip5 reported (Xie et al., 2019).

We are grateful to the reviewer for underscoring the significance of this aspect. We are highly confident in our ability to characterize protein solubility, oligomerization, associative interactions, and material properties accurately. These biochemical and biophysical characterizations are crucial for establishing a reconstitution system that faithfully recapitulates the dynamics of multicomponent assembly through stoichiometric titration.

Related to the above, to the best of my knowledge, the NFYB2/YC9 form soluble dimers, independently from CO: why weren't these complexes analyzed? Are these soluble? Is purified CO soluble?

We thank the reviewer for asking this, which is a fundamental question for all phase separation studying when using the existing knowledge. Here, we elaborate our dissection with the following facts and our understanding.

Firstly, it is well-known NF-YB/YC associate together in vivo which thus enable NF-YB to be transported from cytoplasm to nucleus as NF-YB by itself lacks the nuclear localization signal (Hackenberg et al., 2012; Kahle et al., 2005; Nardini et al., 2013). In such studies, the importance of tri-partnership is established, but not what would be the dynamic stoichiometry and associative assembly to form a functional complex, which is the functional nano-condensates described in this study. Indeed, we have also analysed the NF-YB2 and NF-YC9 complexes in our manuscript. In the initial figures, we have shown a robust interaction between the recombinant full-length proteins of NF-YB2 and NF-YC9 with a $K_d=41.6$ nM (Fig. 3a, in the panel of NF-YB2 flowing in chip coated with NF-YC9). In the newly conducted experiment, we have observed NF-YB2 associates with NF-YC9 at single particle level (Extended Data Fig. 6d). Those data together indicate the purified proteins still maintain their activities in vitro regarding intermolecular interaction. In addition, from the SPR sensorgram of NF-YB2 flowing in chip coated with NF-YC9, after initial first-order association, a relatively fast and repeatable dissociation at a series concentration of NF-YB2 could be observed during washing step (Fig. 3a), suggesting NY-YB2 and NY-YC9 form soluble complex but not irreversible aggregates in vitro. Finally, regarding CO's solubility, please refer to the response to the last comment.

Secondly, studies in structural biology focus solely on interaction motifs of NF-YC/NF-YB/CO, which describe a heterodimeric form and employ a "lock-and-key" binding approach to highlight strong affinity and interfaces. However, these studies typically do not encompass the full-length proteins, which undergo multivalent associations for functional assembly, utilizing the dimeric interface as a strong adhesive for condensation. It is imperative to consider both aspects and differentiate them in macromolecular assembly studies.

Thirdly, Co-IP-based experiments are instrumental in isolating interaction partnerships in various scenarios, including monomeric proteins with robust interactive surfaces and weak to intermediate affinity partnerships of multivalent interaction partners. However, these experiments do not provide information on stoichiometry, as the chemicals and isolation methods used in Co-IP experiments can alter cooperative binding ability and change physiological stoichiometry. While Co-IP experiments may yield a binary outcome indicating partnership, they do not capture the nuances of dynamic assembly states of multicomponent.

We believe that our elucidation of protein interactions using information from both structural biology and IP-based biochemical assays provides a scientifically justified rationale for why our designed biochemical and biophysical experiments, conducted at the single-molecular level, strongly support our investigation of such multi-component assembly studies.

FRAP experiments indicate that *nfy9* is necessary for diffusing CO condensates which otherwise aggregate in larger foci (fig. 1e). Based on these results, it might be inferred that 35S:mCherry-CO should behave like 35S:mCherry-CO 35S:NFYC9-GFP *ycT*. I believe that this control should be added.

We appreciate the reviewer for the suggestions. Following the reviewer's advice, we have now added the data of 35S:mCherry-CO FRAP experiments into Fig.1f, and measured the FRAP recovery intensities as shown in Fig.1g.

Some links between the physical states of CO complexes and their physiological consequences are over-interpreted. For example, 'Taken together, these data indicate that NF-YC-dependent CO fluidic condensates are indispensable for the expression of the FT gene and floral transition.' and 'In the presence of both NF-YC9 and NF-YB2, the CO complex dramatically induced FT promoter-driven GUS activity in Arabidopsis protoplasts (Fig. 2f). These findings suggest the assembly status and material property-dependent function of CO/NF-YB2/NF-YC9 condensates in Arabidopsis seedlings (Fig. 1).' While the different levels of FT transcription and the flowering time of transgenic plants depend on the genotype, whether they are caused by the physical properties of the aggregates is not demonstrated. There is a correlation that doesn't necessarily imply a causal relationship.

Thanks for the suggestions. We have now revised them as the following to avoid potential over-statement. 'These findings suggest that NF-YC-dependent CO fluidic condensates are strongly linked to FT gene expression, crucial for floral transition.' on page 7; and 'These findings suggest a likely mechanistic relationship between the assembly status and material property-dependent function of CO/NF-YB2/NF-YC9 condensates in FT-expression (Fig. 1).' on page 9.

Additionally, I suppose that for the clusters to be active on FT transcription, they should sit on the FT promoter. In Arabidopsis there are two copies of such promoter, but many clusters are visible. Do they represent additional loci regulated by CO-NFYB2/YC9? If so, it might be more convincing to show expression of other targets, in addition to FT.

We thank the reviewer's suggestion on providing directing evidence showing CO/NF condensates on FT locus. Although our laboratory doesn't specialize in FISH experiments, we have tried our best to establish the new system and perform additional experiments upon Reviewer's request. We designed FISH probes recognizing the CO binding motifs on FT promoter in mcherry-CO transgenic lines. We eventually could detect FISH signal in the root cells, although the hybridization efficiency is low (given the nature that there is maximum only two copies of FT promoter in one nucleus, that's also why most people are doing RNA-FISH but not DNA-FISH to detect nucleotide on demand). However, the mCherry-CO signal was not visible after multiple steps of handling, including chemical treatments and high or low temperature treatments during FISH experiment. Alternatively, we also have tried to image the mCherry-CO signal first right after fixation and then perform FISH to detect FT promoter. But we eventually failed to localize back the original cells that has been recorded the mCherry-CO signal due to most of the original cells were gone after multistep of washing. Additionally, we also tried to establish a dCas9 recognition assay by creating a fusion of dCas9-GFP which contains the gRNA that targets the FT locus and co-expressing this system with mCherry-CO. But the expression of such dCas9 recognition system was too low to allow us get enough data for quantitative analysis for a conclusive claim. Hence, despite investing over half a year in extensive efforts and numerous optimization attempts, we regret to report that we were unable to surmount the technical challenges posed by the assays requested by the reviewer

Here, as an alternative approach to support CO-percolation-mediated activation of FT-transcription, we have conducted mathematical modelling based molecular simulation, which showed locally accumulated CO/NF-Y complex would enhance its binding affinity on FT promoter. The new data was now showing in Fig. 1e and Extended Data Fig. 1d, e (please also see pages 6 for the new results). Additionally, we indeed have shown adding FT promoter, but not non-CO target DNA fragment would alter the fluidity of CO/NF-Y condensates (now in Fig. 3e, f and Extended Data Fig. 6j, k), suggesting those CO/NF-Y condensates recognise the FT promoter.

In addition, CO is not only for FT. It is known that CO is a transcriptional factor that regulate multiple promoters (Samach et al., 2000), such as FT and SOC1 (Hou et al., 2014), while FT is an essential floral transition target. We have now also revised these statements in results for accuracy on page 7.

We hope that the reviewer finds our explanation of the technical challenges with substantial trials and our newly implemented mathematical simulation-based approach acceptable.

Another problematic issue relates to the use of the 35S promoter in all experiments. I am concerned that the use of such strong promoter might give artifacts. For example, in the experiments shown in fig. 2a-b, CO alone forms high order condensates in cells which contain physiological amounts of NFYB/C. When YC9 and YB2 are over expressed, they distribute CO to lower order complexes, but couldn't this reflect a specific binding to several more loci, the effect being brought about by excess NFYB2/C9 caused by the 35S promoter? This experiment would be more convincing using NFYB2/C9s expressed under their own promoter. Furthermore, what is CO concentration at which the low/high order assembly are formed as compared to physiological level of the protein in inducing conditions?

Thanks for point out this important point for our research and also for the entire field of studying nano-condensation in signaling. Please refer to the above response in pages 1-2 for the overall evaluation to the same concern. Below, we provide our response on how to address this issue within the current stage of technology development, enabling a conceptual contribution to the field of nano-scale condensation studies.

The overarching process of associative molecular condensation starts with percolation clustering, progresses through phase transition at a crucial percolation concentration, and culminates in further associative interactions. In numerous instances, including the multi-component CO condensates in our study, this process can develop beyond the critical saturation concentration of phase separation, exhibiting droplet-like behavior (Miao et al., 2023; Pappu et al., 2023; Shen et al., 2023).

Firstly, we would like to address that 35S promoter-driven expression of CO was applied not to prove that a similar level of condensation occurs under physiological conditions. Instead, it was designed to understand the underlying mechanism of tuneable molecular assembly of CO/NF-YB/YC complexes for the following reasons, which was also supported with additional functional validation.

- a. The 35S promoter-CO is functional and regulated by known photoperiod switches (Valverde et al., 2004). When 35S promoter-driven CO transgenic plants were exposed to light, we observed that the initially non-existent CO fluorescent signals gradually accumulated (as seen in Extended Data Fig. 1a, b). During this process, the protein level ranged from negligible to high, encompassing all possible physiologically relevant concentrations of CO protein, which supports the functionality of the 35S-driven expression line.
- b. Due to the fluorescent image sensitivity, our imaging used the possible sensitive equipment to follow the initial nucleation/clustering of CO condensates with a spinning disk confocal (Yokogawa CSU-W1) and a super high quantum-yield camera (up to 95% QE, ORCA-Fusion BT Digital CMOS camera). The native promoter condition does not allow us to see both diluted and condensed phases, due to the limited photons for capturing.
- c. To the best of our understanding, no other studies have managed to achieve the level of detection sensitivity in plants that we have accomplished in this research. Notably, the ability to detect the initial nucleation of CO-condensates is paramount, as it reflects the functional states. By using the 35S promoter, we were able to extend the range of assembly states of a functional CO-complex. This enables us to embark on fundamental studies of CO condensation and function, starting from nucleation and clustering, which is the key functional assembly status found from this study.

Secondly, the native CO is specifically expressed in phloem companion cells to promote flowering with a daily rhythm in CO abundance (An et al., 2004; Valverde et al., 2004), which can be challenging to be observed due to the short laser penetration depth at high magnification, especially for those transcription factors with low abundance.

Thirdly, to mimic the low abundance of transcription factor *in vivo*, we have provided further biochemical assays by single particle imaging to analyze the multi-component assembly of CO/NF-YC9/NF-YB2 complex at the nanoscale. The single particle image analysis clearly demonstrates the intermolecular interactions of CO, NF-YB2, and NY-YC9, either bi- or tri-component assembly, start from a nanomolar concentration (Fig. 3b; Extended Data Fig. 6d). Most importantly, it shows the finetuning of CO accumulation by NF-YB and NY-YC, cooperatively, to prevent CO nanocluster over-assembly (Fig. 3c, d; Extended Data Fig. 6e, f). Those data, together with the bulky characterisation assays on CO condensation (Extended Data Fig. 6 g-i), unveil the regulatory role of NF-YB2 and NF-YC9 on CO assembly across a spectrum of protein abundances spanning from nanoscale to macroscale (see pages 10-11), which also supports the robustness of employing the 35S promoter to elucidate the mechanism of tuneable molecular assembly of CO/ NF-YB/YC complexes *in vivo*. It is important to note that phase separation in thermodynamics does not inherently incorporate timing, which is a kinetic concept. Therefore, through concentration-dependent assays, we were able to describe the progressive assembly of biomolecules, thereby addressing kinetics as intended.

Lastly, determining the physiological concentration of the CO proteins in inducing conditions presents a significant challenge. To our knowledge, no such study has been done on transcription factor in plants. Again, the low/high order assembly of CO in transient expression system do not necessarily capture the physiological level of the CO proteins. It was applied to enable robust imaging and quantitative analysis to understand the tuneable molecular assembly of CO/ NF-YB/YC complexes.

In summary, we conducted substantial additional single-molecule image experiments at nanomolar protein concentrations, and provided further explanation on how the phase transition stages we presented allow us to track the progression of nanomolar condensation to understand their assembly strategies and the tuneable material properties. This gives us insights into how the various forms of condensation - from nano- to micro- or meso-scale - align with their functionality and physiological relevance.

Referee #3:

Huang and coworkers investigated CO with NF-YB2/NF-YC9/FT precisely regulates FT transcriptional activation via co-phase separation to control floral transition in Arabidopsis. The authors focus on the functional state of CO condensates in a B-box-motif-dependent manner for FT activation and floral transition, which NF-YC9 and NF-YB2 interact with CO to prevent the overassembly of CO into high-order condensates. While this study is interesting, and the authors provided genetic and biochemical evidences to interrogate the material properties of intracellular condensates, I have several major comments.

The drawback of this manuscript is that only overexpression of CO was used to investigate its condensates. As we know, concentration could be a key factor in protein condensation. Also, CO protein concentration is in dynamic control by light. Under the physiological concentration, does CO functions via condensation? The authors should observe CO condensation in the native promoter-driven transgenic lines.

We thank the reviewer for raising up this concern to help up improve the quality of this study. Please refer to the above response in pages 1-2 for the overall evaluation to the same concern.

Below, we provide our response on how to address this issue within the current stage of technology development, enabling a conceptual contribution to the field of nano-scale condensation studies.

The overarching process of associative molecular condensation starts with percolation clustering, progresses through phase transition at a crucial percolation concentration, and culminates in further associative interactions. In numerous instances, including the multi-component CO condensates in our study, this process can develop beyond the critical saturation concentration of phase separation, exhibiting droplet-like behavior (Miao et al., 2023; Pappu et al., 2023; Shen et al., 2023).

Firstly, we would like to address that 35S promoter-driven expression of CO was applied not to prove that a similar level of condensation occurs under physiological conditions. Instead, it was designed to understand the underlying mechanism of tuneable molecular assembly of CO/NF-YB/YC complexes for the following reasons, which was also supported with additional functional validation.

- a. The 35S promoter-CO is functional and regulated by known photoperiod switches (Valverde et al., 2004). When 35S promoter-driven CO transgenic plants were exposed to light, we observed that the initially non-existent CO fluorescent signals gradually accumulated (as seen in Extended Data Fig. 1a, b). During this process, the protein level ranged from negligible to high, encompassing all possible physiologically relevant concentrations of CO protein, which supports the functionality of the 35S-driven expression line.
- b. Due to the fluorescent image sensitivity, our imaging used the possible sensitive equipment to follow the initial nucleation/clustering of CO condensates with a spinning disk confocal (Yokogawa CSU-W1) and a super high quantum-yield camera (up to 95% QE, ORCA-Fusion BT Digital CMOS camera). The native promoter condition does not allow us to see both diluted and condensed phases, due to the limited photons for capturing.
- c. To the best of our understanding, no other studies have managed to achieve the level of detection sensitivity in plants that we have accomplished in this research. Notably, the ability to detect the initial nucleation of CO-condensates is paramount, as it reflects the functional states. By using the 35S promoter, we were able to extend the range of assembly states of a functional CO-complex. This enables us to embark on fundamental studies of CO condensation and function, starting from nucleation and clustering, which is the key functional assembly status found from this study.

Secondly, the native CO is specifically expressed in phloem companion cells to promote flowering with a daily rhythm in CO abundance (An et al., 2004; Valverde et al., 2004), which can be challenging to be observed due to the short laser penetration depth at high magnification, especially for those transcription factors with low abundance.

Thirdly, to mimic the low abundance of transcription factor *in vivo*, we have provided further biochemical assays by single particle imaging to analyze the multi-component assembly of CO/NF-YC9/NF-YB2 complex at the nanoscale. The single particle image analysis clearly

demonstrates the intermolecular interactions of CO, NF-YB2, and NY-YC9, either bi- or tri-component assembly, start from a nanomolar concentration (Fig. 3b; Extended Data Fig. 6d). Most importantly, it shows the finetuning of CO accumulation by NF-YB and NY-YC, cooperatively, to prevent CO nanocluster over-assembly (Fig. 3c, d; Extended Data Fig. 6e, f). Those data, together with the bulky characterisation assays on CO condensation (Extended Data Fig. 6 g-i), unveil the regulatory role of NF-YB2 and NY-YC9 on CO assembly across a spectrum of protein abundances spanning from nanoscale to macroscale (see pages 10-11), which also supports the robustness of employing the 35S promoter to elucidate the mechanism of tuneable molecular assembly of CO/ NF-YB/YC complexes in vivo. It is important to note that phase separation in thermodynamics does not inherently incorporate timing, which is a kinetic concept. Therefore, through concentration-dependent assays, we were able to describe the progressive assembly of biomolecules, thereby addressing kinetics as intended.

Lastly, determining the physiological concentration of the CO proteins in inducing conditions presents a significant challenge. To our knowledge, no such study has been done on transcription factor in plants. Again, the low/high order assembly of CO in transient expression system do not necessarily capture the physiological level of the CO proteins. It was applied to enable robust imaging and quantitative analysis to understand the tuneable molecular assembly of CO/ NF-YB/YC complexes.

In summary, we conducted substantial additional single-molecule image experiments at nanomolar protein concentrations, and provided further explanation on how the phase transition stages we presented allow us to track the progression of nanomolar condensation to understand their assembly strategies and the tuneable material properties. This gives us insights into how the various forms of condensation - from nano- to micro- or meso-scale - align with their functionality and physiological relevance.

The main findings were overlaid on the basis that CO condensates are functional, but the evidence is lacking. The sole evidence might be deletion of two B-box motifs is required for the assembly of CO condensates. I suggest that mutation(s) that block condensation or replacement of oligomerization domain should be tested.

Thank you for providing us the chance to clarify this important point and elucidate the various experiments that indicate CO's functionality is dependent on an optimal condensation status. As the evidence for CO condensation is required for its functionality, the reviewer suggested mutation(s) that block condensation or replacement of oligomerization domain indeed have been done in another study (Zeng et al., 2022). Zeng's work showed the deletion of two B-box motifs or mutations of key residues mediating B-box oligomerization led to a non-functional CO, while replacing the B-box motifs of CO with a human p53-derived motif (which assemble to a tetramer) partially but not entirely rescue the CO mediated FT activation and flowering phenotype (please see the copied Figure 4 of Zeng et al., 2022 here). These pieces of evidence were intended to demonstrate that B-box-mediated binding is crucial for driving macromolecular assembly, rather than indicating the functional dependence of multicomponent CO-condensates on assembly status. The aim of the molecular condensation-mediated functional regulation in this study was not solely to confirm partnerships for known components or to validate that a critical binding motif can disrupt the assembly and function. Instead, our primary focus was on understanding how each

component controls macromolecular assembly patterns and material properties to facilitate function.

In line with those genetic characterization, here we have added a set of new data where we have conducted mathematical modelling based molecular simulation, which showed monomeric, trimeric, tetrameric and pentameric CO/NF complexes displayed progressively enhanced binding affinity on *FT* promoter (Fig. 1e; Extended Data Fig. 1 d, e). Together, those data underscore the fundamental principle of how local accumulation of CO regulates its functionality on *FT* activation. We have emphasized those points in discussion in the revised manuscript (page 15). Our results take a significant step further by providing important molecular insights built upon previously established foundations.

Figure 4. Multimeric assembly of the CONSTANS (CO) protein through N-terminal B-box domains is critical for CO function
(A) Schematic representation of CO structure and CO transgenes. p53_{TD}, p53 tetramerization domain. **(B)** Flowering time of the indicated transgenic plants grown in long days (LDs). Twelve plants per line were scored, and error bars indicate SD. **(C)** Functional analysis of the two B-box domains in the N terminus of CO. Total leaf number at flowering of the T₁ transgenic plants of *co-9* expressing the indicated CO transgenes were scored. Plants were grown in LDs; 12 for each line were scored, and error bars for SD.

It should be noted that Zeng's work has determined the CO B-box motifs assemble to a continuous head-to-tail tandem oligomer structurally. Only when being tagged with solubility enhancer tag (MBP tag was applied in Zeng's work), it tends to form trimer to tetramer (which is the reason that tetramer motif was applied to replace the B-boxes for validation). That information actually greatly supports our observation that CO by itself tend to over-assemble to high-order condensate, which could be finetuned by the addition of NF-YC9/NF-YB2 to reach an optimal condensation for *FT* activation (Fig. 2a; Extended Data Fig. 3).

In the transient expression assay, the transfection efficiency varies a lot, resulting in varied concentrations of CO protein, therefore the appearance of three populations could be resulted from concentration variation. This data does not prove that under physiological condition, CO exists in these states.

The response to this point is encompassed in the above reply regarding the 35S promoter. We agree with the reviewer that the different population of CO assembly might be resulted from concentration variation, we therefore have provided very detailed quantitative analysis of how the addition of NF-YB2, NF-YC9 or both would shift the population of each type of CO condensation and change the physical properties of CO condensation (Fig. 2; Extended Data Fig. 2-4). The low/high order assembly of CO in transient expression system do not necessarily

capture the physiological level of the CO proteins. It was applied to enable robust imaging and quantitative analysis to understand the mechanism of tuneable molecular assembly of CO/NF-YB/NF-YC complexes.

The authors found that the dynamicity of CO condensates differed with and without NF-YC9. The data in Figure 1e, to me, is comparing condensed phase to diffused phase, of course the diffused phase has higher recovery rate.

Unless the authors could specifically bleach the small condensates. Since that YC9 tunes the material property is the major point of this manuscript, the authors must improve the resolution of imaging to more carefully compare the dynamicity of condensed phases in big and small condensates.

The interdisciplinary nature of phase separation research arises from the interdisciplinary techniques required to dissect the underlying mechanisms of molecular assembly progression, spanning from nucleation to growth across nano- to micrometer scales. Traditional IP or Western blot assays are incompatible with studying weak multivalent interactions, and each technique has its limitations within specific sensitivity ranges for deciphering biomolecular function, from single molecules to tissue levels. Therefore, conventional diffusion dynamic studies in phase separation, typically used for detecting micromolecular bright signals, are impractical for analyzing weak multivalent nano-scale assembly in this study. To overcome this challenge, highly sensitive detection methods beyond current technological developments are necessary. As a result, we have endeavoured to provide a broad spectrum of molecular interactions and biochemical reactions to understand how would such process evolve, providing scientific insights and contributions using the most sensitive systems and quantitative methods available, aiming to advance knowledge in the field.

We have tried to apply probably one of the most sensitive systems currently available for fluorescence imaging in plant cells, which couples spinning disk confocal with the sensitive sCOMS camera (Hamamatsu) with a quantum efficiency of 95%. The looks diffused signal in the bleached region is because the low-order condensation status and low quantum yield of mCherry, as low laser power and short exposure time were set to minimize photobleaching for time-series imaging. As evidence, the clear mCherry-CO condensations could be captured in the still images with longer exposure time and higher laser power (Fig. 1c, d). To improve the image quality, we have performed Huygens deconvolution on those time-series images and revised the presentation (Fig. 1f). The deconvoluted image clearly showed low-order of CO condensations were presented in the bleached area and the recoverable CO condensation in *mCherry-CO*, *mCherry-CO NF-YC9-GFP ycT* but non-recoverable CO condensation in *mCherry-CO ycT* (Fig. 1f).

Also in Figures 2d, 3b, S10d, immediately after bleaching, the overall signal is reduced, which does suggest that the CO molecules are quite mobile. The inability of the bleached area to recover could be because the bleaching area is too large such that a large population of GFP-CO molecules are switched off and the amount available for exchange is too small. The authors should specifically bleach each condensate.

We have added one panel of the Extended Data Fig. 4b to show the high contrast images of Fig. 2c and Extended Data Fig. 4a (originally all shown in Fig. 2d). Those high contrast images enable the visualization of the diffused GFP-CO signal. The exchange of those mobile GFP-CO signal from unbleached regions to bleached regions happens in second scale, mostly reaching plateau at >20s. This indicates the overall signal reduction immediately after photobleaching, happening in millisecond scale, was not caused by the fast diffusion of GFP-CO molecules but by the overall signal bleaching, which is usually happened in FRAP experiment. Indeed, when we quantify the recovery rate, to subtract the overall signal bleaching during time-series image acquisition, the intensity of an area with the same size in unbleached region was measured to normalize the signal bleaching (please see the Methods section page 19).

It is not possible to target each individual condensate. This is because these condensates have diameters smaller than the diffraction limit and are surrounded by each other in high-density. To the best of our knowledge, there is no existing microscope capable of selectively bleaching individual condensates amidst such densely packed clusters. However, it is important to note that the photobleaching of multiple clusters on the same time does not compromise our conclusions regarding molecular diffusion rates. At such nanoscale condensation, the FRAP is based on the molecule exchanged between condensed and diluted phase, which are not affected by the bleaching of single or multiple clusters. This is also revealed in the newly added Extended Data Fig. 4b, where it shows clear punctate cluster signal in the bleached regions of fluidic CO condensation but not clear in the slow-diffusive CO condensation.

To test whether altered material properties influence CO function, the authors used the GUS reporter, which is helpful, but I think the binding of target DNA, either *in vitro* or *in vivo* using ChIP, is very important.

We appreciate the reviewer's insightful query. As we interpret it, the reviewer is interested in knowing whether different material properties of CO condensation could influence its binding to the *FT* promoter.

Our *in vitro* molecular condensation experiments involving the CO-NFYB2/YC9 complex, both with and without the *FT*-promoter, provide direct evidence of CO-condensate recognition of the *FT* promoter. These experiments, which also included a non-CO target DNA fragment as a control, are now depicted in Fig. 3e, f and Extended Data Fig. 6j, k.

At the current stage, there is a lack of practical approach allowing an isolation of nuclear transcriptional factor condensate and conduct following ChIP assay. We appreciate the reviewer highlights the importance of this in the field. It is beyond the capability we can achieve and the scope of this manuscript. Nevertheless, a development of such technique would definitely make significant contribution in the field studying condensation function in transcriptional factor condensates.

Minor points

1. In lines 748, 836, and legend of Figures 3, S6, "20HEPES" should be "20mM HEPES"?
2. FRAP buffer described in the method is inconsistent with that described in the legend.
3. Line 509: "MgCl" should be "MgCl₂"

Our apologies for these errors. We have now revised them. For the point 2, the FRAP buffer described in the method has been corrected as “20 mM HEPES, 150 mM NaCl, pH 7.4” to be consistent with the legend.

References

- An, H., Roussot, C., Suárez-López, P., Corbesier, L., Vincent, C., Piñeiro, M., Hepworth, S., Mouradov, A., Justin, S., Turnbull, C., *et al.* (2004). CONSTANS acts in the phloem to regulate a systemic signal that induces photoperiodic flowering of *Arabidopsis*. *Development* *131*, 3615-3626.
- Anunziata, O., Asherie, N., Lomakin, A., Pande, J., Ogun, O., and Benedek, G.B. (2002). Effect of polyethylene glycol on the liquid-liquid phase transition in aqueous protein solutions. *Proc Natl Acad Sci U S A* *99*, 14165-14170.
- Banani, S.F., Lee, H.O., Hyman, A.A., and Rosen, M.K. (2017). Biomolecular condensates: organizers of cellular biochemistry. *Nat Rev Mol Cell Biol* *18*, 285-298.
- Delarue, M., Brittingham, G.P., Pfeffer, S., Surovtsev, I.V., Pinglay, S., Kennedy, K.J., Schaffer, M., Gutierrez, J.I., Sang, D., Poterewicz, G., *et al.* (2018). mTORC1 Controls Phase Separation and the Biophysical Properties of the Cytoplasm by Tuning Crowding. *Cell* *174*, 338-349.e320.
- Frontini, M., Imbriano, C., Manni, I., and Mantovani, R. (2004). Cell cycle regulation of NF-YC nuclear localization. *Cell Cycle* *3*, 217-222.
- Gnesutta, N., Kumimoto, R.W., Swain, S., Chiara, M., Siriwardana, C., Horner, D.S., Holt, B.F., 3rd, and Mantovani, R. (2017). CONSTANS Imparts DNA Sequence Specificity to the Histone Fold NF-YB/NF-YC Dimer. *Plant Cell* *29*, 1516-1532.
- González, J.M., Jiménez, M., Vélez, M., Mingorance, J., Andreu, J.M., Vicente, M., and Rivas, G. (2003). Essential cell division protein FtsZ assembles into one monomer-thick ribbons under conditions resembling the crowded intracellular environment. *J Biol Chem* *278*, 37664-37671.
- Hackenberg, D., Wu, Y., Voigt, A., Adams, R., Schramm, P., and Grimm, B. (2012). Studies on Differential Nuclear Translocation Mechanism and Assembly of the Three Subunits of the *Arabidopsis thaliana* Transcription Factor NF-Y. *Molecular Plant* *5*, 876-888.
- Hou, X., Zhou, J., Liu, C., Liu, L., Shen, L., and Yu, H. (2014). Nuclear factor Y-mediated H3K27me3 demethylation of the SOC1 locus orchestrates flowering responses of *Arabidopsis*. *Nat Commun* *5*, 4601.
- Kahle, J., Baake, M., Doenecke, D., and Albig, W. (2005). Subunits of the heterotrimeric transcription factor NF-Y are imported into the nucleus by distinct pathways involving importin beta and importin 13. *Mol Cell Biol* *25*, 5339-5354.
- Kumimoto, R.W., Adam, L., Hymus, G.J., Repetti, P.P., Reuber, T.L., Marion, C.M., Hempel, F.D., and Ratcliffe, O.J. (2008). The Nuclear Factor Y subunits NF-YB2 and NF-YB3 play additive roles in the promotion of flowering by inductive long-day photoperiods in *Arabidopsis*. *Planta* *228*, 709-723.
- Kumimoto, R.W., Zhang, Y., Siefers, N., and Holt III, B.F. (2010). NF-YC3, NF-YC4 and NF-YC9 are required for CONSTANS-mediated, photoperiod-dependent flowering in *Arabidopsis thaliana*. *The Plant Journal* *63*, 379-391.
- Lodge, T.P., Maxwell, A.L., Lott, J.R., Schmidt, P.W., McAllister, J.W., Morozova, S., Bates, F.S., Li, Y., and Sammler, R.L. (2018). Gelation, Phase Separation, and Fibril Formation in Aqueous Hydroxypropylmethylcellulose Solutions. *Biomacromolecules* *19*, 816-824.

Lv, X., Zeng, X., Hu, H., Chen, L., Zhang, F., Liu, R., Liu, Y., Zhou, X., Wang, C., Wu, Z., *et al.* (2021). Structural insights into the multivalent binding of the Arabidopsis FLOWERING LOCUS T promoter by the CO-NF-Y master transcription factor complex. *Plant Cell* **33**, 1182-1195.

Miao, Y., Guo, X., Zhu, K., and Zhao, W. (2023). Biomolecular condensates tunes immune signaling at the Host-Pathogen interface. *Curr Opin Plant Biol* **74**, 102374.

Nardini, M., Gnesutta, N., Donati, G., Gatta, R., Forni, C., Fossati, A., Vonrhein, C., Moras, D., Romier, C., Bolognesi, M., *et al.* (2013). Sequence-specific transcription factor NF-Y displays histone-like DNA binding and H2B-like ubiquitination. *Cell* **152**, 132-143.

Pappu, R.V., Cohen, S.R., Dar, F., Farag, M., and Kar, M. (2023). Phase Transitions of Associative Biomacromolecules. *Chem Rev* **123**, 8945-8987.

Peskett, T.R., Rau, F., O'Driscoll, J., Patani, R., Lowe, A.R., and Saibil, H.R. (2018). A Liquid to Solid Phase Transition Underlying Pathological Huntingtin Exon1 Aggregation. *Mol Cell* **70**, 588-601.e586.

Samach, A., Onouchi, H., Gold, S.E., Ditta, G.S., Schwarz-Sommer, Z., Yanofsky, M.F., and Coupland, G. (2000). Distinct roles of CONSTANS target genes in reproductive development of Arabidopsis. *Science* **288**, 1613-1616.

Shen, Z., Jia, B., Xu, Y., Wessén, J., Pal, T., Chan, H.S., Du, S., and Zhang, M. (2023). Biological condensates form percolated networks with molecular motion properties distinctly different from dilute solutions. *Elife* **12**.

Valverde, F., Mouradov, A., Soppe, W., Ravenscroft, D., Samach, A., and Coupland, G. (2004). Photoreceptor regulation of CONSTANS protein in photoperiodic flowering. *Science* **303**, 1003-1006.

Xie, Y., Sun, J., Han, X., Turšić-Wunder, A., Toh, J.D.W., Hong, W., Gao, Y.G., and Miao, Y. (2019). Polarisome scaffolder Spa2-mediated macromolecular condensation of Aip5 for actin polymerization. *Nat Commun* **10**, 5078.

Zeng, X., Lv, X., Liu, R., He, H., Liang, S., Chen, L., Zhang, F., Chen, L., He, Y., and Du, J. (2022). Molecular basis of CONSTANS oligomerization in FLOWERING LOCUS T activation. *J Integr Plant Biol* **64**, 731-740.

Dear Dr Miao,

Thank you for submitting a revised version of your manuscript and for the productive discussions regarding the concerns raised by the referees during re-review. We have now carefully considered the arguments raised in your point-by-point reply and have decided that these are reasonable and sufficiently address all the remaining concerns. There remain only a few mainly editorial points that have to be addressed before I can extend formal acceptance of the manuscript:

- Please convert the ms file into a .docx format, and remove the figures from the ms file, please also make sure that there are no track changes left.
- Please make sure that all authors are included in EJP (Tiedong Sun and Lanyuan Lu)
- Please double-check to make sure to all relevant funding information in the manuscript is also entered into our submission system.
- On the abstract page of the manuscript, please include 4-5 general keyword terms to enhance searchability.
- Please rename the Conflict of Interest section into "Disclosure and Competing Interests Statement", in accordance with our updated Guide to Authors
- As we are switching from a free-text author contribution statement towards a more formal statement based on Contributor Role Taxonomy (CRediT) terms, please remove the present Author Contribution section and instead specify each author's contribution(s) directly in the Author Information page of our submission system during upload of the final manuscript. See <https://casrai.org/credit/> for more information.
- There is a reference to "data not shown" Figure EV1D legend and in Results. According to our policy, which does not permit references to "data not shown", please include this information in the manuscript. Please see also <https://www.embopress.org/page/journal/14602075/authorguide#unpublisheddata>.
- Please adjust the in-text callouts for individual figures and figure panels: Supplementary Table 1 and Supplementary Table 2 called out in ms, but no tables uploaded
- Please provide either a "Yes" or a "Not Applicable" answer to each one of the questions in your Author Checklist (<https://www.embopress.org/pb-assets/embo-site/EMBO%20Press%20Author%20Checklist-1642513524327.xlsx>). In the last column of this checklist, only the sections of the manuscript where the relevant information can be found should be listed (the information per se should be included in the main manuscript file).

Please upload all figures individual, high-resolution figure files with legends moved below the References

- Since we can only accommodate up to 5 Expanded View Figures that will be directly included in the HTML version, some of the currently 12 EV Figures have to be turned into "Appendix Figures". Appendix Figures should be uploaded in a single APPENDIX PDF, headed by a brief Table Of Contents with page numbers of the included items. The nomenclature should be throughout the Appendix PDF and manuscript file: Appendix Figure Sx with the appropriate callouts
- Please provide the Reagent and Tools Table. For more information, please check <https://www.embopress.org/page/journal/14602075/authorguide#structuredmethods> and download the template for Reagent Table (https://www.embopress.org/pb%2Dassets/embo-site/Reagents_Tools_Table_TEMPLATE.docx)
- Please make sure to provide all the requested Source data files listed in the uploaded and attached Source Data checklist file, which will be sent by my colleague Hannah Sonntag. Please complete the Source Data checklist and upload it to our online system. Source data files need to be saved in a scheme one figure/folder and then uploaded as .zip files. E.g. all the Source data files for figure 1 need to be saved in a single folder and this needs to be zipped and then uploaded as "SD figure 1.zip" file.
- Please provide suggestions for a short 'blurb' text prefacing and summing up the conceptual aspect of the study in two sentences (max. 250 characters), followed by 3-5 one-sentence 'bullet points' with brief factual statements of key results of the paper; they will form the basis of an editor-written 'Synopsis' accompanying the online version of the article. Please also provide an altered synopsis image, making sure that the aspect ratio conforms to our website's format - it should be exactly 550 pixels wide and between 300-600 pixels high.
- Please provide exact p values in the legends of figures 1b, h; 2f; 3d; 4d-f; extended data figures 1b; 9c-d, f; 11b.
- Please provide the information related to n in the legends of figures 1h; 2f; 4e-f, extended data figure 9d.

- Although 'n' is provided, please describe the nature of entity for 'n' in the legends of figures 2e; 4d, extended data figures 1b; 9c; 11b.
- Please define the error bars in the legends of figures 1h; 2f; 4e-f, extended data figures 1b; 9d; 11b.
- Please define the scale bar for extended data figures 4a-b.
- Please add and define a scale bar for extended data figure 2c.
- Please define the yellow dashed circles in the legend of extended data figure 2c, extended data figures 4b; 6a, g, j.
- Please label the axis gaps appropriately in figure 3f; extended data figures 6h-i, k.
- Please adjust the order of the manuscript sections: Title page with complete author information, Abstract, Keywords, Introduction, Results, Discussion, Methods, Data Availability Section, Acknowledgements, Disclosure and Competing Interests Statement, References, Main figure legends, Tables, Expanded Figure Legends.
- Please check the first and last cell image in Figure Extended Data 6E - It appears the cell image is repeated.

With best regards,

Cornelius Schneider

Cornelius Schneider, PhD
 Editor | The EMBO Journal
 c.schneider@embojournal.org

We realize that it is difficult to revise to a specific deadline. In the interest of protecting the conceptual advance provided by the work, we recommend a revision within 3 months (16th Dec 2024). Please discuss the revision progress ahead of this time with the editor if you require more time to complete the revisions. Use the link below to submit your revision:

Referee #1:

The authors have thoroughly addressed the previous concerns raised, providing a comprehensive rebuttal supported by new data. Their efforts are commendable, as they have clarified and strengthened their model with convincing and solid evidence. That said, I noticed that not all important points necessary to clarify confusion are included in the revised manuscript. I suggest the authors incorporate these important points to facilitate readers' understanding.

Overall, this work presents a novel and intriguing perspective on the role of phase separation in fine-tuning key flowering regulators, thereby controlling plant flowering. It is a valuable contribution to the field, and I highly recommend its publication after incorporating minor revisions to address the remaining points.

Referee #2:

In the revised version of this manuscript, the Authors Xiang Wang and coworkers implemented their original findings with improved images of FRAP experiments and with requested controls addressing some of the major criticisms. They included novel data at nanoscale concentrations employing single particle imaging techniques and in silico modeling with DNA binding simulations of CO oligomers to support the relevance of multimolecular assemblies in the functional mechanisms of CO regulated FT expression.

While the data and text revisions have improved the overall quality and clarity of the manuscript, I still feel that the current evidence fall short in supporting the principal manuscript claim i.e. that fine-tuning of phase separation events are at the basis of the floral transition gating mediated by the CO/NF-YB/NF-YC (i.e. NF-CO) complex.

One major concern was referred to the overexpression system employed to visualise and characterize the state of matter of CO protein in living cells, which clearly led to the formation of (non-functional) high order aggregates of CO.

While the overexpression of the YB/YC proteins (or complementation of ycT with 35S driven YC) leads to substantially reduced "overassembly" of CO, in my view it is yet to be determined which of the other physical states described represents the functional CO(NF-YB/YC) complex: either diffuse or spherical condensate. The last is further subdivided into the "liquid" or "slow-diffusive" subtypes, as the diffuse state is considered non relevant to the functional role of CO by the Authors. As I may agree that the slow-diffusive state may not represent a functional asset of the separated CO, one may argue that the diffuse fraction of CO may be sufficient, in the presence of functional YB/YC, for promoting the inducing conditions.

Exclusion of a functional role of the non-condensed state of CO in the formation of functional heterotrimeric complexes is deduced from the finding that B-box deletion not only impedes CO clustering into either form of condensate, but also impairs its function, resulting in diffuse distribution of CO. It has to be noted that if oligomerization potential (mediated by the Bbox domains) would be necessary for CO function in transcriptional activation, such property does not imply that the protein may ever reach in vivo the concentrations that promote the phase transition into the observed (liquid-type) condensates, and be fully functional in a diffused (oligomeric) state.

Regarding this, accumulation of CO after light exposure into clusters, as reported to prove the functionality of the 35S promoter, does not indicate which is the concentration threshold that would allow the transcriptional activity on the FT promoter, or necessarily imply that it coincides with the physical state transitions of CO.

Regarding the mutagenesis experiments, on the NF-Y side, one may also argue that deletion of the sole Glu-rich portion of YC9 (otherwise positioned at the C-term of the HFD in other homologs, and possibly represented by an acidic domain in yeast (see alignments in extended fig 12) not only reduces the fluidity of CO condensates, but also deprives the complex of a Transcriptional Activation domain.

Single particle analysis of purified proteins mixtures at nanomolar concentrations does not help understanding of the functional

state of the assemblies, as colocalized proteins seem to be present in a minority of the total particles observed (fig 3b ext data6d), with similar shapes of other particles and variable intensities of each of the components. Titration experiments with YB in the presence of YC seem to reduce the size of CO puncta rather than increasing the size of separated clusters.

As stated in the text, purified protein assemblies which are observed by FRAP at μM concentrations do not display any dynamic property, while FT DNA addition does not significantly increase fluorescence recovery, which shows rather unstable signal intensities, supporting the fact that large condensates are still visibly formed (fig 3f), despite improvement of the quality of images displayed.

The molecular exchange between separated and diffusive liquid compartments is not addressed by the SPR technique, as such approach measures bi-(tri-)molecular interactions on a solid surface that does not recapitulate the condensed state of matter. Furthermore, regarding evaluating the established assembly of a functional NF-CO (-Hd1) trimer with a dimeric NF-YB/YC scaffold and its relationship with CO phase separation properties, the Authors still don't consider the experimental setup of the addition of equimolar amounts of the NF-Y HFD subunits in titration experiments of CO condensates, nor the appropriate opposite configuration, where increasing amounts of CO are added to soluble (or immobilised) heterodimers.

For this, in figure 2 - and in all figures indicating the proportion of liquid/slow-diffusive condensates, e.g. fig 4 examining the YC9 polyQ mutants, extended data fig. 8, 9, 10 for delta IDR mutagenesis- the Authors should have indicated the % of nuclei showing the diffuse state, so the reader may better appreciate the distinction between all the different states and their relative proportion within each experimental setup (as in fig 2B first manuscript version, which was replaced by fig 2A with relative frequency distribution), and directly evaluate whether any changes in such diffuse population may or may not be related to the functional state of the assemblies.

Regarding the newly added data on the simulation of DNA binding of modeled NF-Y-CO mono/oligomeric complexes on the FT promoter (why disordered regions of full-length NF-Y proteins were not included?), it provides some indication of increased DNA association properties of NF-CO oligomers, but it is still to be explained whether such increased association occurs on different sites of the same -or of different- DNA molecules, as they do not address the relevant point of cooperative oligomeric assembly that might occur on the FT promoter region, possibly promoting the functional (phase separated or simply oligomeric?) state of the complex.

Referee #3:

The authors performed additional experiments and explanations that have addressed most of my concerns except the following:

1. In the Introduction section, Fang et al., 2019 should also be referred in the sentence "RNA processing of FLOWERING LOCUS C and its antisense transcript (COOLAIR), the major target genes in the vernalization pathway, are regulated by phase-separated regulatory condensates"

2. NF-YC9-GFPs were diffusely distributed in the nuclei of 35S:NF-YC9-GFP *ycT* cells as shown in Extended Data Fig. 1c, but NF-YC9-GFP assembled into smaller foci in 35S:mCherry-CO 35S:NF-YC9-GFP *ycT* plants (Fig. 1b), arguing that endogenous CO does not form condensates and that the condensates observed are resulted from CO overexpression. The percolation of CO, if any, was below optical diffraction limit. The authors should be cautious assuming the small foci being the percolation clusters.

3. Regarding the computer simulation, the authors should explain more clearly what the CO/FT association probability means in Fig. 1e. The authors stated in the Methods that "the binding statistics were performed by counting the number of molecular contacts between the CCT-NF-Y complex and DNA/COREs", is the number of molecular contacts for each CCT-NF-Y complex or an oligomer? if oligomerization increases the contacts for each CCT-NF-Y complex, what's the underlying molecular basis, conformational change?

Referee #1 (Report for Author)

The authors have thoroughly addressed the previous concerns raised, providing a comprehensive rebuttal supported by new data. Their efforts are commendable, as they have clarified and strengthened their model with convincing and solid evidence.

That said, I noticed that not all important points necessary to clarify confusion are included in the revised manuscript. I suggest the authors incorporate these important points to facilitate readers' understanding.

Overall, this work presents a novel and intriguing perspective on the role of phase separation in fine-tuning key flowering regulators, thereby controlling plant flowering. It is a valuable contribution to the field, and I highly recommend its publication after incorporating minor revisions to address the remaining points.

We appreciate the reviewer's suggestion and positive evaluation. We have carefully checked the Referee #1's previous comments and incorporate missed points into the rerevised manuscript for clarification. Please see discussion on pages 15-16.

Referee #2 (Report for Author)

In the revised version of this manuscript, the Authors Xiang Wang and coworkers implemented their original findings with improved images of FRAP experiments and with requested controls addressing some of the major criticisms. They included novel data at nanoscale concentrations employing single particle imaging techniques and in silico modeling with DNA binding simulations of CO oligomers to support the relevance of multimolecular assemblies in the functional mechanisms of CO regulated FT expression. While the data and text revisions have improved the overall quality and clarity of the manuscript, I still feel that the current evidence fall short in supporting the principal manuscript claim i.e. that fine-tuning of phase separation events are at the basis of the floral transition gating mediated by the CO/NF-YB/NF-YC (i.e. NF-CO) complex.

One major concern was referred to the overexpression system employed to visualise and characterize the state of matter of CO protein in living cells, which clearly led to the formation of (non-functional) high order aggregates of CO.

We thank the reviewer for agreeing with one of our key conclusions that over-assembled high order CO aggregates were non-functional. Here, we want to emphasize that such non-functional CO aggregation was only shown in *ycT* mutant (Fig. 1g, organ line) but not in the presence of YC9-GFP (Fig. 1g, cyan and black lines). Plants expressing these 35S driven fusions in YC9 WT lines do not show defects in growth and flowering (Fig. 1a). This is consistent with transient expression assay where CO expression alone displayed mostly irreversible aggregates while CO/NF-Y co-expression shifted CO assembly to mostly

spherical condensation with liquid property (Fig. 2a-e), which upregulated the FT transcription (Fig. 2f).

While the overexpression of the YB/YC proteins (or complementation of *ycT* with 35S driven YC) leads to substantially reduced "overassembly" of CO, in my view it is yet to be determined which of the other physical states described represents the functional CO (NF-YB/YC) complex: either diffuse or spherical condensate. The last is further subdivided into the "liquid" or "slow-diffusive" subtypes, as the diffuse state is considered non relevant to the functional role of CO by the Authors. As I may agree that the slow-diffusive state may not represent a functional asset of the separated CO, one may argue that the diffuse fraction of CO may be sufficient, in the presence of functional YB/YC, for promoting the inducing conditions.

Exclusion of a functional role of the non-condensed state of CO in the formation of functional heterotrimeric complexes is deduced from the finding that B-box deletion not only impedes CO clustering into either form of condensate, but also impairs its function, resulting in diffuse distribution of CO. It has to be noted that if oligomerization potential (mediated by the Bbox domains) would be necessary for CO function in transcriptional activation, such property does not imply that the protein may ever reach in vivo the concentrations that promote the phase transition into the observed (liquid-type) condensates, and be fully functional in a diffused (oligomeric) state.

Regarding this, accumulation of CO after light exposure into clusters, as reported to prove the functionality of the 35S promoter, does not indicate which is the concentration threshold that would allow the transcriptional activity on the FT promoter, or necessarily imply that it coincides with the physical state transitions of CO.

We thank the reviewer for raising up this concern, which would guide us to have a more insightful discussion in the manuscript. Zeng et al's work in 2022 demonstrated that deleting B-box motifs or mutating key residues essential for B-box oligomerization rendered CO non-functional. However, substituting CO's B-box motifs with a human p53-derived motif (which assembles into a tetramer) only partially rescued CO-mediated FT activation and the flowering phenotype (Zeng, Lv et al., 2022). The results suggest that a tetrameric form of CO with an altered B-box is not fully functional. Given that the B-box plays roles in transcriptional activation beyond just promoting CO oligomerization, we agree with the reviewer's point that a simple B-box deletion supports but does not fully validate the function driven by oligomerization. With a same consideration, to address this, we have employed additional approaches in the last revision, such as simulations, which indicate that CO oligomerization indeed enhances binding probability to the FT promoter.

We agree with the reviewer's point that there is no evidence to suggest a complete dysfunction of the diffuse fraction of CO, and we did not make such a claim. The diffuse fraction of CO in vivo might be functional, such as in a resting state with low oligomeric CO, making it extremely difficult to quantify. However, determining the stoichiometry of in

vivo CO assembly and nuclear condensation for FT transcriptional activation remains a significant challenge. Instead, we emphasize the importance of low-order oligomers in enhancing transcriptional activation. Precisely characterizing the oligomeric state of CO under physiological conditions requires accurate *in vivo* single-particle stoichiometry, which is a complex but crucial issue that many researchers in this field are striving to address. A recent application of SlimVar single molecule tracking microscope in plant cells could be an excellent solution in the near future to understand the *in vivo* single particle stoichiometry and the functional species of multi-component assembly of transcription factors, especially in deep tissue cells (Jang, Payne-Dwyer et al., 2024, Payne-Dwyer, Jang et al., 2024), which would be a future direction following this study in CO.

Nevertheless, it does not change our major conclusion that CO by itself tend to be over-assembled to be non-functional, and low-oligomeric CO clusters provided enhanced activity than the microscopic diffusive CO population. CO needs the NF-Y components that synergistically optimize the CO assembly to a proper assembly state and provide the conformation flexibility to activate the FT transcription. That is the key information we have provided from our results comparing with previous known knowledge that CO, NF-YC and NF-YB are both needed for FT transcription activation. In response to the reviewer's comments, we have carefully clarified these claims to prevent any potential for misunderstanding or overstatement. Additionally, we have expanded our discussion on the limitations of our study and outlined future research directions to enhance understanding of CO condensation. These revisions can be found on pages 15-16 of the updated manuscript.

Regarding the mutagenesis experiments, on the NF-Y side, one may also argue that deletion of the sole Glu-rich portion of YC9 (otherwise positioned at the C-term of the HFD in other homologs, and possibly represented by an acidic domain in yeast (see alignments in extended fig 12) not only reduces the fluidity of CO condensates, but also deprives the complex of a Transcriptional Activation domain.

We would like to clarify a misunderstanding here. We were trying to delete or add the Glutamine (Gln, Q) residue in NF-YC9 but not the Glutamic acid (Glu, E), of which the latter (E) is the one contributing to acidic transcriptional activation domain. Glutamine (Q) is a charge-neutral residue, it would theoretically not affect the acidic transcriptional activation domain. As another piece of evidence, we have now performed complex structure prediction for CO/NF-YC9/NF-YB2/ DNA (CORE2) with original NF-YC9 or NF-YC9 (0Q/37Q) through AlphaFold3 (Abramson, Adler et al., 2024). The polyQ region is embedded in α -helices, which is shorten or extended in NF-YC9 (0Q) or NF-YC9 (37Q), respectively. This α -helices does not show clear interaction or close conformation with the predicted transcriptional activation domain of CO (Erijman, Kozlowski et al., 2020). The new data was now adding to the Extended Data Fig. 10, page 13 in the revised manuscript as a clarification.

CO + NF-YC9 + NF-YB2 + DNA (CORE2)

Rebuttal Figure 1. Structure prediction of CO/NY-Y/DNA (CORE2) complex with original NF-YC9, NF-YC9 (0Q) or NF-YC9 (37Q). The DNA fragment input contains the CORE2 site for CO/NF-Y binding. Zoomed regions show the polyQ region embedded a-helices, the polyQ repeats were highlighted with the black box. Red arrow heads indicate the predicted acidic transcriptional activation domain of CO.

Single particle analysis of purified proteins mixtures at nanomolar concentrations does not help understanding of the functional state of the assemblies, as colocalized proteins seem to be present in a minority of the total particles observed (fig 3b ext data6d), with similar shapes of other particles and variable intensities of each of the components. Titration experiments with YB in the presence of YC seem to reduce the size of CO puncti rather than increasing the size of separated clusters.

We thank the reviewer for raising this point allowing us to elaborate more on the multistage of phase separation. Single particle reconstitution allows us to recapture the initial stage of multi-component assembly. The random interaction of single molecule is the nature of phase separation which starts from a kinetic unstable interaction at nucleation stage and propagates to a kinetic stable stage with multi-component assembly. The variable intensities of each of the components imply different stoichiometry of the three components, that is consistent with the feature of tuneable stoichiometry at the initial stage of phase separation. Now, we have added a better interpretation on this point at page 11 in the revised manuscript.

An addition of YB in the presence of YC (but not in absence of YC, Extended Fig. 6e) reduces CO assembly size. This is consistent with the *in vivo* data in Fig. 2a (CO/YC9 vs. CO/YC9/YB2). Collectively suggesting the synergies effect of YB and YC in fine-tuning CO assembly. The reviewer mentioned YB increase the size of separated clusters could be happened in a high concentration like micromolar scale, where YB then act as a

scaffolder to regulate the CO/YC9 condensation. Now, the previous ambiguities were addressed by such clarifications and additional discussions in the revised manuscript.

As stated in the text, purified protein assemblies which are observed by FRAP at μM concentrations do not display any dynamic property, while FT DNA addition does not significantly increase fluorescence recovery, which shows rather unstable signal intensities, supporting the fact that large condensates are still visibly formed (fig 3f), despite improvement of the quality of images displayed.

The unstable signal intensity fluctuation could be because of the DNA engagement into CO/NF-Y complex varies a lot as CO/NF-Y complex by itself displayed relatively very low flexibility. In addition, we applied low laser power and exposure time to do the time-lapse image to avoid overall photobleaching during image acquisition as long-term image (8 min after bleaching) was conducted to record the signal recovery, which could be another reason causing signal variation. To address this point, we have now provided statistical analysis of the signal recovery plateau of CO-mCherry, which shows CO-mCherry has significant higher recovery plateau in + FT versus + DNA control (Fig. 3g). We also quantified the size of CO-mCherry condensation by analyzing the total intensity of CO-mCherry condensates, which did not show significant difference in the samples of + FT versus + DNA control (Extended Data Fig. 6j), pages 11-12.

The molecular exchange between separated and diffusive liquid compartments is not addressed by the SPR technique, as such approach measures bi-(tri-)molecular interactions on a solid surface that does not recapitulate the condensed state of matter.

We agree with the reviewer that SPR assay cannot determine the molecular exchange between separated and diffusive liquid compartments, which was measured by FRAP but not SPR in this manuscript. If we are understanding correctly, the reviewer is concerning on the claim on Line 447-449. We apologize for the confusion caused here, we have revised this sentence to avoid any misleading.

Original statement: 'our SPR and in vitro reconstitution assays indicate NF-YC and NF-YB cooperatively regulate the material properties of the CO condensate to maintain its functionality in a balanced manner.'

Revised statement on page 17: 'our SPR and in vitro reconstitution assays at nanomolar scale indicate NF-YC and NF-YB cooperatively regulate a proper assembly of the CO condensate to maintain its functionality in a balanced manner.'

Furthermore, regarding evaluating the established assembly of a functional NF-CO (-Hd1) trimer with a dimeric NF-YB/YC scaffold and its relationship with CO phase separation properties, the Authors still don't consider the experimental setup of the addition of equimolar amounts of the NF-Y HFD subunits in titration experiments of CO condensates, nor the appropriate opposite configuration, where increasing amounts of CO are added to soluble (or immobilised) heterodimers.

We thank the reviewer for pointing this out, which allows us to elaborate and further clarify. CO-CCT interacting with NF-Y HFD domain and binding with DNA is well-known and has been structurally resolved. Such solid interaction is the basis of the CO/NF-Y tri-component complex assembly. However, the regulation of the complex condensation assembly and material properties are mostly through the unstructured regions located at both N-terminal or C-terminal regions flanking HFD domain (Extended Data Fig. 7 and 12). This is evidenced in Extended Data Fig. 8b and 8c, Extended Data Fig. 10c and Fig. 4b, where the deletion of IDR regions or mutations in IDR regions of NF-YC9 did not affect the CO/NF-Y tri-component interaction but changed the CO condensation assembly and material properties (Fig. 4c and 4d, Extended Data Fig. 8d). Therefore, the HFD is a core interaction domain, while the IDR region is the regulation domain. We agree with the reviewer adding NF-Y HFD subunits in titration experiments of CO condensate would help to dissect whether the core interaction domain of NF-Y also participates in regulating CO condensation. Indeed, we have shown in vivo where YC9- Δ IDR1&2 shifted the CO condensates to a more slow-diffusion status comparing with full length of YC9 and did not recover flowering phenotype in *yc9* (Extended Data Fig. 9), at least suggesting HFD domain alone is not enough to tune the CO assembly to be fully functional.

To address the second concern, we would like to clarify that our primary focus is on understanding how NF-YC regulates the CO condensation assembly and its material properties, rather than the reverse. Accordingly, in our titration assay, we added NF-Y to a fixed concentration of CO, not vice versa. However, we agree with the reviewer that titrating CO into the NF-Y complex could offer additional insights into the intermolecular regulation of these three components. We have included a discussion on this point in the revised manuscript on page 17.

For this, in figure 2 - and in all figures indicating the proportion of liquid/slow-diffusive condensates, e.g. fig 4 examining the YC9 polyQ mutants, extended data fig. 8, 9, 10 for delta IDR mutagenesis- the Authors should have indicated the % of nuclei showing the diffuse state, so the reader may better appreciate the distinction between all the different states and their relative proportion within each experimental setup (as in fig 2B first manuscript version, which was replaced by fig 2A with relative frequency distribution), and directly evaluate whether any changes in such diffuse population may or may not be related to the functional state of the assemblies.

We thank the review's excellent suggestion to improve the quality of data presentation. We have now added the related population percentage data in the main text, accordingly. Please see pages 8, 12, 14.

Regarding the newly added data on the simulation of DNA binding of modeled NF-Y-CO mono/oligomeric complexes on the FT promoter (why disordered regions of full-length NF-Y proteins were not included?), it provides some indication of increased DNA association properties of NF-CO oligomers, but it is still to be explained whether such increased association occurs on different sites of the same -or of different- DNA molecules, as they

do not address the relevant point of cooperative oligomeric assembly that might occur on the FT promoter region, possibly promoting the functional (phase separated or simply oligomeric?) state of the complex.

NF-Y IDR regions do not affect its bind with CO and DNA (Extended Data Fig. 8A and 8b). Therefore, to achieve practice simulation system, IDR regions were excluded to directly simulate the core elements of CO/NF-Y complex and DNA molecules. Adding the full length NF-Y would need an extremely long computation time to reach equilibrium while not addressing the targeted points. In addition, the full length of CO was applied in the simulation as it is proven that CO's oligomerization was mainly driven by its N-terminal B-box domain (Zeng et al., 2022). Adding full length of CO allows us to do a structure prediction of oligomerized CO by AlphaFold.

We appreciate the reviewer for raising this intriguing second question. If we understand correctly, the reviewer is interested that whether the CO/NF-Y complex surrounding DNA molecules might provide additional binding sites, leading to cooperative binding and thereby promoting CO condensation. Computer simulations, as complementary techniques, allow us to quickly grasp the interaction patterns of molecules and how they interfere with each other on a larger scale. However, we are unable to model the 3D genome organization within the simulation box to address how different DNA fragments with localized condensing patterns might facilitate further binding between the DNA-binding motif and the CO/NF-Y complex. Given only two copies of FT in the genome, it is challenging to establish a robust theoretical framework for simulating such a system to test multi-DNA cooperative binding, though we cannot completely rule out this possibility. The scenario the reviewer mentioned could occur under physiological conditions where the FT promoters are closely associated with other potential CO-binding fragments. We have included a discussion on this point in the revised manuscript on page 16.

Referee #3 (Report for Author)

The authors performed additional experiments and explanations that have addressed most of my concerns except the following:

1. In the Introduction section, Fang et al., 2019 should also be referred in the sentence "RNA processing of FLOWERING LOCUS C and its antisense transcript (COOLAIR), the major target genes in the vernalization pathway, are regulated by phase-separated regulatory condensates"

We have cited this publication here in the revised manuscript.

2. NF-YC9-GFPs were diffusely distributed in the nuclei of 35S:NF-YC9-GFP yct cells as shown in Extended Data Fig. 1c, but NF-YC9-GFP assembled into smaller foci in 35S:mCherry-CO 35S:NF-YC9-GFP yct plants (Fig. 1b), arguing that endogenous CO does not form condensates and that the condensates observed are resulted from CO

overexpression. The percolation of CO, if any, was below optical diffraction limit. The authors should be cautious assuming the small foci being the percolation clusters.

We thank the reviewer pointing out this interesting question. It should be noted endogenous CO is specifically expressed in phloem companion cells but not in the root epidermal cells (An et al., 2004). As such, there is no CO in the nucleus showing in Extended Data Fig. 1c. However, in 35S:mCherry-CO 35S:NF-YC9-GFP/ycT seedlings, CO is ubiquitously expressed under light conditions, which causes the NF-YC9-GFP assembled into small condensates in root epidermal cells. To avoid misleading here, we have added a note in the figure legend of Extended Data Fig. 1c to improve the clarity.

3. Regarding the computer simulation, the authors should explain more clearly what the CO/FT association probability means in Fig. 1e. The authors stated in the Methods that "the binding statistics were performed by counting the number of molecular contacts between the CCT-NF-Y complex and DNA/COREs", is the number of molecular contacts for each CCT-NF-Y complex or an oligomer? if oligomerization increases the contacts for each CCT-NF-Y complex, what's the underlying molecular basis, conformational change?

We apologize for any lack of clarity in our description of the simulation and quantification methods. The numbers of CO/NF-Y complexes (corresponding to monomeric CO/NF-Y complexes) and DNA fragments containing the CO binding sites were the same in the simulation boxes. We quantified the number of CO/NF-Y complexes bound to DNA versus those unbound. The CO/FT association probability indicates the frequency of CO/NF-Y complexes bound to DNA, specifically reflecting the binding between DNA and each monomeric CCT-NF-Y complex, not the oligomer. We proposed that the underlying mechanism might involve reduced diffusion entropy once CO is oligomerized, as the diffusion of each CCT-NF-Y complex within the oligomer is restricted (discussed on page 6). The input structure of the CCT-NF-Y complex in the simulation box was fixed (PDB: 7CVO). Therefore, any conformational changes that occur in the simulation box would only affect DNA molecules, which cannot be fully characterized through simulation alone. The exact underlying mechanism could be diverse (Peeples & Rosen, 2021, Sang, Shu et al., 2022) and would need further study to investigate. Following reviewer's suggestion, we have discussed more on this point at page 16 and revised the description of the simulation methods for clarity in the revised manuscript on page 24.

Reference

- Abramson J, Adler J, Dunger J, Evans R, Green T, Pritzel A, Ronneberger O, Willmore L, Ballard AJ, Bambrick J, Bodenstein SW, Evans DA, Hung C-C, O'Neill M, Reiman D, Tunyasuvunakool K, Wu Z, Žemgulytė A, Arvaniti E, Beattie C et al. (2024) Accurate structure prediction of biomolecular interactions with AlphaFold 3. *Nature* 630: 493-500
- Erijman A, Kozłowski L, Sohrabi-Jahromi S, Fishburn J, Warfield L, Schreiber J, Noble WS, Söding J, Hahn S (2020) A High-Throughput Screen for Transcription Activation Domains Reveals Their Sequence Features and Permits

Prediction by Deep Learning. *Mol Cell* 78: 890-902.e6

Jang G-J, Payne-Dwyer AL, Maple R, Wu Z, Liu F, Lopez SG, Wang Y, Fang X, Leake MC, Dean C (2024) *In vivo* properties of Arabidopsis FCA condensates involved in RNA 3' processing. 2024.04.06.588283

Payne-Dwyer AL, Jang G-J, Dean C, Leake MC (2024) SlimVar: rapid *in vivo* single-molecule tracking of chromatin regulators in plants. 2024.05.17.594710

Peeples W, Rosen MK (2021) Mechanistic dissection of increased enzymatic rate in a phase-separated compartment. *Nat Chem Biol* 17: 693-702

Sang D, Shu T, Pantoja CF, Ibáñez de Opakua A, Zweckstetter M, Holt LJ (2022) Condensed-phase signaling can expand kinase specificity and respond to macromolecular crowding. *Mol Cell* 82: 3693-3711.e10

Zeng X, Lv X, Liu R, He H, Liang S, Chen L, Zhang F, Chen L, He Y, Du J (2022) Molecular basis of CONSTANS oligomerization in FLOWERING LOCUS T activation. *J Integr Plant Biol* 64: 731-740

Dear Prof. Miao,

I am pleased to inform you that your manuscript has been accepted for publication in the EMBO Journal.

Yours sincerely,

Cornelius Schneider, PhD
Editor
The EMBO Journal
c.schneider@embojournal.org
